



Climate
of the Past

# In situ cosmogenic [10]Be–[14]C–[26]Al measurements from recently deglaciated bedrock as a new tool to decipher changes in Greenland Ice Sheet size

Nicolás E. Young[1], Alia J. Lesnek[2], Josh K. Cuzzone[3], Jason P. Briner[4], Jessica A. Badgeley[5],
Alexandra Balter-Kennedy[1], Brandon L. Graham[4], Allison Cluett[4], Jennifer L. Lamp[1], Roseanne Schwartz[1],
Thibaut Tuna[6], Edouard Bard[6], Marc W. Caffee[7,8], Susan R. H. Zimmerman[9], and Joerg M. Schaefer[1]

[1]Lamont–Doherty Earth Observatory, Columbia University, Palisades, NY 10964, USA
[2]Department of Earth Sciences, University of New Hampshire, Durham, NH 03824, USA
[3]Department of Earth System Science, University [CE1] of California Irvine, Irvine, CA 92697, USA
[4]Department of Geology, University at Buffalo, Buffalo, NY 14260, USA
[5]Department of Earth and Space Sciences, University of Washington, Seattle, WA 98195, USA
[6]CEREGE, Aix-Marseille University, CNRS, IRD, INRAE, Collège de France,
Technopôle de l'Arbois, Aix-en-Provence, France
[7]Department of Physics and Astronomy, PRIME Lab, Purdue University, West Lafayette, IN 47907, USA
[8]Department of Earth, Atmospheric, and Planetary Sciences, Purdue University, West Lafayette, IN 47907, USA
[9]Center for Accelerator Mass Spectrometry, Lawrence Livermore National Laboratory, Livermore, CA 94550, USA

**Correspondence:** Nicolás E. Young (nicolasy@ldeo.columbia.edu)

**Abstract.** [TS1]Sometime during the middle to late Holocene (8.2 ka BP to ∼ 1850–1900 CE), the Greenland Ice Sheet (GrIS) was smaller than its current configuration. Determining the exact dimensions of the Holocene ice-sheet minimum and the duration that the ice margin rested inboard of its current position remains challenging. Contemporary retreat of the GrIS from its historical maximum extent in southwestern Greenland is exposing a landscape that holds clues regarding the configuration and timing of past ice-sheet minima. To quantify the duration of the time the GrIS margin was near its modern extent we develop a new technique for Greenland that utilizes in situ cosmogenic [10]Be–[14]C–[26]Al in bedrock samples that have become ice-free only in the last few decades due to the retreating ice-sheet margin at Kangiata Nunaata Sermia ($n = 12$ sites; KNS), southwest Greenland. To maximize the utility of this approach, we refine the deglaciation history of the region with stand-alone [10]Be measurements ($n = 49$) and traditional [14]C ages from sedimentary deposits contained in proglacial threshold lakes. We combine our reconstructed ice-margin history in the KNS region with additional geologic records from southwestern Greenland and recent model simulations of GrIS change to constrain the timing of the GrIS minimum in southwest Greenland and the magnitude of Holocene inland GrIS retreat, as well as to explore the regional climate history influencing Holocene ice-sheet behavior. Our [10]Be–[14]C–[26]Al measurements reveal that (1) KNS retreated behind its modern margin just before 10 ka, but it likely stabilized near the present GrIS margin for several thousand years before retreating farther inland, and (2) pre-Holocene [10]Be detected in several of our sample sites is most easily explained by several thousand years of surface exposure during the last interglaciation. Moreover, our new results indicate that the minimum extent of the GrIS likely occurred after ∼ 5 ka, and the GrIS margin may have approached its eventual historical maximum extent as early as ∼ 2 ka. Recent simulations of GrIS change are able to match the geologic record of ice-sheet change in regions dominated by surface mass balance, but they produce a poorer model–data fit in areas influenced by oceanic and dynamic processes. Simulations that achieve

Published by Copernicus Publications on behalf of the European Geosciences Union.

the best model–data fit suggest that inland retreat of the ice margin driven by early to middle Holocene warmth may have been mitigated by increased precipitation. Triple $^{10}$Be–$^{14}$C–$^{26}$Al measurements in recently deglaciated bedrock provide a new tool to help decipher the duration of smaller-than-present ice over multiple timescales. Modern retreat of the GrIS margin in southwest Greenland is revealing a bedrock landscape that was also exposed during the migration of the GrIS margin towards its Holocene minimum extent, but it has yet to tap into a landscape that remained ice-covered throughout the entire Holocene.

## 1   Introduction

The Greenland Ice Sheet (GrIS) has expanded and contracted repeatedly throughout the Quaternary. During glaciations the GrIS margin extends onto the continental shelf, whereas during interglaciations, the dimensions of the GrIS are often similar to or smaller than today (de Vernal and Hillaire-Marcel, 2008; Hatfield et al., 2016; Knutz et al., 2019). Direct evidence of former GrIS maxima is found in offshore sedimentary deposits (e.g., Ó Cofaigh et al., 2013; Knutz et al., 2019), and the pattern of retreat from the most recent ice-sheet maximum can be reconstructed in detail through a combination of well-dated marine and terrestrial sedimentary archives (Bennike and Bjork, 2002; Funder et al., 2011; Kelley et al., 2013; Hogan et al., 2016; Jennings et a., 2017; Young et al., 2020a). Reconstructing the size and timing of ice-sheet minima, however, is extremely challenging because terrestrial evidence relating to ice-sheet minima has been overrun and destroyed by subsequent glacier re-expansion or resides in a largely inaccessible environment beneath modern glacier footprints. In place of direct terrestrial evidence, sediment-based proxy records contained in offshore depocenters have been used to infer the dimensions and timing of paleo-GrIS minima (Colville et al., 2011; Reyes et al., 2014; Bierman et al., 2016; Hatfield et al., 2016). These sediment-based approaches are not able to provide direct constraints on the magnitude or timing of GrIS minima, but they have the advantage of generally providing continuous records of inferred ice-sheet change.

Cosmogenic isotope measurements from recently deglaciated bedrock surfaces or those still residing under ice provide key constraints on the timing and magnitude of glacier and ice-sheet minima (e.g., Goehring et al., 2010 TS2; Schaefer et al., 2016; Pendleton et al., 2019). These bedrock surfaces serve as fixed benchmark locations, and nuclide accumulation can only occur under extremely thin ice (e.g., in situ $^{14}$C) or, more commonly, in the absence of ice cover when surfaces are exposed to the atmosphere (i.e., a direct ice-margin constraint). The primary caveat of this method, however, is that measured nuclide inventories have non-unique solutions and only provide a measure of integrated surface exposure and burial. Moreover, drilling through

extant glaciers and ice sheets to bedrock is logistically challenging, expensive, and can only be done after lengthy site consideration (e.g., Spector et al., 2018). Nonetheless, groundbreaking measurements of cosmogenic in situ $^{10}$Be and $^{26}$Al in bedrock beneath the GISP2 borehole revealed that the GrIS likely disappeared on several occasions during the Pleistocene (Schaefer et al., 2016).

Contemporary retreat of the GrIS margin from its historical maximum extent is exposing a fresh bedrock landscape, and inventories of cosmogenic nuclides in this newly exposed bedrock can provide clues to past ice-sheet minima without having to drill through ice. Abundant geological evidence reveals that sometime during the middle Holocene, the GrIS was slightly smaller than today (e.g., Weidick et al., 1990; Long et al., 2011; Lecavalier et al., 2014; Larsen et al., 2015; Young and Briner, 2015; Lesnek et al., 2020). The mid-Holocene minimum was forced by regional temperatures that were likely as warm or warmer than today, and elucidating the behavior of the GrIS during this interval can provide key insights into GrIS behavior in a warming world. Bedrock emerging today from beneath the GrIS margin was potentially ice-free during the middle Holocene, and cosmogenic nuclides in these surfaces can constrain the magnitude and duration of inland GrIS retreat.

Here, we present in situ cosmogenic $^{10}$Be–$^{14}$C–$^{26}$Al measurements from recently exposed bedrock surfaces ($n = 12$ sites) in the Kangiata Nunaata Sermia (KNS) forefield, southwestern Greenland (Figs. 1 and 2). Triple $^{10}$Be–$^{14}$C–$^{26}$Al measurements have, to the best of our knowledge, rarely been made (e.g., Miller et al., 2006; Briner et al., 2014) and have not been utilized in any systematic fashion in recently deglaciated environments. To aid interpretation of our $^{10}$Be–$^{14}$C–$^{26}$Al measurements, we refine the early Holocene deglaciation history of the landscape immediately outboard of the historical GrIS maximum extent and constrain when the GrIS retreated inboard of its present position through a combination of stand-alone $^{10}$Be measurements ($n = 49$) and traditional $^{14}$C-dated sediment sequences from proglacial threshold lakes. We combine our new results with previously published records of deglaciation in southwestern Greenland to estimate when the GrIS was behind its present position and reached its minimum extent. We compare geologic records of ice-sheet change to recent model simulations of Holocene GrIS change to further assess the timing and magnitude of mid-Holocene GrIS retreat.

## 2   Settings and methods

### 2.1   Overview

The study region is characterized by mountainous terrain dissected by a dense fjord network in which KNS resides (Fig. 1). Bedrock in the region consists primarily of Archean gneiss (Henriksen et al., 2000). Decades of research have resulted in a robust record of regional deglaciation. Minimum-

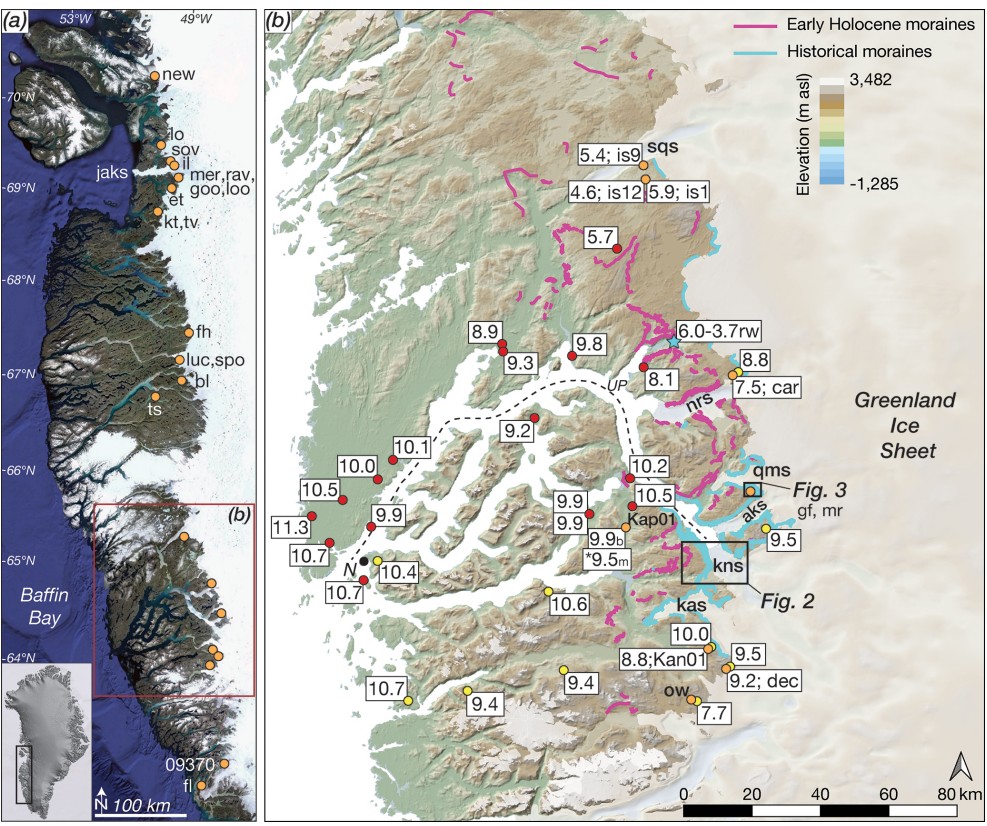

**Figure 1. (a)** Southwestern Greenland, with the locations of proglacial threshold lakes discussed in the text. Map data: © Google Maps, Landsat, US Geological Survey. Jaks: Jakobshavn Isbræ (Sermeq Kujalleq); new: Newspaper Lake (Cronauer et al., 2015); lo: Lake Lo (Håkansson et al., 2014); sov: South Oval Lake, il: Ice Boom Lake, mer: Merganzer Lake, rav: Raven Lake, goo: Goose Lake, loo: Loon Lake, et: Eqaluit taserssuat (Briner et al., 2010); kt: Kuussuup Tasia, tv: Tininnillik Valley (Kelley et al., 2012); fh: Four Hare Lake (Lesnek et al., 2020); luc: Lake Lucy, spo: Sports Lake (Young and Briner, 2015; Lesnek et al., 2020); bl: Baby Loon Lake, ts: Tasersuaq (Lesnek et al., 2020); 09379, fl: Frederikshåb Isblink (Larsen et al., 2015). **(b)** The Nuuk (N) region with existing radiocarbon ages (red dots) and $^{10}$Be ages (yellow dots) in ka BP (1950 CE) that constrain the timing of regional deglaciation. The Kapisigdlit stade moraines are shown in pink, and historical moraines are in blue (modified from Pearce et al., 2018). For figure clarity we only show the mean deglaciation age at each location without uncertainties; see Table S1 (radiocarbon) and Table S2 ($^{10}$Be) for details. Also shown are the locations of proglacial threshold lakes discussed in the text (orange dots) and the location of marine bivalves reworked into the historical maximum (blue star) near Ujarassuit Paavat (up; Weidick et al., 2012; rw indicates reworked). The dotted line marks the flow line used to assess model–data fit in Fig. 18 – is1, is9, and is12 (Levy et al., 2017); car: Caribou Lake, gf: Goose Feather Lake (Lesnek et al., 2020); mr: Marshall Lake (this study); dec: Deception Lake, ow: One-way Lake (Lesnek et al., 2020); Kan01 (Larsen et al., 2015); Kap01 (Larsen et al., 2014 and this study; b: bulk sediments; m: macrofossils). Glaciers are as follows – kns: Kangiata Nunaata Sermia; kas: Kangaasarsuup Sermia; aks: Akullersuup Sermia; qms: Qamanaarsuup Sermia; nrs: Narsap Sermia; sqs: Saqqap Sermia.

limiting radiocarbon ages and $^{10}$Be ages reveal that initial coastal deglaciation occurred at $\sim 11.3$–$10.7$ ka BP and the inner fjord region was ice-free by $\sim 10.5$–$10.0$ ka BP (Fig. 1; Tables S1 and S2 in the Supplement). Punctuating early Holocene deglaciation was deposition of an extensive moraine system during a period locally referred to as the Kapisigdlit stade. Although no direct moraine ages exist, deposition of the Kapisigdlit stade moraines likely occurred sometime between $\sim 10.4$ and $10.0$ ka BP based on the timing of deglaciation from regional radiocarbon and $^{10}$Be constraints and a single maximum-limiting radiocarbon age of $10.17 \pm 0.34$ cal ka BP from reworked marine sed-

iments (Weidick et al., 2012; Larsen et al., 2014; Table S1). Following early Holocene deglaciation, retreat of the GrIS continued inboard of its current margin before readvancing during the late Holocene. In the study area, the GrIS reached its historical maximum extent during the early to mid-18th century (Weidick et al. 2012), which is marked by a prominent moraine and trim line (Weidick et al., 2012; Figs. 1 and 2). In some locations the GrIS still resides at or near its historical maximum extent (Kelley et al., 2012), whereas in the KNS forefield, GrIS retreat from the historical maximum is slightly more pronounced and has exposed fresh bedrock surfaces.

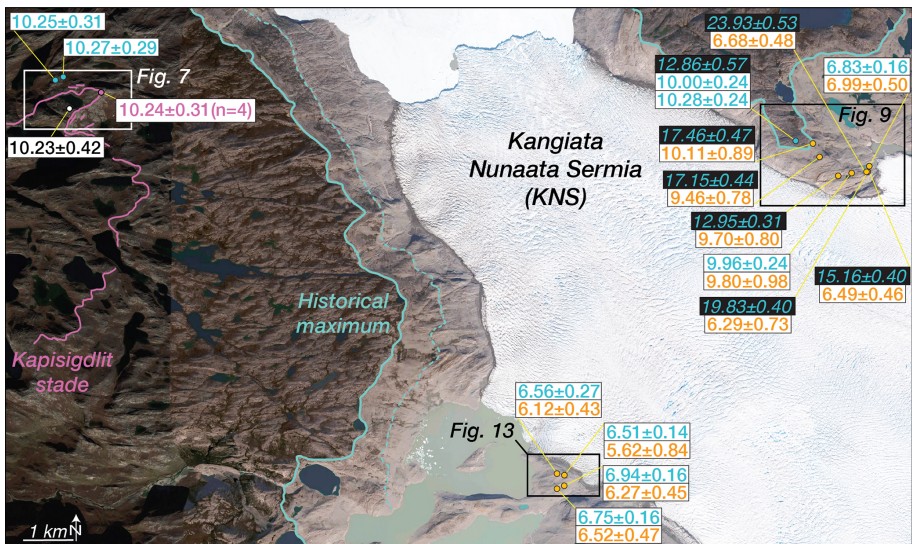

**Figure 2.** Kangiata Nunaata Sermia with the Kapisigdlit stade moraines (pink) and the historical maximum extent (blue), which is marked by prominent moraines and trim lines. The dashed line marks the 1920 CE stade. For figure clarity, we only show $^{10}$Be ages (ka BP ± 1 SD) that constrain deposition of the Kapisigdlit stade moraine and do not include outliers; see Fig. 7 and Table S3 for a detailed view and the complete $^{10}$Be dataset. For bedrock samples located inside the historical maximum extent, $^{10}$Be ages (ka BP ± 1 SD) are shown in blue text, and the corresponding in situ $^{14}$C age from the same sample is in orange text (ka BP ± 1 SD). $^{10}$Be ages influenced by isotopic inheritance are in italics with black boxes. More detailed views of the bedrock sampling locations and the corresponding $^{26}$Al/$^{10}$Be ratios are shown in Figs. 9 and 13. Map data: © Google Maps, Maxar Technologies.

## 2.2  Field methods

Fieldwork was completed in 2017 CE and was primarily concentrated in the KNS forefield and north of KNS at Qamanaarsuup Sermia (Figs. 1–3). In addition, we collected samples for $^{10}$Be dating near Narsap Sermia, located ∼ 55 km north of KNS, and near Kangaasarsuup Sermia located ∼ 20–25 km southwest of KNS (Lesnek et al., 2020; Fig. 1). Moraine crests were mapped prior to fieldwork and updated in the field. This mapping follows previous efforts (Weidick, 1974; Weidick et al., 2012; Pearce et al., 2018), with the exception that we distinguish between early to middle Holocene moraines and moraines marking the GrIS historical maximum extent (Figs. 2 and 3). Across the broader KNS region, the distinction is obvious. Moraines and trim lines attributed to the historical maximum extent of the GrIS are close to the modern ice margin and are fresh in appearance due to a lack of vegetation and lichen cover. Early Holocene moraines are typically located well outboard of the historical moraines and have extensive lichen cover. The key exceptions to this spatial relationship are regions where ice is topographically confined near small outlet glaciers (Figs. 1 and 3). In these locations, the historical maximum and early Holocene moraines are closely stacked yet are still easily distinguishable based on their morphologies and degree of lichen cover (Figs. 3 and 4).

Samples for cosmogenic nuclide analysis were collected using a Hilti brand AG500-A18 angle grinder–circular saw with diamond bit blades, as well as a hammer and chisel.

Sample locations and elevations were recorded with a handheld GPS device with a vertical uncertainty of ±5 m, and topographic shielding was measured using a handheld clinometer. GPS units were calibrated to a known elevation each day, either sea level or the stated elevation of a lake derived from topographic maps.

Sediment cores from two proglacial threshold lakes at Qamanaarsuup Sermia were collected using a universal percussion corer and a Nesje-style percussion–piston coring device (Fig. 3). Goose Feather Lake (informal name) is located ∼ 2 km from the GrIS margin and currently receives GrIS meltwater. We collected two piston cores at a water depth of 12.6 m (17GOOF-A3 and 17GOOF-A4; 64.45328° N, 49.44373° W). Marshall Lake (informal name) is located ∼ 1 km from the GrIS margin and does not presently receive meltwater from the GrIS. We collected two cores from Marshall Lake with the universal percussion corer system at a water depth of 5.95 m (17MAR-A2 and 17MAR-C1; 64.46361° N 49.44373° W).

## 2.3  $^{10}$Be and $^{26}$Al geochemistry and AMS measurements

We completed 61 $^{10}$Be and 12 $^{26}$Al measurements; 54 of the $^{10}$Be samples and all of the $^{26}$Al samples were processed at the Lamont–Doherty Earth Observatory (LDEO) cosmogenic dating laboratory (Tables S3, S4, and S5 in the Supplement). The remaining seven $^{10}$Be samples were processed at the University at Buffalo Cosmogenic Isotope Laboratory

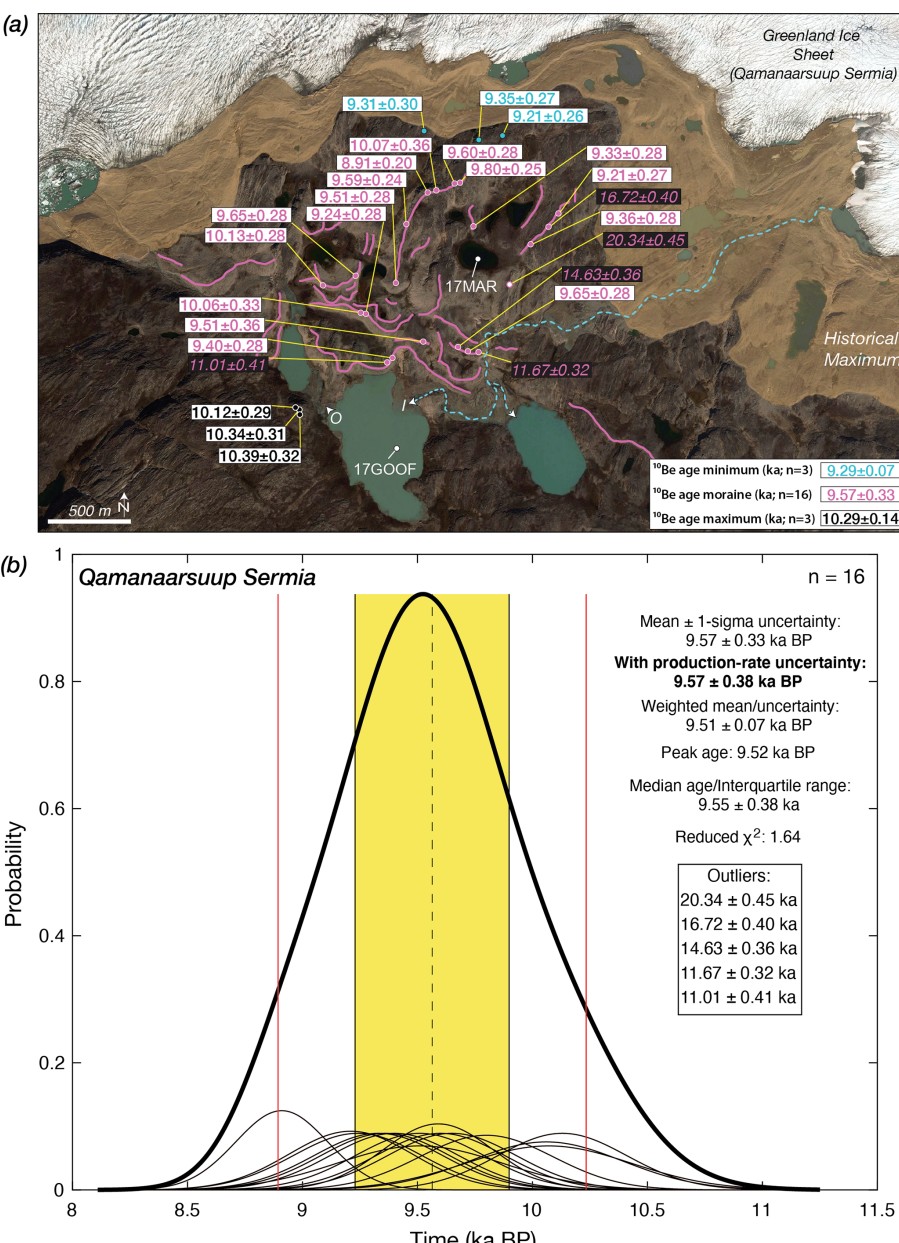

**Figure 3. (a)** Qamanaarsuup Sermia region depicting the Kapisigdlit stade moraines (pink) and retreat since from the historical maximum extent (brown shading); moraines were mapped in the field. $^{10}$Be ages (ka BP ± 1 SD) are from three morphostratigraphic groups: (1) erratic boulders perched on bedrock outboard of the Kapisigdlit stade moraines (black symbols and text), (2) Kapisigdlit stade moraine boulders (pink symbols and text), and (3) erratic boulders perched on bedrock inside the Kapisigdlit stade moraines (blue symbols and text). $^{10}$Be ages influenced by isotopic inheritance are in italics with black boxes. Also shown are the sediment coring locations in Goose Feather Lake (17GOOF) and Marshall Lake (17MAR). The dashed blue line marks the route of meltwater from the GrIS to the Goose Feather Lake inflow (I). The outflow (O) for Goose Feather Lake routes meltwater back towards the GrIS. Map data: © Google Maps, Maxar Technologies. **(b)** Normal density estimates for the Kapisigdlit stade moraines from panel **(a)**. The age in bold includes the production rate uncertainty.

(Tables S3 and S4). In both laboratories, quartz separation as well as Be and Al isolation followed well-established protocols (Schaefer et al., 2009). We quantified the amount of native $^{27}$Al in each quartz aliquot and then added varying amounts of $^{27}$Al carrier to ensure that $\sim 1400$–1750 mg of

$^{27}$Al was achieved (Table S5). Total $^{27}$Al was quantified after sample digestion using inductively coupled plasma optical emission spectrometry analysis of replicate aliquots. AMS analysis for $^{10}$Be samples was split between the Purdue Rare Isotope Measurement (PRIME) Laboratory ($n =$

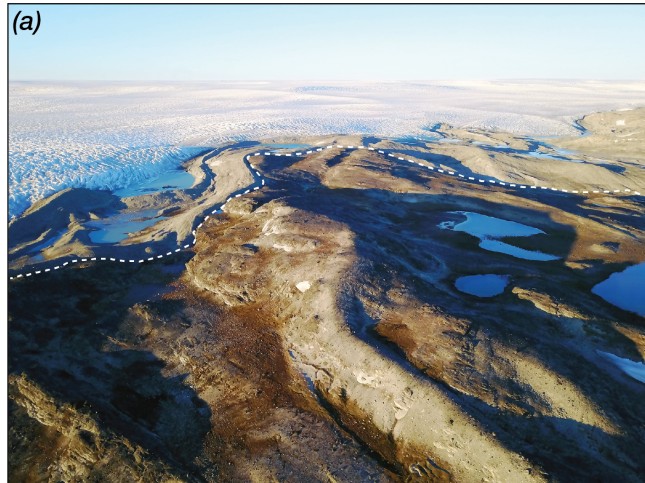

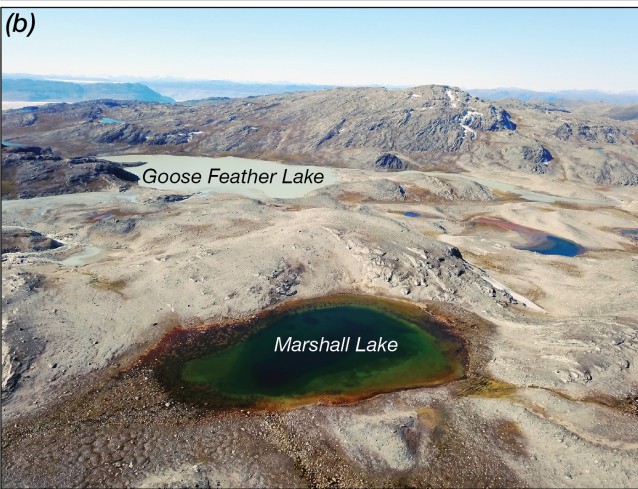

**Figure 4. (a)** View to the northeast in the Qamanaarsuup Sermia region (Fig. 3). In the foreground is a Kapisigdlit stade moraine crest resting directly adjacent to the historical maximum extent of this sector of the GrIS (dashed line). The GrIS is in the background, and there has been minimal retreat from the historical maximum extent here. **(b)** View to the southwest in the Qamanaarsuup Sermia region showing Marshall Lake and Goose Feather Lake. Note the color contrast between the two lakes. Marshall Lake currently does not receive meltwater from the GrIS. Goose Feather Lake is currently a proglacial lake that receives silt-laden GrIS meltwater; the lake catchment currently extends beneath the modern GrIS footprint (Figs. 3a and 6).

36) and the Center for Accelerator Mass Spectrometry at Lawrence Livermore National Laboratory (LLNL-CAMS; $n = 25$); all $^{26}$Al samples were measured at PRIME.

All $^{10}$Be samples were measured relative to the 07KNSTD standard with a $^{10}$Be/$^{9}$Be ratio of $2.85 \times 10^{-12}$ (Nishiizumi et al., 2007), and $^{26}$Al samples were measured relative to the KNSTD standard with the value of $1.82 \times 10^{-12}$ (Nishiizumi, 2004). For $^{10}$Be samples measured at PRIME, the $1\sigma$ analytical error ranged from 1.9 % to 3.9 % with an average of 2.7 % ± 0.5 % ($n = 36$; Table S3), and the $1\sigma$ analytical

error for $^{26}$Al measurements ranged from 1.9 % to 5.0 % with an average of 3.8 % ± 0.9 % ($n = 12$; Table S5). For $^{10}$Be samples measured at LLNL-CAMS, $1\sigma$ analytical error ranged from 1.6 % to 4.2 %, with an average of 2.3 % ± 0.8 % ($n = 25$; Table S3). Process blank corrections for all $^{10}$Be and $^{26}$Al samples were applied by taking the batch-specific blank value (expressed as the number of atoms) and subtracting this value from the sample atom count (Tables S4 and S5). In addition, we propagate through a 1.5 % uncertainty in the carrier concentration when calculating $^{10}$Be concentrations. We assume half-lives of 1.387 and 0.705 Ma for $^{10}$Be and $^{26}$Al (Chmeleff et al., 2010; Nishiizumi, 2004).

## 2.4 In situ $^{14}$C measurements

We completed 12 in situ $^{14}$C extractions at the LDEO cosmogenic dating laboratory following well-established LDEO extraction procedures (Lamp et al. 2019; Table S6 in the Supplement). All measured fraction modern values are converted to $^{14}$C concentrations following Hippe and Lifton (2014). The LDEO in situ $^{14}$C extraction laboratory has historically converted samples to graphite prior to measurement by AMS and LLNL-CAMS; however, we have recently transitioned to gas-source measurements with the AixMICADAS instrument at CEREGE, which can directly measure $\sim 10$–$100 \,\mu g$ C and largely removes the need for the addition of a carrier gas added to typical in situ $^{14}$C samples (Bard et al., 2015; Tuna et al., 2018). Here, two samples underwent traditional graphitization and were measured at LLNL-CAMS (Table S6), whereas the remaining 10 samples were measured with the CEREGE AixMICADAS instrument (Table S6). Both sets of samples underwent the same in situ $^{14}$C extraction and $^{14}$C sample gas clean-up procedures, with only the samples measured at LLNL-CAMS undergoing an additional graphitization procedure (Lamp et al., 2019). Because we use two different measurement approaches and our extraction efforts span the transition between sample graphitization and gas-source measurements, we briefly discuss our data reduction methods for both sets of measurements (Table S6).

Samples 17GRO-14 and 17GRO-74 were measured at LLNL-CAMS with $1\sigma$ analytical uncertainties of 2.2 % and 3.0 % (Table S6). In situ $^{14}$C concentrations were blank-corrected using a long-term mean blank value of $116\,894 \pm 37\,307$ $^{14}$C atoms with the uncertainty in the blank correction propagated in quadrature ($n = 27$; updated from Lamp et al., 2019). In addition, we propagate an additional 3.6 % uncertainty in $^{14}$C concentrations based on the long-term scatter in internal graphite-based CRONUS-A standard measurements ($698\,109 \pm 25\,380$ atoms g$^{-1}$; $n = 13$; updated from Lamp et al., 2019); stated in situ $^{14}$C concentrations for samples measured at LLNL-CAMS have total uncertainties of 7.7 % and 10.4 %, respectively (Table S6).

Samples measured at CEREGE have $1\sigma$ analytical uncertainties that range between 1.0 % and 2.8 % with a mean

of $1.4\% \pm 0.5\%$ ($n = 10$; Table S6). For our gas-source measurements presented here and for future lab measurements, we recharacterized our extraction and measurement procedure with a new set of process blank and CRONUS-A standard measurements (Table S6). We completed six process blank gas-source measurements at CEREGE with values ranging from $\sim 73\,000$–$175\,000$ $^{14}$C atoms (Table S6). One blank measurement is anomalously high ($174\,813 \pm 3582$ atoms; Table S6) and we suspect this blank was contaminated by the atmosphere during collection in a breakseal. The remaining blank values have a mean of $85\,768 \pm 12\,070$ $^{14}$C atoms ($n = 5$), and we tentatively suggest that removing the graphitization procedure may also remove a source of $^{14}$C that was contributing to LDEO background $^{14}$C blank values. Here, we use running-mean blank values of $81\,094 \pm 6972$ ($n = 4$) and $85\,768 \pm 12\,070$ $^{14}$C atoms ($n = 5$) to correct sample $^{14}$C concentrations and uncertainties in the blank corrections are propagated in quadrature (Table S6). In addition, we made five CRONUS-A standard measurements at CEREGE. Our CRONUS-A measurements are remarkably consistent, with a mean value of $662\,132 \pm 9849$ atoms g$^{-1}$ ($n = 5$; 1.5 % uncertainty), and are comparable to our graphite-based value of $698\,109 \pm 25\,380$ atoms g$^{-1}$ (updated from Lamp et al., 2019). Nonetheless, despite the promising consistency of our gas-source CRONUS-A measurements, we conservatively propagate through an additional 3.6 % uncertainty in our sample concentrations based on the scatter in long-term LDEO CRONUS-A measurements. Total $^{14}$C concentration uncertainties for samples measured at CEREGE range from 4.3 % to 5.2 % with a mean uncertainty of $4.6\% \pm 0.2\%$ (Table S6).

## 2.5 $^{10}$Be and in situ $^{14}$C age calculations

$^{10}$Be and in situ $^{14}$C surface exposure ages are calculated using the Baffin Bay $^{10}$Be production rate calibration dataset (Young et al., 2013a) and the West Greenland in situ $^{14}$C production rate calibration dataset (Young et al., 2014). All ages are presented using time-variant "Lm" scaling (Lal, 1991; Stone, 2000), which accounts for changes in the magnetic field, although these changes are minimal at this high latitude ($\sim 64°$ N); using "St" scaling, which does not account for changes in the magnetic field, results in almost identical ages ($< 10$ years) because the calibration sites are all located at high latitudes. The $^{10}$Be and in situ $^{14}$C calibration datasets are both derived from sites in western Greenland with early Holocene exposure histories, and the in situ $^{14}$C calibration measurements are derived from the same geologic samples as one of the $^{10}$Be calibration sites (Young et al., 2014). This combination of calibration datasets ensures that the production rates and the $^{14}$C/$^{10}$Be production ratio are regionally constrained. All ages are calculated in MATLAB using code from version 3 of the exposure age calculator found at https://hess.ess.washington.edu/ TS4, which implements an updated treatment of muon-based nuclide pro-

duction (Balco et al., 2008; Balco, 2017). We do not correct nuclide concentrations for snow cover or subaerial surface erosion; samples are almost exclusively from windswept locations, and many surfaces still retain primary glacial features. In addition, we make no correction for the potential effects of isostatic rebound on nuclide production because both the production rate calibration sites and sites of unknown age have experienced similar exposure and uplift histories (i.e., the correction is "built in"; Young et al., 2020a, b). Individual $^{10}$Be and in situ $^{14}$C ages are presented and discussed with $1\sigma$ analytical uncertainties, and moraine ages exclude the $^{10}$Be production rate uncertainty when we compare other $^{10}$Be-dated features. When moraine ages are compared to independent records of climate variability or ice-sheet change, the production rate uncertainty (1.8 %; Young et al., 2013a) is propagated through in quadrature. To allow for direct comparison to traditional radiocarbon constraints in the region, all $^{10}$Be and in situ $^{14}$C surface exposure ages are presented in thousands of years BP (1950 CE); exposure ages relative to the year of sample collection can be found in the Supplement (2017 CE; Tables S3 and S6).

## 2.6 Traditional $^{14}$C ages from proglacial threshold lakes

Five radiocarbon ages from aquatic macrofossils were obtained from Marshall Lake (Figs. 3 and 4; Table S7 in the Supplement), and one radiocarbon age from an aquatic macrofossil was obtained from lake Kap01 (Fig. 1; Table S7). In addition, we discuss two previously reported radiocarbon ages from Goose Feather Lake, located adjacent to Marshall Lake (Lesnek et al., 2020; Table S7). Aquatic macrofossils were isolated from surrounding sediment using deionized water washes through sieves. Samples were freeze-dried and sent to the National Ocean Sciences Accelerator Mass Spectrometry Facility (NOSAMS) at Woods Hole Oceanographic Institution for age determinations. We targeted aquatic macrofossils for dating because terrestrial macrofossils may persist on the relatively low-energy Arctic landscape for hundreds of years before washing into a lake basin; dating terrestrial macrofossils could skew our interpretations. Hard-water effects on the $^{14}$C ages, which could make age determinations erroneously old, are unlikely in our study area because lake catchments are dominated by Archean gneiss, and the study lakes are all well above local marine limits. All new radiocarbon ages are calibrated using CALIB 8.2 and the INTCAL20 dataset, and previously reported radiocarbon ages are recalibrated in the same manner using the INTCAL20 and MARINE20 datasets (Stuiver et al., 2020; Reimer et al., 2020; Heaton et al., 2020; Tables S1 and S7).

## 2.7 Ice-sheet model simulations of southwestern GrIS change

We utilize recent paleo-simulations of southwestern GrIS change using the high-resolution Ice Sheet and Sea-level System Model (ISSM; Larour et al., 2012; Cuzzone et al., 2018, 2019; Briner et al., 2020). The model setup has been previously described in Cuzzone et al. (2019) and Briner et al. (2020), but here we briefly describe model attributes. The model domain extends from the present-day coastline to the GrIS divide. The northern and southern boundaries of the domain are far to the north and south of our study area. The model resolution relies on anisotropic mesh adaptation to produce an unstructured mesh that varies based on bedrock topography; bedrock topography is from BedMachine v3 (Morlighem et al., 2017). For the southwestern GrIS, high horizontal model mesh resolution is necessary in areas of complex bed topography to prevent artificial ice-margin variability resulting from interaction with bedrock artifacts that occur at coarser resolution (Cuzzone et al., 2019). Thus, the model mesh varies from 20 km in areas where gradients in the bedrock topography are smooth to 2 km in areas where bedrock relief is high. In the KNS region the mesh varies from 2 to 8 km.

The ice model applies a higher-order approximation (Blatter 1995; Pattyn 2003) to solve the momentum balance equations and an enthalpy formulation (Ashwanden et al., 2012), with geothermal heat flux from Shapiro and Ritzwoller (2004), to simulate the thermal evolution of the ice. Quadratic finite elements (P1 × P2) are used along the $z$ axis for the vertical interpolation, which allows the ice-sheet model to capture sharp thermal gradients near the bed, while reducing computational costs associated with running a linear vertical interpolation with increased vertical layers (Cuzzone et al., 2018). Sub-element grounding-line migration (Serrousi et al., 2013) is included in these simulations; however, due to prohibitive costs associated with running a higher-order ice model over paleoclimate timescales these simulations do not include calving parameterizations or any submarine melting of floating ice.

Nine ice-sheet simulations are forced with paleoclimate reconstructions from Badgeley et al. (2020), who used paleoclimate data assimilation to merge information from paleoclimate proxies and global climate models. The temperature reconstructions rely on oxygen isotope records from eight ice cores; the precipitation reconstructions use accumulation records from five ice cores, and all are guided by spatial relationships derived from the transient climate model simulation TraCE-21ka (Liu et al., 2009; He et al., 2013). The climate reconstructions are shown to be in good agreement with independent paleoclimate proxy data (Badgeley et al., 2020, and references therein). Along with a main temperature and precipitation reconstruction, Badgeley et al. (2020) provide two sensitivity precipitation reconstructions due to uncertainty in the accumulation records and four sensitivity temperature reconstructions due to uncertainty in the relationship between oxygen isotopes and surface air temperature. Briner et al. (2020) pair three of the temperature reconstructions with each of the three precipitation reconstructions to yield nine combinations that are used as transient climate boundary conditions to force the nine ice-sheet simulations. Two of the five temperature reconstructions were not used because they yield Younger Dryas ice-sheet margins that are inconsistent with geologic data.

We use a positive degree day (PDD) method (Tarasov and Peltier, 1999) to compute the surface mass balance from temperature and precipitation, and we use degree day factors of 4.3 mm °C$^{-1}$ d$^{-1}$ for snow and 8.3 mm °C$^{-1}$ d$^{-1}$ for ice, with allocation for the formation of superimposed ice (Janssens and Huybrechts, 2000). We use a lapse rate of 6 °C km$^{-1}$ to adjust the temperature of the climate forcings to the ice-surface elevation.

## 3 Results

Adjacent to Qamanaarsuup Sermia, 27 $^{10}$Be ages from moraine boulders and boulders perched on bedrock range from $20.34 \pm 0.45$ ka BP to $8.91 \pm 0.20$ ka BP (Figs. 3 and 5; Table S3). Sediments in Goose Feather Lake are composed of a lower gray silt unit overlain by organic sediments, which is in turn overlain by gray silt (Lesnek et al., 2020). A single radiocarbon age from bulk sediments at the basal sediment contact is $8280 \pm 90$ cal yr BP, and a radiocarbon age from aquatic macrofossils at the upper contact between organic and minerogenic sediments is $820 \pm 80$ cal yr BP. Sediments in Marshall Lake display the same silt–organic–silt stratigraphy as sediments in Goose Feather Lake (Fig. 6). A radiocarbon age from aquatic macrofossils from the basal sediment contact is $8720 \pm 350$ cal yr BP, and a radiocarbon age from aquatic macrofossils at the uppermost contact is $520 \pm 20$ cal yr BP (Fig. 6). Three additional radiocarbon ages from aquatic macrofossils between the lowermost and uppermost contacts are $7250 \pm 70$, $3650 \pm 50$, and $940 \pm 20$ cal yr BP and are in stratigraphic order (Fig. 6).

In the KNS region, 27 $^{10}$Be ages from moraine boulders, erratics perched on bedrock, and abraded bedrock surfaces range from $23.93 \pm 0.53$ to $5.38 \pm 0.23$ ka BP (Table S3), and 12 in situ $^{14}$C ages from bedrock range from $10.11 \pm 0.89$ to $5.62 \pm 0.84$ ka BP (Table S6). In addition, 12 $^{26}$Al–$^{10}$Be ratios range from $7.35 \pm 0.33$ to $6.01 \pm 0.25$ (all bedrock). A single radiocarbon age from aquatic macrofossils at the basal contact between silt and organic sediments in lake Kap01 is $9450 \pm 440$ cal yr BP (Fig. 1; Table S7).

North of Narsap Sermia near Caribou Lake (Fig. 1), three $^{10}$Be ages from boulders perched on bedrock are $9.07 \pm 0.32$, $8.66 \pm 0.31$, and $8.66 \pm 0.31$ ka BP (Fig. 1; Table S3). South of KNS at Deception Lake, two $^{10}$Be ages from boulders perched on bedrock are $10.66 \pm 0.34$ and $9.52 \pm 0.32$ ka BP, and near One-way lake, two $^{10}$Be ages from boulders perched

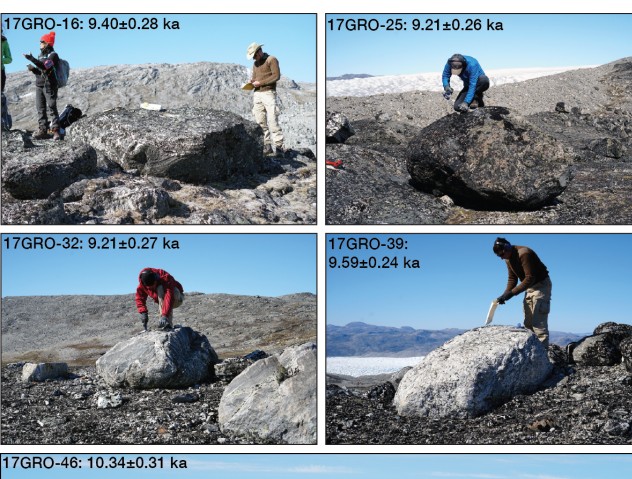

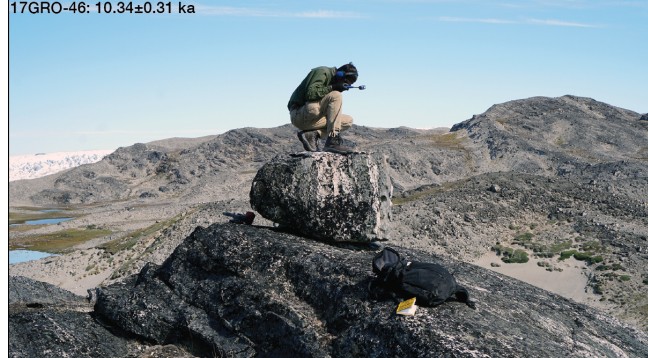

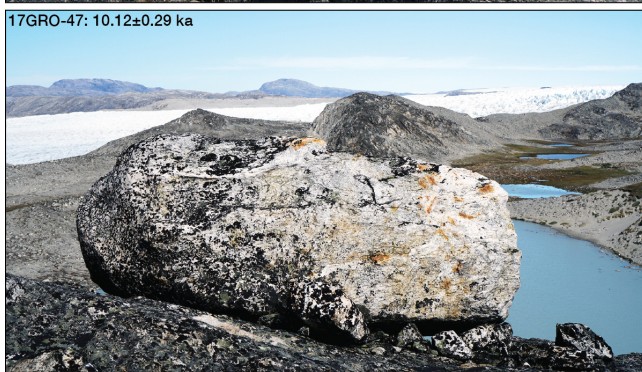

**Figure 5.** Representative boulder samples from the Qamanaarsuup Sermia region (Fig. 3). Samples 17GRO-16, 17GRO-32, and 17GRO-39 are moraine boulder samples. 17GRO-25 is an erratic boulder perched on bedrock inside the Kapisigdlit stade moraine located only a few meters outboard of the historical maximum moraine, which can be seen in the background. Samples 17GRO-46 and 17GRO-47 are erratic boulders resting on bedrock located outboard of the Kapisigdlit stade moraine.

on bedrock are $16.43 \pm 0.49$ and $7.72 \pm 0.26$ ka BP (Fig. 1; Table S3).

## 4 Deglaciation chronologies

### 4.1 Qamanaarsuup Sermia

A total of 27 $^{10}$Be ages from the Qamanaarsuup Sermia region range $20.34 \pm 0.45$ to $8.91 \pm 0.20$ ka BP (Figs. 3 and 5; Table S3); however, the $^{10}$Be ages are from three distinct morphostratigraphic units. Here, the Kapisigdlit stade is marked by numerous closely spaced moraine crests located immediately outboard of the historical moraines. Three $^{10}$Be ages from boulders resting on bedrock located outside the entire Kapisigdlit moraine suite are $10.39 \pm 0.32$, $10.34 \pm 0.31$, and $10.12 \pm 0.29$ ka BP, and they have a mean age of $10.29 \pm 0.14$ ka BP, which serves as a maximum-limiting age for the Kapisigdlit stade moraines in the region. Three $^{10}$Be ages from boulders resting on bedrock immediately inboard of all Kapisigdlit stade moraines, but outboard of the historical moraine, are $9.35 \pm 0.27$, $9.31 \pm 0.30$, and $9.21 \pm 0.26$ ka BP; they provide a minimum-limiting age of $9.29 \pm 0.07$ ka BP (Fig. 3). Of the 21 $^{10}$Be ages from Kapisigdlit stade moraine boulders, 5 $^{10}$Be ages are likely influenced by $^{10}$Be inheritance as they are older than the maximum-limiting $^{10}$Be ages and similar in age to deglacial constraints found $\sim 140$ km west at the modern coastline ($11.67 \pm 0.32$ and $11.01 \pm 0.41$ ka BP; Figs. 1 and 3; Table S3), or they date to when the GrIS margin was likely situated $> 140$ km to the west somewhere on the continental shelf ($20.34 \pm 0.45$, $16.72 \pm 0.40$, and $14.63 \pm 0.36$ ka BP; Figs. 1 and 3; Table S3). The remaining 16 $^{10}$Be ages from the Kapisigdlit moraine set show no trend with distance from the ice margin. These $^{10}$Be ages overlap at $1\sigma$ uncertainties with each other, the minimum-limiting $^{10}$Be ages, or the maximum-limiting $^{10}$Be ages, suggesting that deposition of this suite of moraine crests happened within dating resolution (i.e., we cannot resolve the ages of different moraine crests). Combined, the 16 $^{10}$Be ages from moraine boulders, excluding outliers, have a mean age of $9.57 \pm 0.33$ ka BP, which is morphostratigraphically consistent with bracketing maximum- and minimum-limiting $^{10}$Be ages of $10.29 \pm 0.14$ and $9.29 \pm 0.07$ ka BP, respectively (Fig. 3). Including the uncertainty in the $^{10}$Be production rate calibration, $^{10}$Be ages from the Qamanaarsuup Sermia region reveal that the GrIS margin approached its modern extent at $10.29 \pm 0.23$ ka BP, deposited the Kapisigdlit stade moraines at $9.57 \pm 0.38$ ka BP, and retreated behind the position of the historical maximum at $9.29 \pm 0.18$ ka BP.

Alternating silt–organic–silt sediment packages found in Goose Feather and Marshall lakes are typical of those found in proglacial threshold lakes in southwestern Greenland (Fig. 6; Briner et al., 2010; Larsen et al., 2015; Young and Briner, 2015; Lesnek et al., 2020). Silt deposition occurs when the GrIS margin resides within the lake catchment but does not override the lake, feeding silt-laden meltwater into the lake. Organic sedimentation occurs when the GrIS margin is not within the lake catchment and meltwater is diverted

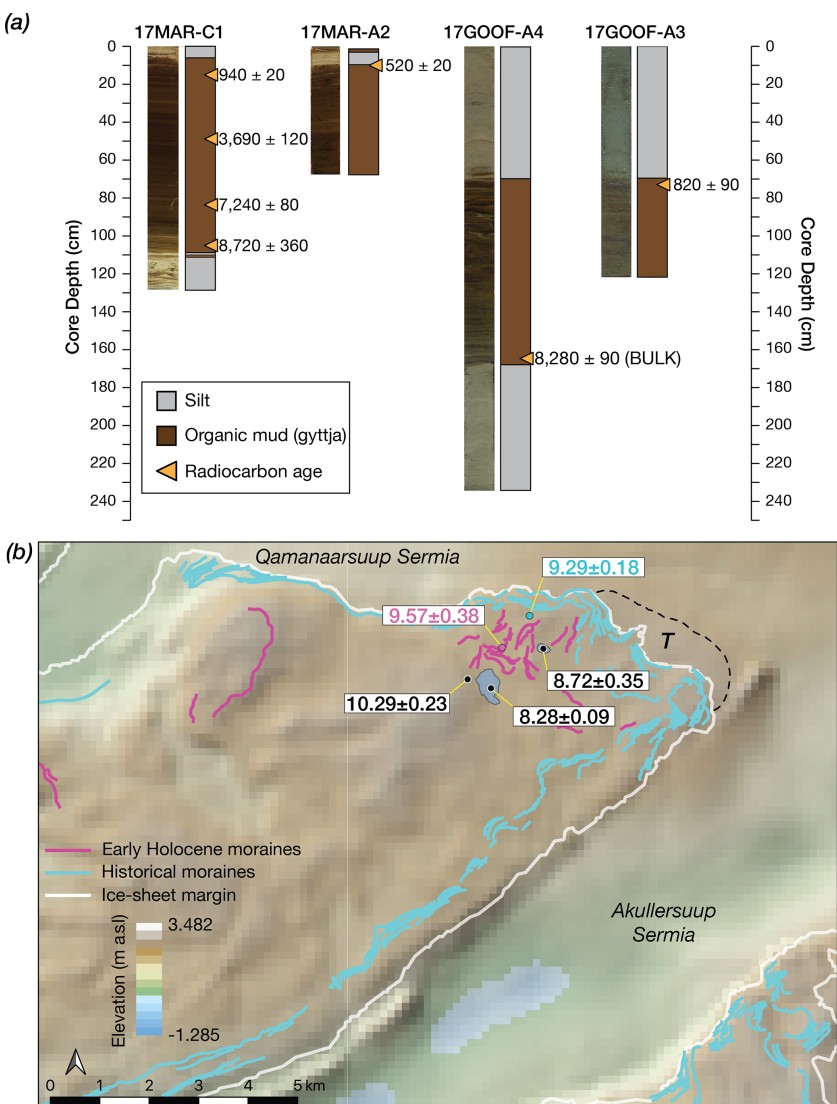

**Figure 6. (a)** Sediment cores and calibrated radiocarbon ages ($\pm 2\sigma$) from Marshall (MAR) and Goose Feather (GOOF) lakes. Note the distinct color differences between silt and organic sediments (see Figs. 3 and 4). Details for radiocarbon ages can be found in Table S7. **(b)** Sub-ice topography in the Qamanaarsuup Sermia region generated using the BedMachine v3 DEM CE3 (Morlighem et al., 2017) compared to our chronology of ice-margin change developed from $^{10}$Be ages and radiocarbon-dated lake sediments (panel **a**). Shown is the average $^{10}$Be age from each feature on the landscape (including production rate uncertainty; see Fig. 3) and basal radiocarbon ages from panel **(a)**. Deglaciation of the landscape just outboard of the Kapisigdlit stade moraines occurred at $10.29 \pm 0.23$ ka BP, followed by moraine deposition at $9.57 \pm 0.38$ ka BP. Ice retreated behind the modern margin at $9.29 \pm 0.18$ ka BP, but the ice margin remained within the drainage catchment of Goose Feather Lake until $8.28 \pm 0.09$ cal ka BP before retreating farther inland. The dashed line delimits the topographic threshold (T) under the modern GrIS that the ice margin must cross in order for Goose Feather Lake to receive silt-laden meltwater. Inflow of meltwater ceases when the GrIS margin retreats behind this topographic threshold, which rests $\sim 1$ km behind the modern margin.

elsewhere. Despite Goose Feather and Marshall lakes residing adjacent to each other on the landscape, their radiocarbon ages suggest slightly different ice-margin histories. Marshall Lake has a small and highly localized drainage catchment and does not receive GrIS meltwater at present. In contrast, because Goose Feather Lake currently receives GrIS meltwater, its drainage catchment extends somewhere beneath the modern GrIS. Goose Feather Lake is fed by meltwater

sourced from an outlet glacier resting in an overdeepening, and once ice thins below the valley edge, GrIS meltwater is diverted elsewhere, likely indicating that the Goose Feather drainage divide resides near the modern ice margin (e.g., Young and Briner, 2015; Lesnek et al., 2020). Indeed, sub-ice topography in the Qamanaarsuup Sermia region reveals that the topographic threshold that dictates whether meltwater is diverted to Goose Feather Lake or elsewhere is located

within $\sim 1$ km of the modern ice margin (Fig. 6). Despite the GrIS margin retreating behind the position of the historical maximum position at $9.29 \pm 0.18$ ka BP, silt deposition in Goose Feather Lake until $8280 \pm 90$ cal yr BP indicates that the ice margin remained within $\sim 1$ km of its present position between $\sim 9.3$ and $8.3$ ka (Fig. 6).

## 4.2  Kangiata Nunaata Sermia

A total of 15 [10]Be ages in the KNS region constrain the timing of deposition of the Kapisigdlit stade moraines and the timing of when the GrIS retreated behind the eventual historical maximum limit (Figs. 2 and 7–9). West of the KNS terminus, three [10]Be ages from erratic boulders perched on bedrock located immediately outboard of the Kapisigdlit stade moraine are $11.44 \pm 0.34$, $11.15 \pm 0.34$, and $10.23 \pm 0.42$ ka BP. Seven [10]Be ages from moraine boulders range from $11.22 \pm 0.35$ to $5.38 \pm 0.23$ ka BP, and two [10]Be ages from erratic boulders perched on bedrock located immediately inside the moraine are $10.27 \pm 0.29$ and $10.25 \pm 0.31$ ka BP (Fig. 7). Deglaciation of the outer coast at Nuuk occurred at $\sim 11.3$–$10.7$ ka BP, suggesting that the single [10]Be age of $10.23 \pm 0.42$ ka BP from just outboard of the Kapisigdlit stade moraine is likely the closest maximum-limiting age on moraine deposition, and [10]Be ages of $11.44 \pm 0.34$ and $11.15 \pm 0.34$ ka BP are likely influenced by a slight amount of inheritance (Figs. 1 and 7). This maximum-limiting [10]Be age is consistent with a maximum-limiting radiocarbon age of $10\,170 \pm 340$ cal yr BP from a bivalve reworked into Kapisigdlit stade till located downfjord (Weidick et al., 2012; Larsen et al., 2014; Fig. 1; Table S1). The 10.23 ka BP age is also consistent with the [10]Be age of boulders outboard of the Kapisigdlit stade moraines in the Qamanaarsuup Sermia study area of $\sim 10.29$ ka BP. There is significant scatter in our moraine boulder [10]Be ages; the oldest [10]Be age of $11.22 \pm 0.35$ ka BP is likely influenced by isotopic inheritance, whereas the younger outliers of $6.73 \pm 0.18$ and $5.38 \pm 0.23$ ka BP reflect post-depositional boulder exhumation (Fig. 7). The remaining [10]Be ages from the Kapisigdlit stade moraine have a mean age of $10.24 \pm 0.31$ ka BP ($n = 4$), which is supported by our minimum-limiting [10]Be ages of $10.27 \pm 0.29$ and $10.25 \pm 0.31$ ka BP from immediately inside the moraine (Fig. 7). Indeed, [10]Be ages from erratic boulders perched on bedrock located immediately inside moraines across southwestern Greenland typically provide constraints that are nearly identical to tightly clustered [10]Be ages from moraine boulders (e.g., Young et al., 2011a, 2013b; Lesnek and Briner, 2018). Furthermore, our statistically identical [10]Be ages from outboard and inboard of the Kapisigdlit stade moraine, as well as from moraine boulders themselves, indicate that moraine deposition occurred rapidly within the resolution of our chronometer. Including the production rate uncertainty, we directly date the Kapisigdlit stade moraine to $10.24 \pm 0.36$ ka BP, and

all available supporting [10]Be and [14]C ages further constrain moraine deposition to $\sim 10.4$–$10.2$ ka BP.

On the east side of KNS, three [10]Be ages from immediately outboard of the historical maximum are $12.86 \pm 0.57$, $10.28 \pm 0.24$, and $10.00 \pm 0.24$ ka BP (Fig. 9). The [10]Be age of $12.86 \pm 0.57$ ka BP is, again, almost certainly influenced by inheritance as this [10]Be age predates the timing of deglaciation at the outer coastline. The remaining [10]Be ages of $10.28 \pm 0.24$ and $10.00 \pm 0.24$ ka BP are consistent with the [10]Be ages from inside the Kapisigdlit stade moraine (and outboard of the historical maximum limit) on the west side of the fjord. Our new and previously published age constraints reveal that deposition of the Kapisigdlit stade moraine in the KNS forefield occurred at ca. 10.4–10.2 ka, followed by retreat of the GrIS within the historical maximum limit shortly thereafter (Fig. 2). Any possible moraine correlative with the $\sim 9.6$ ka moraine found at Qamanaarsuup Sermia would have been overrun by the historical advance of KNS.

Lastly, a basal minimum-limiting radiocarbon age of $9450 \pm 440$ cal yr BP from Kap01 is, within uncertainties, identical to a previously reported basal radiocarbon age of $9850 \pm 290$ cal yr BP from the same lake (Tables S1 and S7; Larsen et al., 2014). We note that our new radiocarbon age is from aquatic macrofossils, whereas the previously published radiocarbon age is from bulk sediments (humic acid extracts). Despite the risk of bulk sediments yielding radiocarbon ages that are too old, basal radiocarbon ages from Kap01 suggest that the offset between macrofossil- and bulk-sediment-based radiocarbon ages is likely minimal during the initial onset of organic sedimentation following landscape deglaciation. We do not advocate for the use of bulk sediments to develop down-core chronologies, but paired macrofossil–bulk sediment measurements from the same horizon often yield similar or indistinguishable radiocarbon ages in southwestern Greenland (e.g., Kaplan et al., 2002; Young and Briner, 2015), suggesting that bulk sediments will not produce significantly erroneous radiocarbon ages in this region. These similarities in southwestern Greenland likely result from several factors: (1) a large fraction of humic acid extracts are aquatic in origin (Wolfe et al., 2004); (2) southwestern Greenland is composed almost entirely of crystalline bedrock, thereby minimizing potential hard-water effects; and (3) there is no significant accumulated carbon pool during the initial phase of ecosystem development (i.e., Wolfe et al., 2004). This latter point may be particularly influential in southwestern Greenland because this region rests well inboard of the GrIS margin during glacial maxima (located on the continental shelf), resulting in a landscape that is likely ice-covered for a significant fraction of each glacial cycle. Furthermore, this sector of the GrIS is primarily warm-based and erosive, thereby further minimizing the likelihood of old carbon accumulating on the landscape at lower elevations.

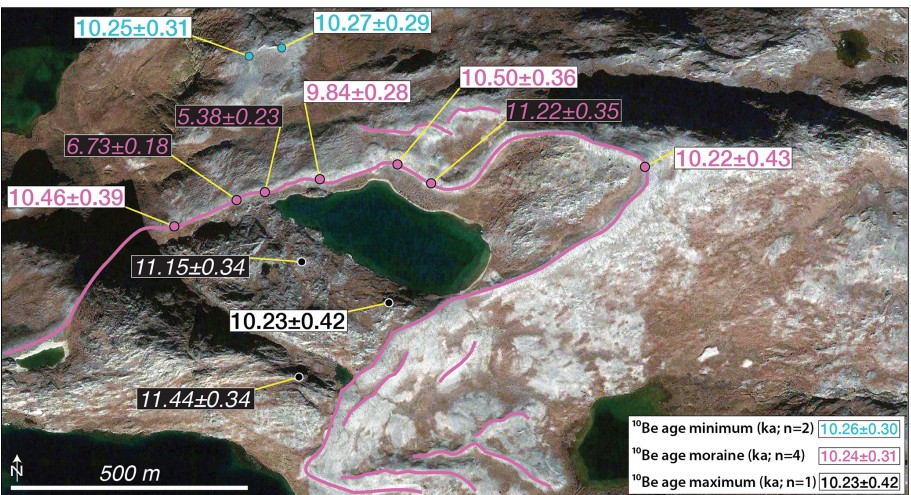

**Figure 7.** Kapisigdlit stade moraine located just west of KNS (Fig. 2). $^{10}$Be ages (ka BP $\pm$ 1 SD) are from three morphostratigraphic groups: (1) outboard of the moraine (black text and symbols), (2) moraine boulders (pink text and symbols), and (3) inside the moraine (blue text and symbols). $^{10}$Be ages that are considered outliers are in black boxes with italics. Map data: © Google Maps, Maxar Technologies.

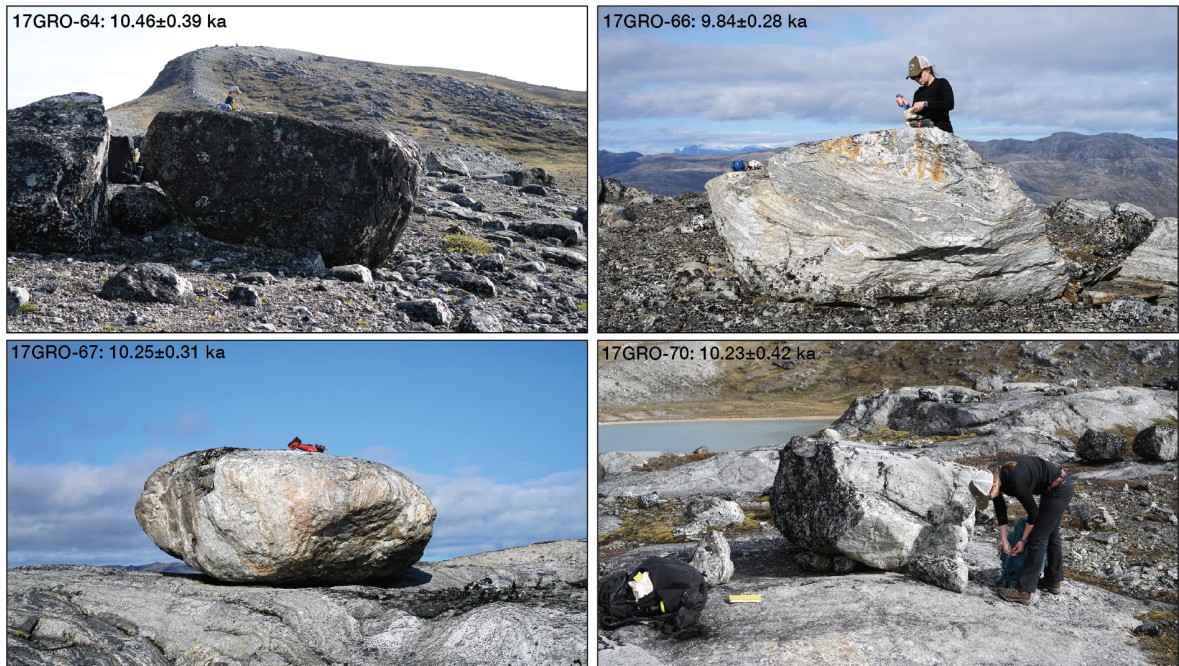

**Figure 8.** Representative boulder samples related to the Kapisigdlit stade moraine west of KNS. 17GRO-64 and 17GRO-66 are moraine boulders, whereas 17GRO-67 and 17GRO-70 are erratic boulders resting on bedrock located immediately inboard and outboard of the Kapisigdlit stade moraine.

## 4.3 Auxiliary sites

At our site near Narsap Sermia, located ∼ 55 km north of KNS, three $^{10}$Be ages from boulders perched on bedrock located outboard of the GrIS historical maximum and inboard of the Kapisigdlit stade limit have a mean age of $8.80 \pm 0.24$ ka BP ($8.80 \pm 0.29$ ka BP including the production rate uncertainty; Table S3), consistent with a minimum-limiting

basal radiocarbon age of $7460 \pm 110$ cal yr BP (Lesnek et al., 2020; Fig. 1; Table S1). Near Kangaasarsuup Sermia, located ∼ 35 km south of KNS, two $^{10}$Be ages from boulders perched on bedrock outboard of the historical limit and inboard of the Kapisigdlit stade limit are $10.66 \pm 0.34$ and $9.52 \pm 0.32$ ka BP. An existing $^{10}$Be and traditional $^{14}$C age of $9.95 \pm 0.19$ ka BP ($n = 2$) and $8790 \pm 190$ cal yr BP from locations slightly more distal from the ice sheet suggest that our

## 5 $^{10}$Be–$^{14}$C–$^{26}$Al measurements from the KNS forefield

Prior to interpreting triple $^{10}$Be–$^{14}$C–$^{26}$Al measurements in abraded bedrock located between the historical maximum extent and the current ice margin, we use our new chronology of early Holocene ice-margin change and historical observations to quantify the maximum duration of Holocene exposure our bedrock samples sites could have experienced. First, we use $^{10}$Be ages from immediately outboard of the historical limit on the northeastern side of KNS and $^{10}$Be ages from just inboard of the Kapisigdlit stade moraine on the southwestern side of KNS to define the potential onset of Holocene exposure at our $^{10}$Be–$^{14}$C–$^{26}$Al bedrock sites. $^{10}$Be ages from outboard of the historical limit overlap at $1\sigma$ uncertainties ($n = 4$; excluding one outlier), and we calculate a mean age of $10.20 \pm 0.14$ ka BP ($10.20 \pm 0.23$ ka BP with production rate uncertainty) as the earliest onset of exposure at our inboard bedrock sites (Fig. 2). This age represents the timing of deglaciation immediately outboard of the historical limit and, assuming the continued retreat of the GrIS margin, the initial timing of exposure for the inboard bedrock sites (e.g., Young et al., 2016). Next, we capitalize on historical observations in the KNS region that constrain ice-margin change beginning in the 18th century (Weidick et al., 2012; Lea et al., 2014, and references therein). Based on scattered first-person observations, the advance towards the eventual historical maximum extent likely began by 1723–1729 CE and culminated in ca. 1750 CE, with initial ice-margin thinning taking place at ca. 1750–1800 CE (Weidick et al., 2012). Broadly supporting this record of ice-margin migration is an early photograph by Danish geologist Hinrich Rink dated to sometime in the 1850s (Fig. 10). The photograph depicts the front of KNS as seen from the northwest and clearly delineates an existing historical maximum trim line, indicating that the local GrIS historical maximum was achieved and initial thinning from this maximum began prior to 1850 CE (Fig. 10; Weidick et al., 2012). Additional first-person descriptions indicate that the KNS ice margin was more extended than today between ca. 1850 CE and at least 1948 CE, punctuated by the 1920 CE stade, which marks a significant readvance of the ice margin (Fig. 2). Aerial photographs reveal that the ice margin was only a few tens of meters east of our bedrock sites on the western side of KNS at 1968 CE, suggesting site deglaciation shortly beforehand (Weidick et al., 2012). Our eastern ice-marginal bedrock sites were likely ice-free in ca. 2000 CE based on satellite imagery.

The majority of our bedrock sites are directly adjacent to the ice margin, and we assume that pre-imagery historical observations of ice-margin change apply to both sampling regions because any differences in ice-covered and ice-free intervals between the two sampling sites are likely negligible for our purposes. The available historical constraints indicate that our bedrock sites became ice-covered in ca. 1725 CE (historical maximum advance phase), and our sites

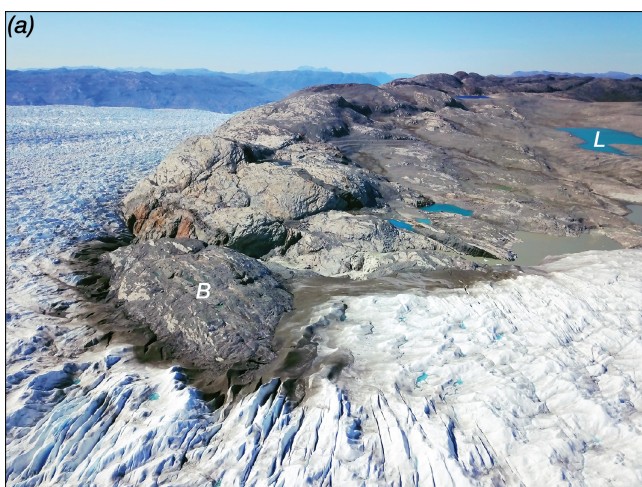

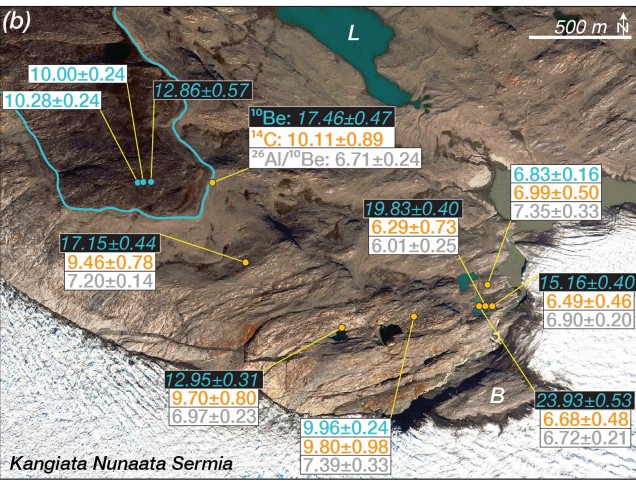

**Figure 9. (a)** Oblique aerial view to the northwest depicting recently deglaciated bedrock that rests between the historical maximum extent and the modern ice margin on the northeastern side of KNS. **(b)** $^{10}$Be and in situ $^{14}$C ages (ka BP $\pm$ 1 SD), along with measured $^{26}$Al/$^{10}$Be ratios, at each bedrock sample site. $^{10}$Be ages influenced by inheritance are in black boxes with italics. The historical maximum extent of KNS is marked in blue. To orient the reader, L and B mark the same feature in each panel. Map data: © Google Maps, Maxar Technologies.

age of $10.66 \pm 0.34$ ka BP is perhaps influenced by a small amount of isotopic inheritance (Larsen et al., 2014; Fig. 1; Tables S2 and S3). The remaining age of $9.52 \pm 0.32$ ka BP is consistent with a minimum-limiting basal radiocarbon age of $9210 \pm 190$ cal yr BP from Deception Lake (Fig. 1; Lesnek et al., 2020). Lastly, south of Kangaasarsuup Sermia near One-way Lake, two $^{10}$Be ages from boulders perched on bedrock outboard of the historical limit are $16.43 \pm 0.49$ and $7.72 \pm 0.26$ ka BP (Fig. 1; Table S3). The older of these two $^{10}$Be ages is influenced by isotopic inheritance, leaving a single $^{10}$Be age of $7.72 \pm 0.26$ ka BP as the only estimate for the timing of local deglaciation.

*(a)*

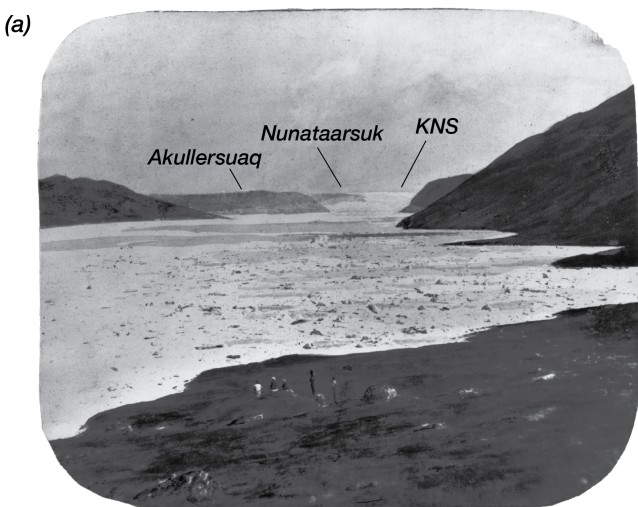

*(b)*

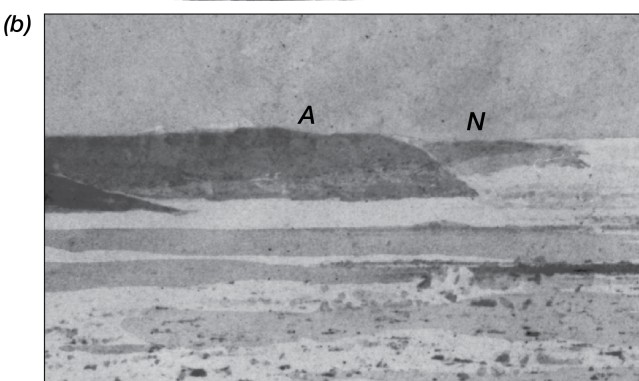

**Figure 10. (a)** Photograph looking up-fjord towards KNS taken some time in the 1850s by Danish geologist Hinrich Rink (Weidick et al., 2012). Our northeastern KNS field site (Fig. 9) is located on the distal side of Nunaatarsuk. **(b)** Close-up of Akullersuaq (A) and Nunaatarsuk (N) that captures the trim line marking the historical maximum extent of KNS. The photograph is housed in the archives of the National Museum in Copenhagen. A digital copy was graciously provided by O. Bennike.

on the western side of KNS likely became ice-free in ca. 1968 CE; ice-marginal sites on the east side of KNS became ice-free in ca. 2000 CE. These observations indicate that the western bedrock sites experienced 243 years of historical ice cover, whereas the eastern sites experienced 275 years of ice cover. With the earliest possible onset of exposure occurring at $10.20 \pm 0.23$ ka BP as constrained by our $^{10}$Be ages, we assume that the maximum duration of Holocene surface exposure at all of our sites is 10.0 kyr. We do note, however, that three of our bedrock sampling sites on the northeastern side of KNS are at a higher elevation and lie closer to the historical maximum limit than the sites adjacent to the modern ice margin (Figs. 2 and 9). These high-elevation sites almost certainly experienced shorter historical ice cover than our lower-elevation ice-marginal sites, likely on the order of $\sim 150$–200 years. But considering the un-

certainties in our chronology and analytical detection limits, we assume they have the same maximum Holocene exposure duration of 10 kyr. Lastly, samples could inherit cosmogenic nuclides from an earlier exposure (i.e., inheritance), and in the strictest sense, the most recent period of exposure for our bedrock sites equates to only the last few decades. Because of the well-constrained maximum possible exposure duration provided by geologic constraints and historical observations, here isotopic inheritance refers to exposure ages older than $10.20 \pm 0.23$ ka BP (i.e., pre-Holocene exposure).

## 5.1  Apparent in situ $^{10}$Be and $^{14}$C surface exposure ages

Apparent $^{10}$Be ages from abraded bedrock surfaces on the northeastern side of KNS, listed from just inboard of the historical maximum limit towards the modern ice margin, are $17.46 \pm 0.47$, $17.15 \pm 0.44$, $12.95 \pm 0.31$, $9.96 \pm 0.24$, $19.83 \pm 0.40$, $23.93 \pm 0.53$, $15.16 \pm 0.40$, and $6.83 \pm 0.16$ ka BP (Figs. 2, 9, 11, and 12; Table S3). On the southwestern side of KNS adjacent to the modern ice margin, $^{10}$Be ages from abraded bedrock surfaces are $6.94 \pm 0.16$, $6.75 \pm 0.16$, $6.56 \pm 0.27$, and $6.51 \pm 0.14$ ka BP, all roughly at equal distance from the present ice margin (Figs. 2 and 13; Table S3). Along our northeastern transect, six of the eight apparent $^{10}$Be ages exceed the maximum allowable Holocene exposure duration ($10.20 \pm 0.23$ ka). Apparent $^{10}$Be ages greater than this indicate the presence of inherited $^{10}$Be accumulated from a period of pre-Holocene exposure and insufficient subglacial erosion during the last glacial cycle to reset the cosmogenic clock. Of the two remaining $^{10}$Be ages not influenced by isotopic inheritance, the $^{10}$Be age of $9.96 \pm 0.24$ ka BP is statistically identical to the maximum allowable duration of Holocene exposure for the landscape located between the historical moraine and the modern ice margin. In addition, a $^{10}$Be age of $6.83 \pm 0.16$ ka directly adjacent to the modern margin is suggestive of less exposure and more burial (or more erosion; see Sect. 5.2) at this site, and it is also statistically identical to apparent $^{10}$Be ages from the southwestern side of KNS, indicating similar Holocene exposure histories.

Although the maximum amount of allowable Holocene exposure is well-constrained, we pair our $^{10}$Be measurements with in situ $^{14}$C measurements to (1) further assess the magnitude of isotopic inheritance in our bedrock samples and (2) constrain post-10 ka BP fluctuations of the GrIS margin. Whereas long-lived nuclides such as $^{10}$Be must be removed from the landscape via sufficient subglacial erosion, the relatively short half-life of $^{14}$C ($t_{1/2} = 5700$ years) allows previously accumulated in situ $^{14}$C to decay to undetectable levels after $\sim 30$ kyr of simple burial of a surface by ice; with the aid of subglacial erosion, in situ $^{14}$C can reach undetectable levels more quickly. In contrast to our apparent $^{10}$Be ages, which display varying degrees of Holocene exposure and isotopic inheritance, in situ $^{14}$C measurements

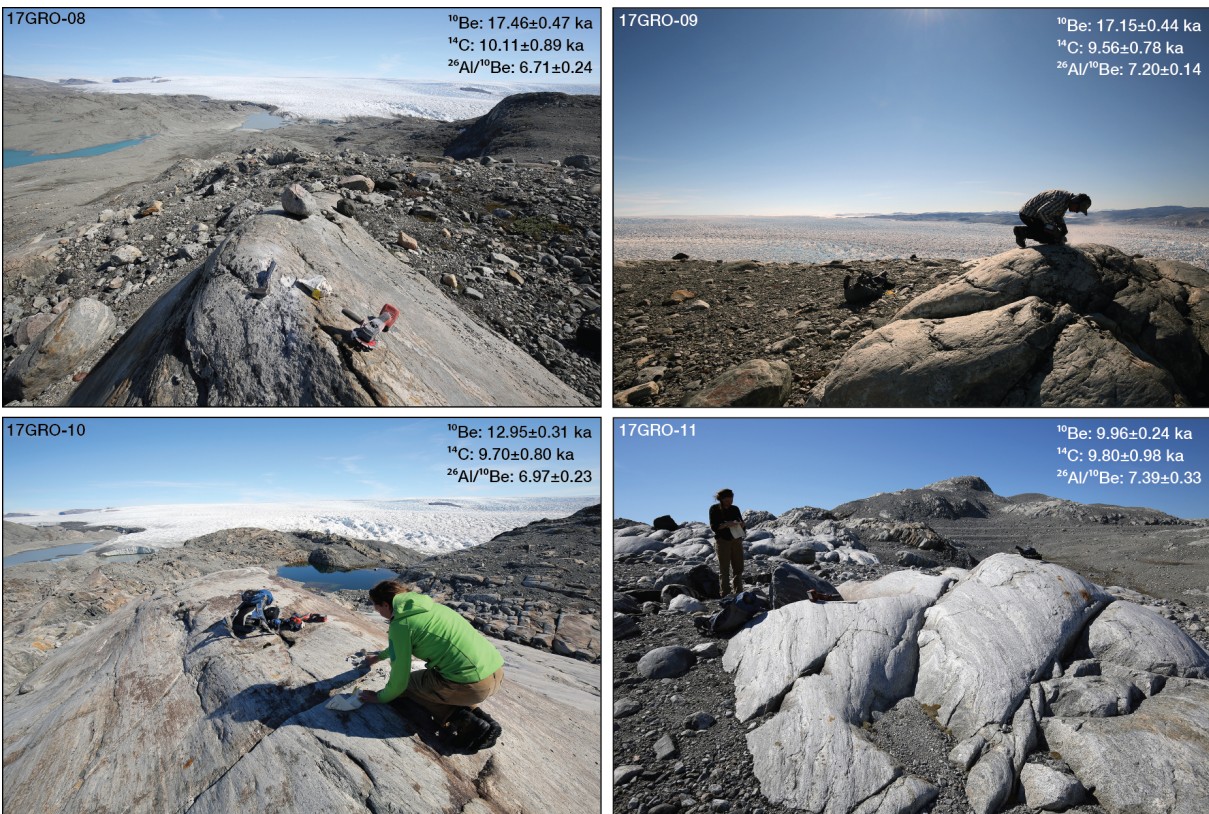

**Figure 11.** Sampled bedrock surfaces located between the historical maximum extent of the GrIS and the modern ice margin on the northeastern side of KNS.

are consistent with Holocene-only exposure histories. On the southwestern side of KNS, four apparent in situ $^{14}$C ages range from $6.59 \pm 0.47$ to $5.62 \pm 0.84$ ka BP, and paired $^{10}$Be–$^{14}$C measurements yield concordant exposure ages (Fig. 13). Within our northeastern bedrock transect, the two highest-elevation samples near the historical maximum limit with $^{10}$Be ages of $17.46 \pm 0.47$ and $17.15 \pm 0.44$ ka BP have significantly younger in situ $^{14}$C ages of $10.11 \pm 0.89$ ka and $9.46 \pm 0.78$ ka BP, respectively (Figs. 2 and 9). The next sample along this transect has a $^{10}$Be age of $12.95 \pm 0.31$ ka BP and an in situ $^{14}$C age of $9.70 \pm 0.80$ ka BP, followed by a sample with concordant $^{10}$Be and in situ $^{14}$C ages of $9.96 \pm 0.24$ ka BP and $9.80 \pm 0.98$ ka, respectively (Figs. 2 and 9). Within our cluster of samples closest to the ice margin along the northeastern transect, the one apparent $^{10}$Be age of $6.83 \pm 0.16$ ka BP is matched by a statistically identical in situ $^{14}$C age of $6.99 \pm 0.50$ ka BP, indicating that this $^{10}$Be age is likely not influenced by isotopic inheritance (Figs. 2 and 9). The remaining three samples in this cluster with $^{10}$Be ages of $23.93 \pm 0.53$, $19.83 \pm 0.40$, and $15.16 \pm 0.40$ ka BP have significantly younger in situ $^{14}$C ages of $6.68 \pm 0.48$, $6.29 \pm 0.73$, and $6.49 \pm 0.46$ ka BP, respectively (Figs. 2 and 9).

There are two modes of apparent in situ $^{14}$C exposure ages: a cluster of in situ $^{14}$C ages at $\sim 10$ ka BP and a sec-

ond cluster at $\sim 6$–$7$ ka BP (Figs. 2 and 14). Perhaps more importantly, however, is that these two modes correlate with two distinct morphostratigraphic surfaces. In situ $^{14}$C ages of $\sim 6$–$7$ ka BP are all from sites located directly adjacent to the modern ice margin. On the southwestern side of KNS, in situ $^{14}$C ages of $\sim 6$–$7$ ka BP are matched by concordant $^{10}$Be ages (Figs. 2 and 14). On the northeastern side of KNS, in situ $^{14}$C ages of $\sim 6$–$7$ ka BP are found closest to the modern ice margin. One of these sites has concordant $^{10}$Be and in situ $^{14}$C ages, while at the remaining sites, in situ $^{14}$C ages of $6$–$7$ ka BP are significantly younger than their paired $^{10}$Be ages, despite all sample sites appearing to have undergone significant subglacial erosion (Figs. 2 and 14). In situ $^{14}$C ages of $\sim 10$ ka BP only exist at our high-elevation sites directly adjacent to the historical maximum limit. Moreover, in situ $^{14}$C ages that range between $10.11 \pm 0.89$ and $9.46 \pm 0.78$ ka BP at our high-elevation sites are statically indistinguishable from the maximum duration of Holocene exposure these sites could have experienced ($\sim 10$ kyr; Figs. 2 and 14). Thus, the in situ $^{14}$C ages of $\sim 10$ ka BP, including the paired $^{10}$Be–$^{14}$C ages of $\sim 10$ ka BP, indicate that following deglaciation of the landscape just outboard of the historical moraine at $10.20 \pm 0.23$ ka BP, the GrIS margin continued to retreat inland and expose our high-elevation bedrock sites immediately thereafter. Moreover, subglacial erosion during

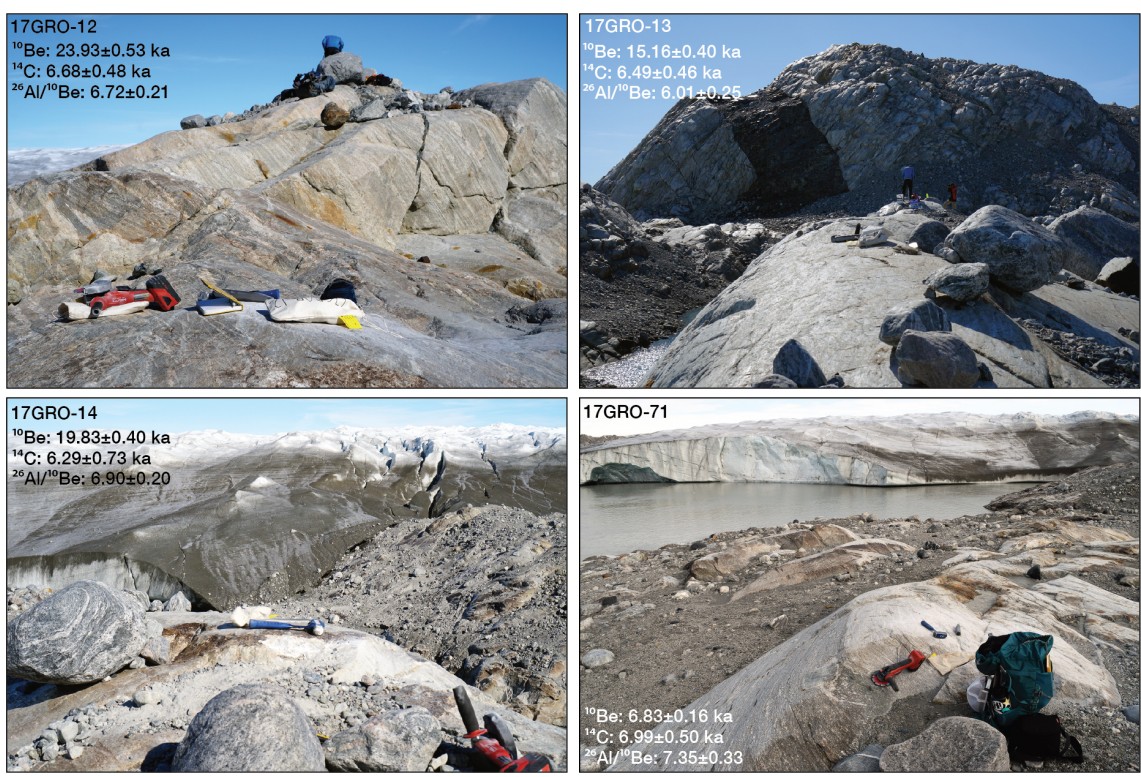

**Figure 12.** Sampled bedrock surfaces adjacent to the modern ice margin on the northeastern side of KNS.

the brief period of historical ice cover was negligible at these high-elevation sites because any significant subglacial erosion would result in apparent in situ $^{14}$C ages that are younger than the maximum Holocene exposure history these sites could have experienced. Our younger $^{10}$Be and in situ $^{14}$C ages of 6–7 ka BP, on the other hand, reflect some combination of less Holocene exposure and/or more subglacial erosion than the high-elevation sites.

## 5.2 $^{14}$C/$^{10}$Be ratios, transient burial, and subglacial erosion

Combining two or more nuclides with different half-lives quantifies the integrated amount of surface exposure and burial (e.g., Bierman et al., 1999; Goehring et al., 2010 TS5). Following a period of surface exposure, burial of a surface by overriding ice will cease nuclide production and lead to the faster decay of the short-lived nuclide relative to the nuclide with a longer (or more stable) half-life. Typically, the longer-lived nuclide is used to constrain the total amount of surface exposure, whereas the nuclide with a shorter half-life functions as the burial chronometer. Because the production ratio of two nuclides of a constantly exposed surface is known, a measured sample ratio below the constant production value represents the duration of surface burial. Over the Holocene timescale considered here, $^{10}$Be functions as an essentially stable nuclide because of its long

half-life ($t_{1/2} = 1.387$ Myr), whereas $^{14}$C, with its short half-life ($t_{1/2} = 5700$ years), acts as the burial chronometer (e.g., Goehring et al., 2010 TS6).

We only consider measured $^{14}$C/$^{10}$Be ratios in samples that do not have inherited $^{10}$Be; inherited $^{10}$Be results in physically unobtainable ratios. Using the average precision of our measured $^{14}$C/$^{10}$Be ratios (5.7 %), we estimate a minimum burial detection limit resolvable to 625 years at $1\sigma$ (Table S8 in the Supplement). All six samples without $^{10}$Be inheritance have $^{14}$C/$^{10}$Be ratios that are indistinguishable from constant exposure (Fig. 14; Table S8). One of these pairings with concordant $^{14}$C–$^{10}$Be ages of $\sim 10$ ka BP on the northeastern side of KNS indicates that the high-elevation bedrock sites likely became reoccupied by the GrIS only once as the ice sheet advanced towards the historical maximum extent $< 625$ years ago (Figs. 2 and 14). The remaining samples with $^{14}$C–$^{10}$Be ages of $\sim 6$–7 ka BP also reveal $< 625$ years of ice burial (Fig. 14). Our measured $^{14}$C/$^{10}$Be ratios are consistent with the observation of $\sim 245$–275 years of recent historical ice cover as being the only period of ice cover these sites experienced after early Holocene deglaciation. Our measured ratios, however, cannot rule out brief pre-18th century advances of the GrIS that may have covered our ice-marginal bedrock sites. Because our ice-marginal sites reside so close to the modern ice margin, it is likely that any brief advance of KNS prior to the 18th century would have covered our sampled bedrock locations.

Clim. Past, 17, 1–32, 2021

https://doi.org/10.5194/cp-17-1-2021

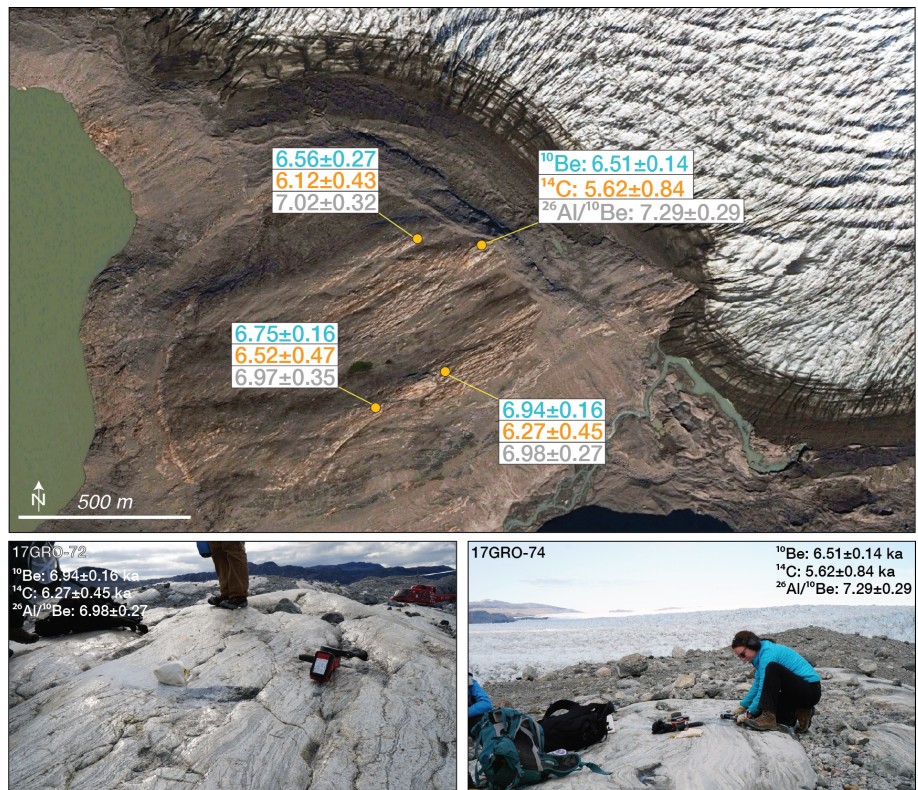

**Figure 13.** $^{10}$Be ages, in situ $^{14}$C ages (ka BP ± 1 SD), and measured $^{26}$Al/$^{10}$Be ratios at each bedrock sample site on the southwestern side of KNS. Also shown are representative bedrock sample sites 17GRO-72 and 17GRO-74. Map data: © Google Maps, Maxar Technologies.

Our measured $^{14}$C–$^{10}$Be ratios do not indicate any significant amounts of burial at our ice-marginal sites, but concordant $^{14}$C–$^{10}$Be ages of ∼ 6–7 ka BP suggest an exposure history and/or degree of subglacial erosion fundamentally different than the high-elevation 10 ka BP landscape (Figs. 2 and 14). The simplest interpretation is that the concordant $^{14}$C–$^{10}$Be ages and constant production ratios reflect one period of surface exposure over the last ∼ 6–7 kyr BP, prior to the period of historical ice cover. In this scenario, deglaciation of the high-elevation bedrock sites occurred at 10.20 ± 0.23 ka BP, but instead of continued inland retreat of the GrIS margin, the ice margin stabilized for several thousand years and then retreated inboard of today's margin around 6–7 ka BP. However, another possibility is that these sites also deglaciated at 10.20 ± 0.23 ka BP or later but later experienced significant subglacial erosion during the period of historical ice cover. Subglacial erosion through the production–depth profile in bedrock would result in younger apparent exposure ages (e.g., Goehring et al., 2010 [TS7]; Young et al., 2016) while maintaining a $^{14}$C/$^{10}$Be ratio consistent with constant exposure as long as erosion was limited to the first few tens of centimeters over which spallation dominates nuclide production.

To explore the possibility that concordant $^{10}$Be and $^{14}$C ages of ∼ 6–7 ka BP are a product of significant subglacial erosion, we cast apparent $^{10}$Be and $^{14}$C ages as a function of total erosional depth into bedrock during the period of historical ice cover assuming varying lengths of total surface exposure (Fig. 15; Table S9 in the Supplement). Because of the significantly larger uncertainties in the in situ $^{14}$C measurements, we only use the $^{14}$C-based erosion depths as a check on the $^{10}$Be-based erosion depths. The more precise $^{10}$Be measurements result in more precise estimates of erosional depth, and because we made paired $^{10}$Be–$^{14}$C measurements (versus single nuclide measurements in separate geological samples at different locations), our $^{10}$Be measurements provide more robust constraints of simulated erosional depth. As an estimate of the maximum amount of total erosion our sample sites could have experienced, we assume that our ice-marginal sites experienced 10 kyr of total exposure, which is constrained by the timing of deglaciation from just outboard of the historical moraines and the estimated duration of historical ice cover (Sect. 5). We then project the $^{10}$Be and $^{14}$C production–depth profiles in bedrock at each site and determine the depth below the theoretical 10 kyr surface to which our measured $^{10}$Be and $^{14}$C concentrations equate (Fig. 15; Table S9). Assuming a total exposure duration of 10 kyr, $^{10}$Be- and $^{14}$C-based total erosional depths are 24.8 ± 1.8 and 20.1 ± 4.1 cm, respectively, during the period of historical ice cover (Fig. 15; Table S9). Using the known dura-

https://doi.org/10.5194/cp-17-1-2021 Clim. Past, 17, 1–32, 2021

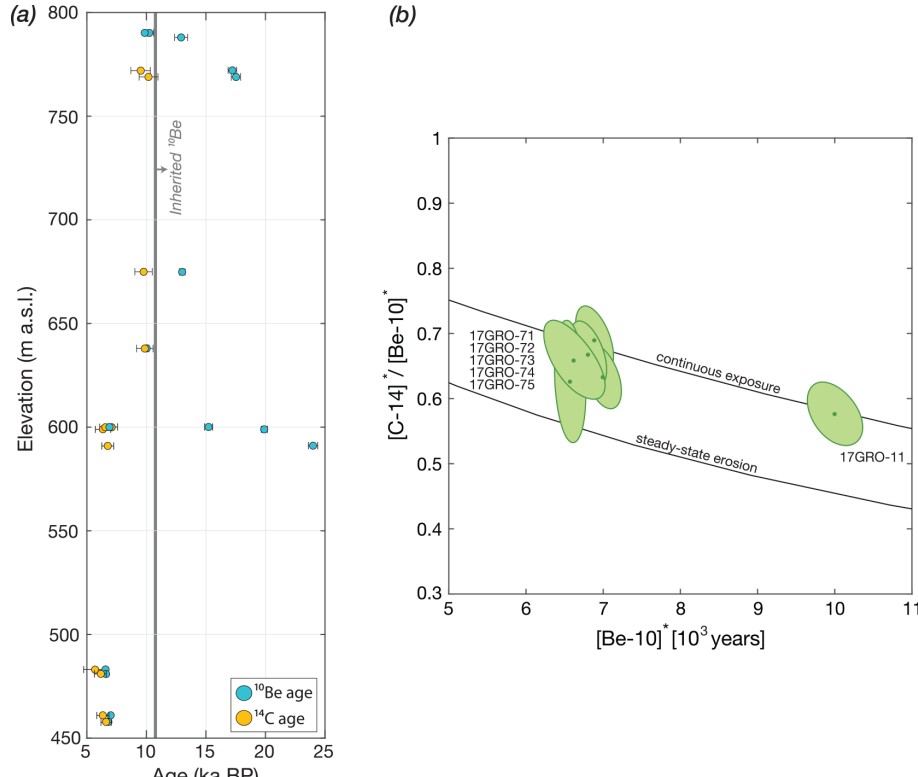

**Figure 14. (a)** $^{10}$Be–$^{14}$C apparent exposure ages from the northeastern and southwestern sides of KNS plotted against sample elevation. Apparent $^{10}$Be ages that are older than $\sim 10.3$ ka BP are influenced by isotopic inheritance, yet their corresponding in situ $^{14}$C ages are younger and consistent with Holocene-only exposure histories. In situ $^{14}$C ages vs. sample elevation are consistent with ice-margin thinning. **(b)** Paired $^{14}$C–$^{10}$Be diagram. The $x$ axis is the measured $^{10}$Be concentration normalized by the site-specific production rate (years); the $y$ axis is the measured $^{14}$C/$^{10}$Be ratio normalized to the production ratio. We use a regionally constrained $^{14}$C/$^{10}$Be spallation production ratio of 3.12 (Young et al., 2014). The simple exposure region (black lines) is defined by the continuous and steady-state erosion lines. Samples that are influenced by isotopic inheritance are excluded because they yield artificially low and meaningless ratios. Remaining samples are all consistent with constant exposure; we estimate a minimum burial detection limit of 625 years.

tion of historical ice cover (Sect. 5), these erosional depths translate to abrasion rates of $1.00 \pm 0.11$ mm yr$^{-1}$ TS8 ($^{10}$Be) and $0.81 \pm 0.19$ mm yr$^{-1}$ ($^{14}$C). Using total exposure durations of 9, 8, and 7.5 kyr yields $^{10}$Be-based erosion depths of $18.3 \pm 1.8$, $11.1 \pm 1.8$, and $7.2 \pm 1.8$ cm, respectively, which equate to abrasion rates of $0.74 \pm 0.10$, $0.45 \pm 0.08$, and $0.29 \pm 0.08$ mm yr$^{-1}$ (Fig. 15; Table S9).

Calculated abrasion rates are all well above the canonical polar subglacial abrasion rate of 0.01 mm yr$^{-1}$ (Hallet et al., 1996) but notably similar to abrasion rates inferred at the Jakobshavn Isbræ forefield constrained by similar methodology ($0.72 \pm 0.26$ mm yr$^{-1}$; Young et al., 2016); however, there are key differences between the two landscapes that aid in further limiting the plausible abrasion rates in the KNS forefield. Dozens of $^{10}$Be ages from the Jakobshavn Isbræ region, most notably from bedrock inside and outside the Jakobshavn Isbræ historical maximum limit, reveal a landscape entirely devoid of isotopic inheritance suggestive of a highly erosive environment (Young et a., 2013b, 2016). In contrast, new $^{10}$Be measurements presented here influ-

enced by isotopic inheritance, including from bedrock, coupled with previous $^{10}$Be measurements from the region (e.g., Larsen et al., 2014), at least qualitatively point to a less erosive GrIS in the KNS region. We think it is unlikely that subglacial abrasion rates at KNS can match or exceed those from the Jakobshavn Isbræ forefield, and we therefore favor an interpretation with less site exposure over significant amounts of subglacial abrasion. Moreover, the in situ $^{14}$C ages (and one $^{10}$Be age) from our high-elevation bedrock sites on the northeastern side of KNS are indistinguishable from the timing of early Holocene deglaciation and indicate that, at least at these high-elevation sites, subglacial abrasion during historical ice cover was negligible.

The most straightforward process to generate statistically identical $^{10}$Be and in situ $^{14}$C ages across all of the ice-marginal bedrock sites is for these sites to have undergone relatively little to no subglacial abrasion following the period of early to middle Holocene exposure. We also doubt that significant subglacial abrasion in the KNS forefield during the period of historical ice cover would result in such strik-

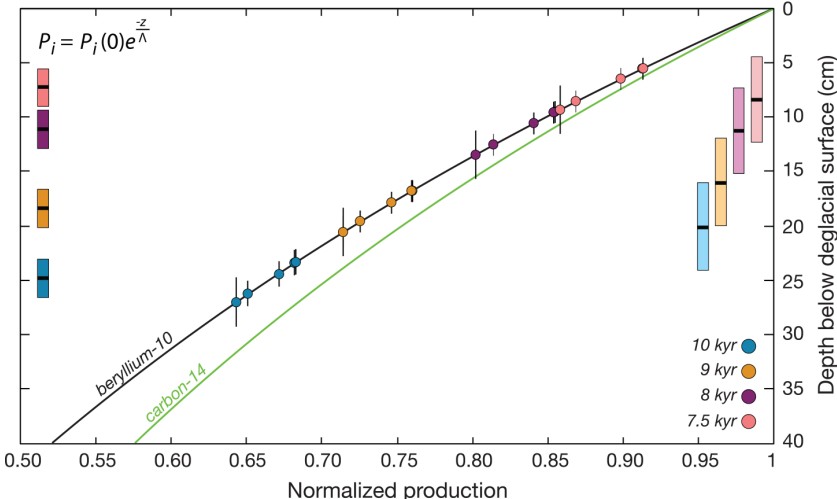

**Figure 15.** Modeled $^{10}$Be (black line) and in situ $^{14}$C (green line) production with depth in bedrock using a typical rock density of 2.65 g cm$^{-3}$ for gneiss. A normalized $^{10}$Be and in situ $^{14}$C concentration of 1 equals a prescribed exposure duration of 10, 9, 8, or 7.5 kyr. Symbols mark the depth below the prescribed deglacial surface along the production–depth profile that the GrIS would have to erode to during the period of historical ice cover that results in our measured nuclide concentrations. The $^{10}$Be production–depth profile is not fit to the data points and any $^{10}$Be measurement will fall somewhere along the depth profile. Bars on the left ($^{10}$Be) and right ($^{14}$C) sides of the diagram are the average erosional depths ($\pm$1 SD) of each exposure–erosion scenario (see Table S9). For figure clarity, we only show where our individual $^{10}$Be measurements fall along the production–depth profile; the average erosional depths based on the in situ $^{14}$C measurements are on the right side of the figure (Table S9). Production by spallation and muons are calculated independently (muons according to Balco, 2017; model 1A). $Z$ is the mass depth below the surface (g cm$^{-2}$) and is the product of depth (cm) and material density (g cm$^{-3}$). $P_i$ is the production rate of $^{10}$Be and in situ $^{14}$C (atoms g$^{-1}$ yr$^{-1}$) by spallation or muons at depth $z$. $P_i$ (0) is the surface production rate via spallation or muons. $\Lambda$ is the effective attenuation length (g cm$^{-2}$). Included here are only $^{10}$Be and in situ $^{14}$C concentrations for samples with concordant apparent exposure ages of $\sim$ 6–7 ka BP (17GRO-71, 17GRO-72, 17GRO-73, 17GRO-74, and 17GRO-75; Tables S6 and S9).

ingly uniform apparent exposure ages (i.e., uniform abrasion rates). On the other hand, it is unrealistic to assume that our ice-marginal sites did not experience some degree of abrasion because this is not a cold-based ice environment and striations were routinely observed at sampling locations; these features were likely formed during the period of historical ice cover. To estimate the likely timing of deglaciation at our ice-marginal sites that have concordant $^{10}$Be and in situ $^{14}$C ages of 6–7 ka BP, we present a range of deglaciation estimates. To constrain the latest possible age of deglaciation, we use all of the bedrock $^{10}$Be ages on the southwestern side of KNS ($n = 4$), combined with the apparent in situ $^{14}$C ages on the northeastern side of KNS that have paired $^{10}$Be measurements influenced by inheritance ($n = 3$) and a single $^{10}$Be age that is not influenced by isotopic inheritance. The mean age of these samples is $6.63 \pm 0.21$ ka BP, which does not include the last $\sim$ 275 years of historical ice cover when minimal to no (i.e., $^{14}$C) isotope production would have occurred, and assumes zero erosion during historical ice cover. Accounting for the period of historical ice cover, the timing of middle Holocene deglaciation from our ice-marginal bedrock sites is $6.91 \pm 0.21$ ka BP. As an upper bound on the timing of deglaciation, we rely on our modeled scenario of 7.5 kyr of exposure resulting in an abrasion rate of $0.29 \pm 0.08$ mm yr$^{-1}$.

As with our measured apparent $^{10}$Be and in situ $^{14}$C ages, the modeled 7.5 kyr scenario does not include the $\sim$ 275 years of historical ice cover. Including historical ice cover results in a deglaciation age of 7.78 ka BP. Combined, our favored interpretation is that the ice-marginal bedrock sites likely first became ice-free sometime between $\sim$ 7.8 and $\sim$ 6.9 ka BP and experienced abrasion during the period of historical ice cover at no more than $\sim$ 0.3 mm yr$^{-1}$. We suggest that this estimate sufficiently accounts for some degree of subglacial abrasion during the period of historical ice cover as suggested by field observations, while also acknowledging that tightly clustered apparent $^{10}$Be and in situ $^{14}$C ages are suggestive of bedrock surfaces that have undergone minimal modification following initial mid-Holocene exposure.

## 5.3   $^{26}$Al/$^{10}$Be ratios

As with $^{14}$C/$^{10}$Be ratios, $^{26}$Al/$^{10}$Be ratios can be used to measure integrated surface exposure and burial, albeit over much longer timescales. The $^{26}$Al–$^{10}$Be pairing utilizes preferential decay of $^{26}$Al ($t_{1/2} = 0.705$ Ma) relative to $^{10}$Be ($t_{1/2} = 1.387$ Ma) to quantify exposure and burial over glacial–interglacial timescales or longer (e.g., Bierman et al., 1999; Fabel et al., 2002; Gjermundsen et al., 2015).

The canonical $^{26}$Al/$^{10}$Be production ratio is considered to be 6.75 (Balco and Rovey, 2008; Balco et al., 2008), but measurements from western Greenland constrain the $^{26}$Al/$^{10}$Be production ratio to $7.3 \pm 0.3$, suggesting that the production ratio scales with latitude and elevation (Corbett et al., 2017). Modeling suggests that the $^{26}$Al/$^{10}$Be production ratio at sea level and high latitudes is $\sim 7.0$–$7.1$ (Argento et al., 2013).

Our $^{26}$Al/$^{10}$Be ratios range from $7.39 \pm 0.33$ to $6.01 \pm 0.25$ in the KNS forefield ($n = 12$; Table S5). Because of the extremely close proximity of all of our bedrock samples, these surfaces must have the same exposure and burial histories over glacial–interglacial timescales; the value of $6.01 \pm 0.25$ is anomalously low relative to our remaining measurements. After removing the lowest ratio, remaining ratios range from $7.39 \pm 0.33$ to $6.71 \pm 0.24$ with a mean of $7.05 \pm 0.24$ ($n = 11$; Table S5). Further limiting this dataset to samples that have no detectable inherited $^{10}$Be results in a mean value of $7.17 \pm 0.20$ ($n = 6$; Table S5). With the exception of our lowest measured ratio ($6.01 \pm 0.25$), each of our $^{26}$Al/$^{10}$Be ratios overlaps with the constant production values at $2\sigma$ uncertainty regardless of which constant production is used. Yet, our $^{26}$Al/$^{10}$Be ratios are systematically greater than 6.75 and suggest that the true production ratio is $> 6.75$, more consistent with recent modeled and empirical estimates (Table S5; Argento et al., 2013; Corbett et al., 2017).

The burial detection limit for the paired $^{26}$Al–$^{10}$Be method is typically on the order of approximately one glacial cycle, although this is dependent on the uncertainty in the measured $^{26}$Al/$^{10}$Be ratio. All of our measured $^{26}$Al/$^{10}$Be ratios suggest constant exposure, including samples with inherited $^{10}$Be (apparent $^{10}$Be ages $> 10$ ka BP), yet the known ice-margin history requires that constant exposure could have only occurred over the last $\sim 10$ kyr (Fig. 16; excluding the brief period of historical ice cover that is undetectable with the $^{26}$Al–$^{10}$Be chronometer). The most likely source of the excess $^{10}$Be in samples with apparent $^{10}$Be ages $> 10$ ka BP is surface exposure during Marine Isotope Stage 5e (MIS; $\sim 129$–$116$ ka BP; Stirling et al., 1998). Brief exposure of our bedrock surfaces during MIS 5e and surface burial between MIS 5d and $\sim 10$ ka BP, followed by re-exposure for the last 10 kyr, is the most straightforward scenario to have measured $^{26}$Al/$^{10}$Be ratios consistent with constant exposure while also containing a slight amount of inheritance. Moreover, this scenario is consistent with the broad outline of GrIS change over the last glacial cycle. Greenland ice-core data and offshore sediment records reveal that the GrIS was smaller than today during MIS 5e (Colville et al., 2011; NEEM, 2013), which suggests that bedrock currently emerging from beneath the GrIS was likely exposed for some period of time during MIS 5e.

At the same time, we cannot rule out the possibility that small amounts of inherited $^{10}$Be, coupled with $^{26}$Al/$^{10}$Be ratios consistent with constant exposure, are a result of exposure during MIS 3. Terrestrial evidence of a restricted GrIS during MIS 3 is limited so far to select sites in northern

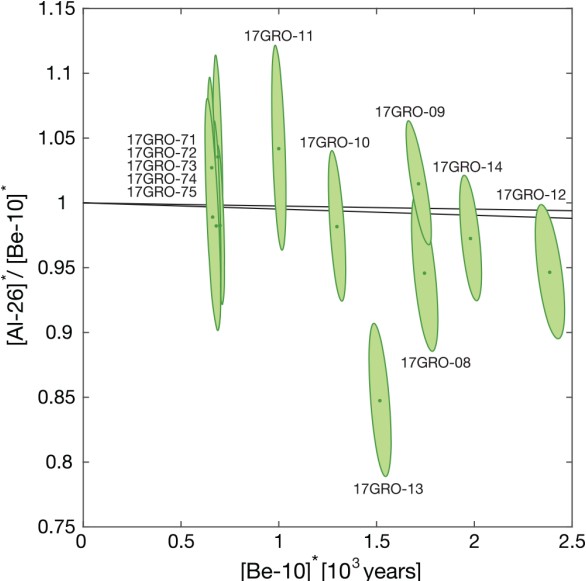

**Figure 16.** Paired $^{26}$Al–$^{10}$Be diagram with all measurements from the KNS forefield ($n = 12$). The $x$ axis is the measured $^{10}$Be concentration normalized by the site-specific production rate (years); the $y$ axis is the measured $^{26}$Al/$^{10}$Be ratio normalized to the production ratio. We use a production ratio of 7.07 (Argento et al., 2013). The black lines define the simple exposure region. All of our measured ratios, with the exception of 17GRO-13, indicate constant exposure of the sample site (values listed in Table S5). Because of the close proximity of these sites, they must have the same glacial–interglacial exposure history, and thus we consider 17GRO-13 a likely outlier.

Greenland (e.g., Larsen et al., 2018), and marine-based sediment records from off southwestern Greenland point to significant MIS 3 recession of the GrIS; however, the MIS 3 ice margin may have still been located off the modern coastline, and thus our sample sites would remain ice-covered (Seidenkrantz et al., 2019). In addition, any significant exposure of our KNS bedrock sites during MIS 3 would require significant amounts of burial (or erosion) in order to have any accumulated in situ $^{14}$C decay away to undetectable levels, as there is no evidence of inherited in situ $^{14}$C in our samples. Moreover, eustatic sea level curves indicate that sea level was at least 30–40 m below present during MIS 3 (Siddall et al., 2003; Grant et al., 2014). Most of this eustatic sea level signature is driven by changes in the Laurentide Ice Sheet, but it is difficult to imagine a complete decoupling of Laurentide and Greenland ice-sheet behavior whereby the Laurentide remains relatively large, while the GrIS is smaller than today during MIS 3. It is certainly possible that our sites were exposed during MIS 3, but we favor the more straightforward explanation of MIS 5e exposure at our sample site, considering the following: (1) ice-core records reveal that the region was likely warmer during MIS 5e vs. MIS 3 (NGRIP, 2004 TS9; NEEM, 2013), (2) balancing the MIS 5e

eustatic sea level budget likely requires a significant contribution from Greenland (Dutton et al., 2015), and (3) there is a lack of any additional terrestrial evidence in southwestern Greenland for an MIS 3 ice-sheet configuration similar to or more restricted than today. While it is perhaps unsurprising that our $^{26}$Al–$^{10}$Be measurements in the KNS forefield suggest surface exposure during a previous interglacial, these measurements nonetheless suggest the GrIS margin in the KNS region was at or behind the present margin during MIS 5e.

## 6 Holocene evolution of the southwestern Greenland Ice Sheet

### 6.1 Early Holocene moraine deposition

Direct $^{10}$Be ages from moraine boulders constrain Kapisigdlit stade moraine deposition to $10.24 \pm 0.31$ ka BP, which is supported by statistically identical bracketing $^{10}$Be ages from erratic boulders located outboard and inboard of the Kapisigdlit stade moraine (Figs. 2 and 7). North of KNS at Qamanaarsuup Sermia, $^{10}$Be ages from moraine boulders indicate that deposition of the so-called Kapisigdlit moraine occurred at $9.57 \pm 0.33$ ka BP, which is supported by bracketing $^{10}$Be ages of $10.29 \pm 0.14$ and $9.29 \pm 0.07$ ka BP (Fig. 3). These two moraines share identical maximum-limiting ages, but the direct moraine and minimum-limiting $^{10}$Be ages between the two sites are statistically distinguishable. Maximum-limiting $^{10}$Be ages from erratic boulders located immediately outboard of a moraine can provide a close constraint on the age of moraine deposition if moraine deposition occurs via a brief stillstand of the ice margin (e.g., Young et al., 2013b). If moraine deposition occurs after a readvance of the ice margin, however, maximum-limiting $^{10}$Be ages need not be close limiting ages. In contrast, minimum-limiting $^{10}$Be ages on erratic boulders from immediately inside a moraine should provide close minimum-limiting constraints regardless of whether moraine deposition occurred through a stillstand of the ice margin during retreat or after a readvance. The distribution of $^{10}$Be ages presented here suggests that deposition of the Kapisigdlit stade moraine in the immediate KNS region occurred via a stillstand of the ice margin or a brief readvance that occurred within the resolution of our chronometer. At Qamanaarsuup Sermia, however, there is a gap of several hundred years between maximum-limiting $^{10}$Be ages and moraine-based $^{10}$Be ages, and a total of $\sim 1$ kyr between maximum- and minimum-limiting $^{10}$Be ages suggests that the Qamanaarsuup Sermia moraine suite was deposited after a readvance of the GrIS.

Another possibility is that the Qamanaarsuup Sermia moraine complex is an amalgamation of moraines relating to chronologically distinct advances or stillstands of the ice margin. The numerous and tightly packed moraines here, vs. the more well-defined Kapisigdlit stade moraine at KNS, are

suggestive of a stagnating or oscillating ice margin. Moreover, our moraine boulder dataset contains more scatter than we typically observe in southwest Greenland (e.g., Young et al., 2020a), suggesting it is possible that we sampled moraine boulders from two or more distinct advances. In this case, combining all of our $^{10}$Be ages from moraine boulders at Qamanaarsuup Sermia would inadvertently mask the timing of two or more advances; for example, if advances occurred at ca. 10.4–10.3 and 9.3–9.0 ka BP, combining all $^{10}$Be ages might result in an average $^{10}$Be age of $\sim 9.7$ ka BP, especially if moraine boulders from each advance are reworked by the ice margin. In several instances in southwest Greenland, $^{10}$Be ages from erratics just inboard of a moraine are statistically identical to the moraine boulders themselves (e.g., Young et al., 2013b, 2020a), perhaps indicating that our minimum-limiting age of $9.29 \pm 0.07$ ka BP constrains an advance of the GrIS in this sector to $\sim 9.3$ ka BP (moraine closest to erratics), and moraines located farther away from the ice margin might relate to an advance of the GrIS closer in age to our maximum-limiting $^{10}$Be ages ($10.29 \pm 0.14$ ka BP; Fig. 3). Our $^{10}$Be ages from moraine boulders at Qamanaarsuup Sermia, however, show no trend across moraines or with distance from the ice margin. We prefer the more conservative interpretation acknowledging that if two or more distinct advances occurred, we cannot resolve these advances with our dataset. We can confidently say that all moraines were deposited between $10.29 \pm 0.14$ and $9.29 \pm 0.07$ ka BP, and a moraine age of $9.57 \pm 0.33$ ka BP is consistent with these bracketing ages.

Including the uncertainty in the $^{10}$Be production rate, the Kapisigdlit stade moraine in the KNS forefield was deposited at $10.24 \pm 0.36$ ka BP, consistent with a maximum-limiting radiocarbon age of $10\,170 \pm 340$ cal yr BP (Weidick et al., 2012; Larsen et al., 2014), and moraine deposition at Qamanaarsuup Sermia occurred at $9.57 \pm 0.38$ ka BP. Note that these moraine ages overlap at $1\sigma$ only when including the $^{10}$Be production rate uncertainty, which results in systematic shifts in age and is only needed when comparing these moraine ages to independent chronometers; these moraines are distinguishable at $1\sigma$ in $^{10}$Be space. Moraine deposition in the KNS forefield at $10.24 \pm 0.36$ ka BP is, within resolution, synchronous with widespread moraine deposition in southwestern Greenland and Baffin Island at ca. 10.4–10.3 ka BP. Indeed, $^{10}$Be ages across several locations in Baffin Bay reveal that sectors of the GrIS, Laurentide Ice Sheet, and independent alpine glaciers on Greenland and Baffin Island all deposited moraines at ca. 10.4–10.3 ka BP, likely in response to freshwater-induced regional cooling (Young et al., 2020a). Contemporaneous moraine deposition indicates that the KNS sector of the GrIS also likely responded to regional cooling at ca. 10.4–10.3 ka. Moraine deposition at $9.57 \pm 0.38$ ka BP at Qamanaarsuup Sermia, however, does not fit a well-established regional pattern of moraine deposition. Similar to moraine deposition at ca. 10.4–10.3 ka BP, widespread moraine deposition across Baf-

fin Bay occurred at ca. 9.3–9.0 ka BP and is thought to be driven by the 9.3 ka cooling event displayed in Greenland ice cores (Young et al., 2020a). The Qamanaarsuup Sermia moraine age (9.57 ± 0.38 ka BP) is, within uncertainties, synchronous with the 9.3 ka cooling event, but $^{10}$Be-dated moraines in Baffin Bay consistently date at or slightly after the 9.3 ka cooling event (Young et al., 2011b, 2020a; Crump et al., 2020). North of KNS, $^{10}$Be ages constrain deposition of a moraine to 9.7 ± 0.7 ka, consistent with moraine deposition at Qamanaarsuup Sermia despite somewhat larger uncertainties (Lesnek and Briner, 2018). In addition to widespread moraine deposition in Baffin Bay at ca. 10.4–10.3 and 9.3–9.0 ka BP, these emerging $^{10}$Be ages tentatively suggest an additional mode of moraine deposition at ca. 9.7 ka, perhaps in a response to freshwater-related cooling (Lesnek and Briner, 2018). Regardless of our ability to correlate the Qamanaarsuup Sermia moraine dated to 9.57 ± 0.38 ka BP with moraines beyond the KNS region, our $^{10}$Be ages reveal that early Holocene moraines across the broader KNS region are not equivalent features. Our results reveal at least two periods of moraine deposition occurred at 10.24 ± 0.36 and 9.57 ± 0.38 ka BP. Lastly, we note that there is no moraine associated with the 8.2 ka abrupt cooling event in the KNS and Qamanaarsuup Sermia regions. Whereas widespread moraine deposition in Baffin Bay occurred in response to the 8.2 ka event (Young et al., 2020a), the GrIS retreated inboard of the eventual historical ice limit prior to 8.2 ka at KNS and Qamanaarsuup Sermia. Any moraine related to the 8.2 ka event that may have existed on the landscape was overrun and destroyed by the historical advance of the GrIS.

## 6.2 Retreat of the GrIS behind the modern margin during the early Holocene

We next consider the timing and significance of when the GrIS margin crossed the historical maximum–modern ice-margin threshold during early Holocene deglaciation. At KNS, $^{10}$Be ages from just outboard of the historical maximum limit indicate that following deposition of the Kapisigdlit stade moraine at 10.24 ± 0.36 ka BP, the GrIS crossed the historical maximum limit soon thereafter at 10.20 ± 0.23 ka (Fig. 2). At Qamanaarsuup Sermia, $^{10}$Be ages from just outboard of the historical maximum reveal that this portion of the GrIS margin crossed the historical maximum limit at 9.29 ± 0.18 ka BP, ∼ 1 kyr later than at KNS (Fig. 3). Near Narsap Sermia (Fig. 1), $^{10}$Be ages from outboard of the historical maximum limit indicate that the area deglaciated at ∼ 8.8 ka BP (Fig. 1), and south of KNS, $^{10}$Be ages suggest that the landscape just outboard of the historical maximum deglaciated between ∼ 10.0 and 9.5 ka BP. Farther south, a single $^{10}$Be age suggests that the GrIS margin did not retreat behind its modern margin until ∼ 7.7 ka BP (Fig. 1). Lastly, at Saqqap Sermia, basal radiocarbon ages suggest that deglaciation of the landscape immediately outboard of the modern margin occurred as late as ∼ 5.4–4.6 ka BP, at least

5 kyr after deglaciation of the landscape outboard of the modern ice margin in the KNS region (Fig. 1; Levy et al., 2017).

At face value, there is > 5 kyr of spread in the timing of deglaciation immediately outboard of the historical maximum limit in the broader KNS region, perhaps suggestive of substantial differences in ice-margin behavior in the early to middle Holocene (Fig. 1). When considering all of the available ice-margin constraints, deglaciation occurred earliest in the immediate KNS region, with deglaciation occurring later in sectors beyond KNS. The timing of deglaciation of the landscape immediately outboard of the historical maximum, however, is dictated by expected differences in the rate of ice-margin retreat and differences in the magnitude of the late Holocene readvance of the ice margin. Older $^{10}$Be ages from outboard of the historical limit at KNS relative to adjacent ice margins could simply reflect the earlier deglaciation of KNS compared to neighboring ice margins. In this scenario, marine-based dynamic processes would likely drive early and rapid deglaciation of KNS, whereas deglaciation would lag behind in adjacent land-based ice margins where marine-based dynamical processes exert less control on ice-margin behavior. Alternatively, the pattern of early deglaciation at KNS with later deglaciation in adjacent margins can be entirely explained by the magnitude of the late Holocene readvance of the ice margin. For example, if the late Holocene readvance of KNS was of greater magnitude than that of adjacent margins, then the KNS terminus would overrun and rest upon a landscape that deglaciated earlier, and thus $^{10}$Be ages from outboard of the historical moraine would be older. The greater the magnitude of ice-margin readvance, the older the $^{10}$Be ages just outboard of the historical moraine will be.

Additional chronological constraints that track the history of the GrIS margin prior to and immediately after the ice margin retreated behind the eventual historical maximum limit place the apparent asynchrony of deglaciation across the KNS region within a broader context. At Qamanaarsuup Sermia, $^{10}$Be ages just outboard of the historical moraine are 9.29 ± 0.18 ka; however, radiocarbon-dated lake sediments suggest that ice remained near the historical maximum extent until ∼ 8.3 ka BP (Figs. 3 and 6). Including $^{10}$Be ages from just beyond the early Holocene moraines, all available ice-margin constraints at Qamanaarsuup Sermia indicate that the position of the GrIS margin in this region underwent minimal changes between ∼ 10.3 and ∼ 8.3 ka BP, and at least ∼ 2 kyr of ice-margin history is represented in a relatively restricted lateral zone on the landscape. Had the late Holocene readvance of the Qamanaarsuup Sermia margin been slightly more extensive, the ice margin would have overrun the early Holocene moraines currently residing immediately outboard of the historical maximum. The resulting historical maximum limit would abut a landscape that deglaciated just prior to 10 ka BP, similar to the relationship between the historical maximum ice limit and $^{10}$Be ages outboard of this limit observed at KNS. Or, had the late Holocene readvance been slightly less extensive, the historical moraine would abut a

landscape that likely deglaciated at $\sim 8.3$ ka BP, similar to the timing of deglaciation at other locations in a broader KNS region (Fig. 1). In a similar manner, $^{10}$Be ages at KNS indicate that the ice margin retreated behind the historical maximum limit at $10.20 \pm 0.23$ ka, but ice did not continue to retreat inland and instead remained near the current margin until $\sim 7.8$–$6.9$ ka BP based on concordant $^{10}$Be and in situ $^{14}$C ages from recently exposed bedrock (Sect. 5.2). Near Narsap Sermia (Fig. 1), $^{10}$Be ages from outboard of the historical maximum limit, paired with a basal radiocarbon age from Caribou Lake, indicate that the area deglaciated at $\sim 8.8$ ka BP, but ice remained in the lake catchment and likely near the modern margin until $\sim 7.5$ ka BP (Fig. 1). Additional $^{10}$Be ages south of KNS near Deception Lake suggest that the landscape just outboard of the historical maximum deglaciated between $\sim 10.0$ and $9.5$ ka BP, but the ice margin was likely near its present limit until $\sim 9.2$ ka BP based on basal radiocarbon ages.

Additional locations with paired $^{10}$Be ages and proglacial threshold lake records along much of the western GrIS margin reveal that after the GrIS margin retreated behind the position of the historical maximum extent, the ice margin remained near this position for several hundred to thousands of years before retreating farther inland (Fig. 17). Notably, paired $^{10}$Be ages and radiocarbon constraints north of Jakobshavn Isbræ (Sermeq Kujalleq) suggest that the ice margin was near its current position between $\sim 10$ and $5.2$ ka BP (Newspaper Lake; Cronauer et al., 2015; Figs. 1 and 17). Closer to Jakobshavn Isbræ, paired $^{10}$Be ages and radiocarbon constraints suggest the ice margin was in a configuration similar to today between 7.8 and 5.5 ka BP (Lake Lo; Håkansson et al., 2014; Figs. 1 and 17). In the Kangerlussuaq region, radiocarbon ages from proglacial threshold lakes reveal that the ice margin remained in a configuration similar to today following initial deglaciation (Figs. 1 and 17; Young and Briner, 2015; Lesnek et al., 2020). Unique to the Kangerlussuaq region is proglacial lake Tasersuaq, where radiocarbon-dated sediments, combined with maps of sub-ice topography, suggest that the GrIS margin never retreated out of the Tasersuaq catchment, which extends only $\sim 1.9$ km behind the modern margin, during the Holocene (Lesnek et al., 2020). Additional proglacial threshold lake records from separate drainage basins in the Kangerlussuaq region suggest that the ice margin retreated $\sim 3.7$ and $\sim 26$ km inland during the Holocene (Lesnek et al., 2020) and broadly support minimal ice-margin recession during the middle Holocene.

The combination of new $^{10}$Be ages, records from proglacial threshold lakes, and paired $^{14}$C–$^{10}$Be measurements from KNS and Qamanaarsuup Sermia defines a window between $\sim 10$–$7$ ka BP when the GrIS margin was likely near its present position. After 7 ka BP, the GrIS margin retreated inland before re-approaching its current configuration sometime in the last millennium. Considering the ice-margin constraints from nearby Saqqap Sermia to the north

(Levy et al., 2017), the GrIS in the KNS region likely remained near its current position as late as $\sim 5$ ka. We suggest that the large range in ages constraining the timing of deglaciation outboard of the historical moraine within the KNS region and along the broader western GrIS margin, coupled with ice-margin constraints from proglacial threshold lakes and cosmogenic isotope measurements from recently exposed bedrock, broadly defines a window between $\sim 10$–$5$ ka BP when the GrIS margin was near its current margin (Fig. 17). The southwestern GrIS margin reached its late Holocene maximum extent during historical times, but records from proglacial threshold lakes indicate that the ice margin had advanced back to near the modern margin by $\sim 2$ ka BP (Fig. 17). Within a relatively narrow $\sim 3$ kyr window between $\sim 5$–$2$ ka BP, the southwestern GrIS margin retreated inland, achieved its minimum extent, and readvanced towards the historical maximum extent, which likely precludes significant inland retreat of the southwestern GrIS margin during the Holocene.

## 6.3 Geologic data–model comparison of ice-margin change in southwest Greenland

Geologic reconstructions of ice-sheet change offer an ideal target for ice-sheet modeling efforts aimed at reconstructing the geometry and ice volume of past ice sheets through model tuning (Simpson et al., 2009; Lecavalier et al., 2014). These reconstructions, however, also serve as an ideal test bed for modeling efforts aimed at evaluating the sensitivity of past ice-margin migration to climatic and oceanic influences (Briner et al., 2020). Paleo-ice-sheet modeling efforts typically rely on coarse-resolution meshes and simplification of ice-flow approximations to achieve computational efficiency. While this approach has enabled more simple models to assess the sensitivity of ice-margin response to model parameters, crude climate forcings are typically used, making it difficult to properly assess ice-margin sensitivity to climate. Here, we further explore the deglaciation history of the KNS region by comparing the geologic record of ice-sheet change to recently completed model simulations of southwestern GrIS evolution through the Holocene (Cuzzone et al., 2019; Briner et al., 2020). These Holocene ice model simulations use the highest horizontal mesh resolution across our field area to date and use a range of state-of-the-science gridded climate reconstructions as the input climate (Badgeley et al., 2020; Briner et al., 2020), thus presenting an opportunity for new insights regarding glacier history and ice-sheet modeling. Our goals in exploring the model simulations are to (a) assess the magnitude of recession inboard of the present margin, (b) compare rates of retreat and timing of ice-margin change in both the model and in the observations, and (c) explore avenues for model improvement in a known problem area for ice-sheet modeling.

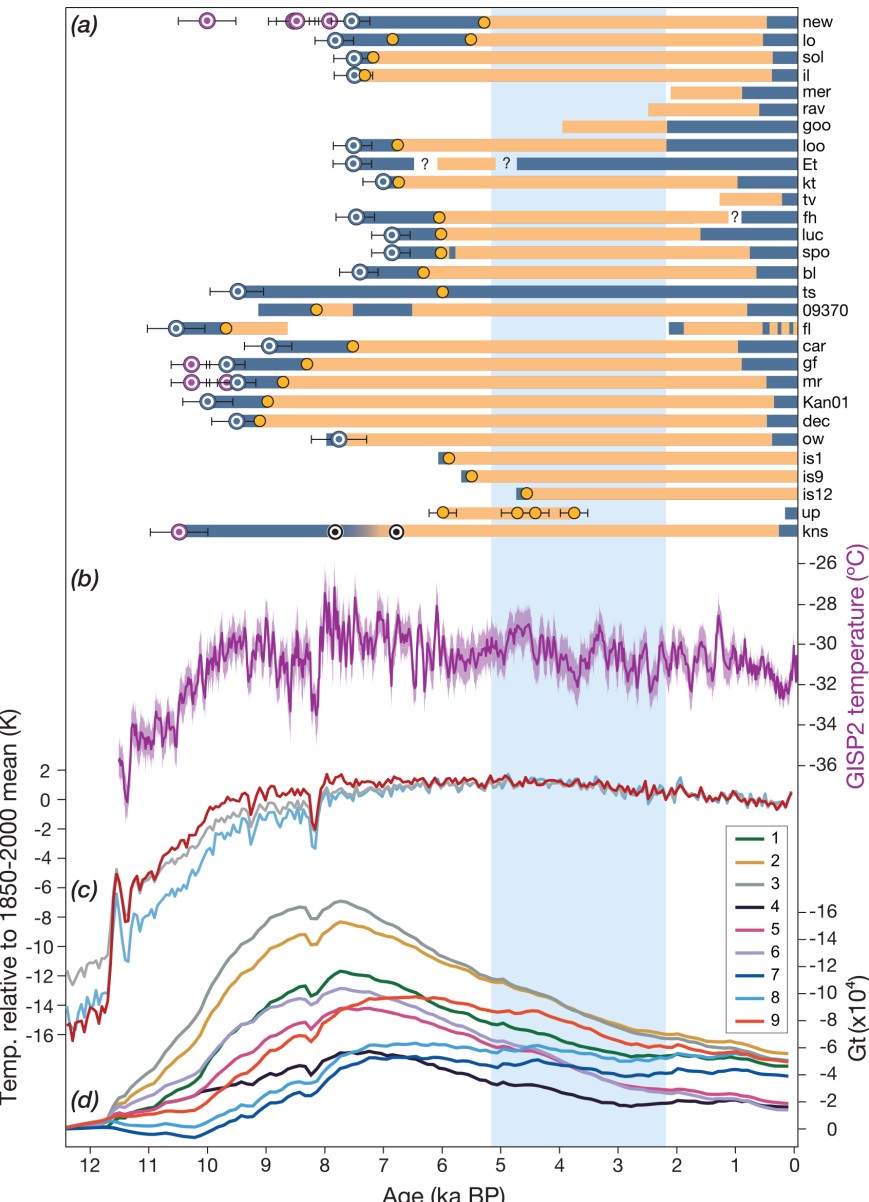

**Figure 17. (a)** Cosmogenic nuclide and $^{14}$C-dated proglacial threshold lake records constraining the behavior of the southwestern GrIS ice margin. Site abbreviations are from Fig. 1. Using sedimentological boundaries in proglacial threshold lakes (organic sediments vs. silt), the orange bars mark periods of time at each site when the GrIS was less extensive than today (organic sediments), and blue bars marks periods of time when the ice margin was in a position similar to today (silt). Orange dots are traditional radiocarbon ages from the contact between basal silt and the overlying organic sediments that constrain when the southwestern GrIS margin retreated out of each lake's catchment in the early to middle Holocene. Blue bullseyes are $^{10}$Be ages from immediately outboard of the historical maximum extent in each proglacial threshold lake drainage catchment, and purple bullseyes are additional $^{10}$Be ages from within ∼ 1 km of the historical maximum extent. Black bullseyes define our estimated timing of inland retreat of KNS based on in situ $^{14}$C measurements (Sect. 5.2) Note that the ice-margin constraints from Ujarassuit Paavat (up) are traditional $^{14}$C ages from marine bivalves reworked into the historical limit, which mark times when the ice margin was behind the historical maximum extent (Fig. 1). Combined, $^{10}$Be from immediately outboard of the historical maximum limit, $^{10}$Be ages from within ∼ 1 km of the historical maximum limit, and basal $^{14}$C ages from threshold lakes define a window when the ice margin was in a position similar to today. Considering all constraints, the southwestern GrIS margin likely achieved its minimum extent after ∼ 5 ka and was approaching its modern position as early as ∼ 2 ka BP (blue shading). **(b)** Mean annual temperatures at the Greenland Ice Sheet Project 2 (GISP2) site reconstructed using gas-phase $\delta$Ar-N$_2$ measurements (±2$\sigma$; Kobashi et al., 2017). **(c)** Temperature anomalies over southwestern Greenland used in recent ISSM modeling runs (Badgeley et al., 2020; Briner et al., 2020). **(d)** Corresponding modeled ice mass for the southwestern GrIS for nine simulations using different climate reconstructions (Briner et al., 2020); note the reversed $y$ axis.

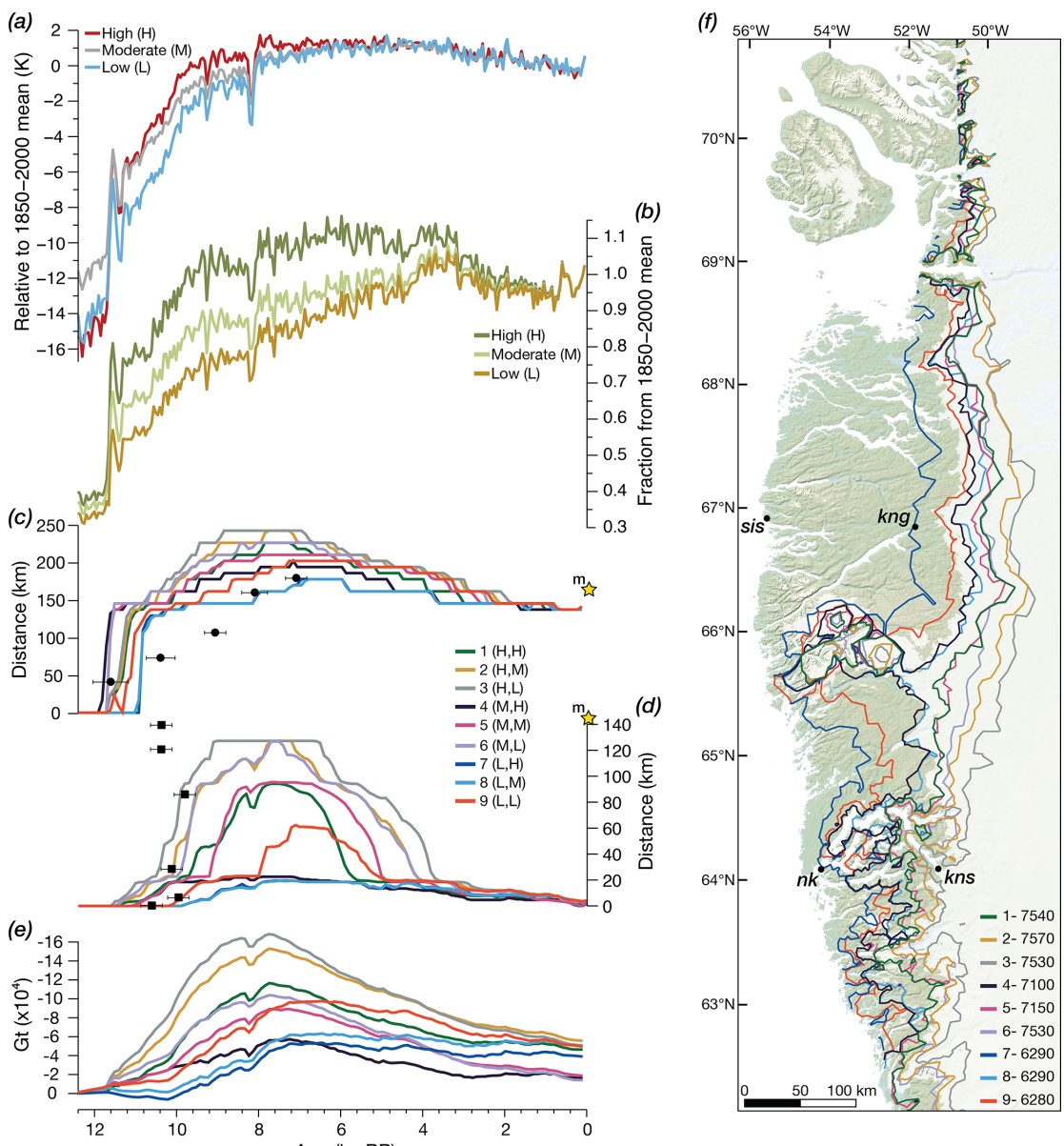

**Figure 18. (a)** Area-averaged mean annual temperature using three different reconstructions across the ISSM model domain (Badgeley et al., 2020; Briner et al., 2020). **(b)** Same as **(a)**, but for mean annual precipitation (Badgeley et al., 2020; Briner et al., 2020). **(c)** Simulated margin position in the Sisimiut (sis)–Kangerlussuaq (kng) region through the Holocene compared to independent observations of ice-margin position based on dated moraines (black dots; 1$\sigma$ age uncertainty; Young et al., 2020a) and the modern ice margin ("m"– orange star; modified from Briner et al., 2020). The $y$ axis is the distance measured from the coast. Letters next to each simulation mark the temperature and precipitation reconstructions from panels **(a)** and **(b)**. **(d)** Simulated margin position in the Nuuk (nk)–Kangiata Nunaata Sermia (kns) region through the Holocene compared to independent observations of ice-margin position based on deglaciation ages (see Fig. 1 for age constraints and up-fjord transect). **(e)** Simulated ice mass through the Holocene across the model domain (Briner et al., 2020); note that the $y$ axis is reversed. **(f)** Simulated lateral ice-margin extent at the time of minimum ice mass in each model run. The year (BP) of the ice-mass minimum in each model run is listed in the legend.

Across the broader southwestern Greenland region, simulations are able to reproduce the observed pattern of ice-margin migration through the Holocene – early to middle Holocene retreat, inland recession, and late Holocene readvance (Fig. 18). In particular, north of the KNS region be-

tween Sisimiut and Kangerlussuaq, the simulated pattern and timing of ice-margin migration generally reproduce the geologic record of ice-margin change (Briner et al., 2020). In the KNS region, these same simulations generally depict an ice-margin that retreats eastwards, achieves a minimum ex-

tent, and then readvances in the late Holocene, thus sharing the broad outline of GrIS change provided by the available geological observations (Fig. 18). However, these simulations consistently depict ice that is too extensive compared to the observed record of ice-margin change in the KNS region (Fig. 18). Similar to the modeled pattern of ice-margin change, simulated ice masses in southwestern Greenland decrease through the early Holocene and achieve a minimum value between $\sim 7.6$ and 6.3 ka BP (Fig. 18).

The better model–data fit in the Kangerlussuaq region compared to the KNS region is likely the result of distinctly different ice-margin environments. The Kangerlussuaq region hosts what is primarily a land-terminating sector of the GrIS where ice-sheet behavior is dominated by surface mass balance (Cuzzone et al., 2019; Downs et al., 2020). Thus, ice-sheet behavior in the Kangerlussuaq region is almost entirely dictated by the climate forcings used in the model runs (Briner et al., 2020). The KNS fjord system, however, hosted a marine-terminating sector of the GrIS influenced by dynamical processes, but the model setup does not include calving or submarine melting of floating ice. Therefore, there is no dynamic mechanism with which to rapidly remove ice from the KNS fjord system in the model. Because the ice-sheet model is built on a high-resolution mesh, down to 2 km in the KNS region, it is able to resolve bed features such as the existing KNS topography and fjord bathymetry, allowing for better representation of mass transport and stress balance than in lower-resolution models. By resolving these features, however, ice is funneled efficiently to the ice margin, allowing the ice to persist in the KNS fjord system in the absence of calving, which is likely responsible for the simulated ice-margin being too extensive compared to the geologic record. We also note that there are portions of the model domain that are below sea level and susceptible to marine influence (Fig. S1 in the Supplement). Throughout our simulations, relative sea level varies through time, which could change the portions of our model domain that are marine- vs. land-terminating. Although marine processes (e.g., submarine melting of floating ice and calving) are not included in our simulations, we do include grounding-line migration, and our model also simulates floating ice at outlet glacier termini through the Holocene. It is difficult to determine how our simulations would be impacted by including marine processes without performing additional experiments, but areas at the ice front and immediately upstream could be particularly affected in warmer climates coincident with the Holocene minimum extent, as fast-flowing ice tends to maintain contact with the ocean.

Evaluating the fit between our ice-sheet model results and the geochronological data in the KNS region must be treated cautiously as climatic and dynamic processes may not be the only influences on model–data mismatches. For example, the simulated GrIS retreat through the KNS region is sensitive to how well bed topography is resolved (Cuzzone et al., 2019). In simulations using a lower-resolution mesh

(i.e., 10 km or greater), fjord bathymetry in the KNS region is not well-resolved, and as the ice surface lowers in response to Holocene warming, it intersects bedrock bumps (i.e., pinning points) that would otherwise be resolved as fjords using a higher-resolution mesh. In this scenario, ice-margin migration influenced by bedrock sticking points due to a lower-resolution mesh might give a false impression of good model fit to the data, but for reasons solely related to model resolution. Regardless, simulated ice in the KNS fjord system that is too extensive suggests that marine forcings not included in our model likely played an important role in ice-margin retreat. While implementation and treatment of calving in ice-sheet models continue to improve (e.g., Benn and Åström, 2018), our results highlight the fact that inclusion of calving is necessary towards a full understanding of ice-margin sensitivity to climate change in the KNS region. Yet, it is difficult to properly simulate calving in paleoclimate ice-sheet model setups that use coarse-resolution grids because fjord systems in Greenland are typically $< 5$ km, and high-resolution grids (1 km) are necessary to capture grounding-line migration (Seroussi et al., 2018). `TS10`

The simulated ice-margin positions in the Kangerlussuaq and KNS regions relative to the geological constraints offer insights into model–data fit across the entire model domain and the possible climatic conditions influencing ice-sheet behavior through the Holocene. Among the individual model runs of Cuzzone et al. (2019) and Briner et al. (2020), some simulated ice masses generally achieve a minimum between $\sim 7.6$ and 7.1 ka BP (simulations 1–6; Fig. 18), but there are three notable exceptions in which the ice-mass minimum occurs later at $\sim 6.3$ ka BP (simulations 7–9; Fig. 18). These simulations that result in a later ice-mass minimum also depict a minimum mass that is less extreme, with significantly less regrowth of the ice sheet following the minimum (Fig. 18). At the same time, these simulations with a later ice-mass minimum depict a more subdued, although still discernible, response to the 8.2 ka cooling event relative to other simulations despite the well-documented response of the southwestern GrIS to 8.2 ka cooling (Figs. 17 and 18; Young et al., 2011b, 2013b, 2020a). Nonetheless, relative sea level records from southwestern Greenland fall below modern sea level at $\sim 4$–3 ka BP before later rising towards modern sea level (Long et al., 2011; Lecavalier et al., 2014), which is interpreted to reflect the reloading of the crust during late Holocene regrowth of the southwestern GrIS initiating in the last few thousand years. Given all of the geologic ice-margin constraints from across southwestern Greenland considered here, the minimum extent of the GrIS likely occurred sometime after $\sim 5$ ka BP. Thus, simulations that produce an ice-mass minimum between $\sim 7.6$ and 7.1 ka BP appear to be inconsistent with the geological record. Simulations that result in an ice-mass minimum at $\sim 6.3$ ka BP are more compatible with the geologic record, especially when considering the fact that these model runs simulate a sub-

tle and broad post-6.3 ka BP plateau followed by slight late Holocene regrowth (Fig. 18).

Simulations 7 through 9 rely on a temperature history that has a muted early Holocene warming compared to other runs, followed by peak Holocene mean annual temperature anomalies above the 1850–2000 mean from 7 ka to 4 ka BP (Badgeley et al., 2020; Briner et al., 2020; Fig. 18). Run 7 relies on a precipitation history that has increased precipitation during 8 ka to 4 ka BP relative to the 1850–2000 CE mean, run 8 uses precipitation anomalies that are similar to the 1850–2000 CE mean, and run 9 uses precipitation anomalies that are lower than the 1850–2000 CE mean. Indeed, in the CE4 SMB-dominated Kangerlussuaq region, runs 7 and 8 appear to provide the best model–data fit of ice-margin position, while also having an ice-mass minimum most consistent with the geologic record (Fig. 18). The higher precipitation scenario used in run 7 is broadly supported by proxy evidence of enhanced wintertime snowfall in southwestern Greenland and inferred precipitation using an ice-sheet flowline model (Thomas et al., 2018; Downs et al., 2020). Despite early Holocene warming, limited evidence suggests that this warmth, and its effect on ice-sheet mass balance, may be offset to some degree by increased precipitation. The timing of the minimum inland extent of the ice margin may occur at slightly different times across southwestern Greenland, and therefore mass does not necessarily equate to the most retracted ice margin. However, across all model runs using different climatologies, the simulated ice-mass minimum generally equates to a modeled ice margin near or slightly inboard of its current position (Fig. 18).

Our results highlight the potential of proxy-informed gridded climate reconstructions and points to where continued improvements to climate reconstructions used in paleo-ice-sheet modeling efforts have the biggest impact; each different climate history applied here results in a slightly different simulated ice-sheet history. Simulated ice-sheet histories provide a better fit to geologic constraints in the SMB-dominated domain in the Kangerlussuaq region, with a relatively poorer model–data fit in the KNS region. Although some model–data mismatch occurs across our domain when considering lateral ice-margin position, rates of GrIS mass loss inferred for our domain (i.e., Briner et al., 2020) remain robust, as differences in simulated ice-margin migration (i.e., slightly too fast near Kangerlussuaq and too slow in the KNS compared to geologic constraints; Fig. 18) likely offset each other to some degree when considered in the context of total ice-mass loss. Ultimately, our results suggest that oceanic and dynamic processes that are not included in our modeling effort likely play a key role in dictating the ice-margin retreat pattern in the KNS region.

## 7 Conclusions

New $^{10}$Be ages from the KNS region, southwestern Greenland, constrain deposition of two separate segments of the Kapisigdlit stade moraines to $10.24 \pm 0.36$ and $9.57 \pm 0.38$ ka BP, indicating that these moraines are likely not equivalent features. The older moraine is synchronous with widespread moraine deposition in Baffin Bay at this time, whereas the younger moraine is consistent with an additional $^{10}$Be-dated moraine in southwestern Greenland, tentatively defining a new mode of GrIS moraine deposition at $\sim 9.7$ to 9.6 ka BP. Following early Holocene moraine deposition in the KNS forefield, the GrIS margin retreated inboard of the eventual historical maximum extent–modern ice margin. The timing of deglaciation of the landscape immediately outboard of the historical maximum extent in the KNS region and along much of the southwestern GrIS margin varies by several thousand years. Yet, additional chronological constraints provided by proglacial threshold lakes and cosmogenic nuclide measurements from recently exposed bedrock surfaces constrain an interval of several thousand years during which the GrIS margin was within but near its modern position and the minimum GrIS extent occurring sometime after $\sim 5$ ka BP. The variability in $^{10}$Be ages just outboard of the historical maximum limit is likely the result of slight variations in ice-sheet retreat and the magnitude of the late Holocene readvance of the GrIS, rather than major differences in ice-margin history or regional climate variability. The southwestern GrIS margin may have advanced back to the eventual historical maximum extent as early as $\sim 2$ ka BP, leaving an approximately 3 kyr window for the ice margin to achieve its minimum inland position.

Triple $^{10}$Be–$^{14}$C–$^{26}$Al measurements in recently exposed bedrock fronting the modern GrIS help constrain the minimum inland extent of the GrIS margin over multiple timescales. Our paired $^{26}$Al–$^{10}$Be and $^{14}$C–$^{10}$Be measurements are unable to detect any surface burial and are consistent with constant exposure of our sampled bedrock sites. $^{14}$C–$^{10}$Be measurements constrain the magnitude of pre-Holocene isotopic inheritance at our bedrock sites and reveal that the period of 18th–20th century ice cover was the only extended period that these bedrock sites became reoccupied by the GrIS following middle Holocene deglaciation. $^{26}$Al–$^{10}$Be measurements also reveal constant exposure of these sites and suggest that the slight amount of inherited $^{10}$Be present in a subset of our bedrock samples is due to a period of surface exposure prior to the Last CE5 Glacial Maximum (LGM), likely during MIS 5e. In situ $^{14}$C inventories indicate that bedrock presently emerging from beneath the GrIS in the KNS forefield was exposed during the middle Holocene. Contemporary retreat of the GrIS has yet to expose a landscape that remained ice-covered throughout the Holocene.

Geologic reconstructions in southwestern Greenland constrain the Holocene behavior of the GrIS across a land-

Please note the remarks at the end of the manuscript.

terminating region dominated by surface mass balance and an area where the ice margin retreated rapidly as the ice front responded to dynamical changes imposed through marine influences (i.e., calving and/or submarine melting of floating ice). As paleo-ice-sheet models continue to improve in terms of both the representation of processes controlling ice-margin migration and the use of high-resolution model meshes, these geologic reconstructions provide robust validation targets with which to benchmark and improve models simulating the past behavior of ice sheets. Continued use of model–data comparisons as ice-sheet modeling efforts, paleoclimate datasets, and geologic reconstructions become more refined will improve our understanding of past ice-sheet sensitivity to climatic and dynamic forcing mechanisms.

**Data availability.** All analytical information for new cosmogenic nuclide and traditional radiocarbon measurements is listed in the tables in the Supplement. Analytical information is also available from the ICE-D: Greenland online database (http://greenland.ice-d.org/ TS11 TS12 , Balco, 2020).

**Supplement.** The supplement related to this article is available online at: https://doi.org/10.5194/cp-17-1-2021-supplement.

**Author contributions.** The project was conceived by NEY, JPB, and JMS. NEY, AJL, and JKC wrote the first draft of the paper with input from JPB, JAB, ABK, and JMS. All authors commented on and edited the paper. NEY, AJL, JPB, JAB, BLG, and AC completed the fieldwork. JKC led the ice-sheet modeling component, and JAB provided the regional input climatologies. RS, NEY, and JMS completed the $^{10}$Be and $^{26}$Al extraction, and NEY completed the in situ $^{14}$C extraction. MWC and SRHZ conducted final $^{10}$Be and $^{26}$Al measurements. TT and EB completed final gas-source $^{14}$C measurements. NEY, ABK, and JLL completed the final in situ $^{14}$C data reduction and interpretations. AJL and JPB isolated organic remains for traditional $^{14}$C dating.

**Competing interests.** The authors declare that they have no conflict of interest.

**Acknowledgements.** We thank the 109th Airlift Wing of the New York Air National Guard for transport to and from Greenland, Air Greenland for helicopter support, and CH2MHill Polar Field Services for additional logistical support. We also thank J. Hanley and J. Frisch TS13 for help processing $^{10}$Be and $^{26}$Al samples. Funding was provided by the National Science Foundation Arctic Natural Sciences and Arctic System Sciences programs: award nos. 1417675 and 1503959 to Nicolás E. Young and Joerg M. Schaefer; award nos. 1417783 and 1504267 to Jason P. Briner. TS14

**Financial support.** This research has been supported by the NAME OF FUNDER (grant no. GRANT AGREEMENT NO). TS15

**Review statement.** This paper was edited by Alessio Rovere and reviewed by Anne Sofie Søndergaard and David Ullman.

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

## Remarks from the language copy-editor

CE1    Please note the slight edits to the affiliations (comma placement adjusted).

CE2    Should AMS be defined here for clarity?

CE3    Should DEM be defined here for clarity?

CE4    Should SMB be defined here for clarity?

CE5    Please check and confirm.

## Remarks from the typesetter

TS1    The composition of Figs. 6 and 14 has been adjusted to our standards.

TS2    Goehring et al., 2010, is not listed in the reference list. Please check.

TS3    Please confirm or provide an alternative short running title.

TS4    Please provide last access date.

TS5    Goehring et al., 2010, is not listed in the reference list. Please check.

TS6    Goehring et al., 2010, is not listed in the reference list. Please check.

TS7    Goehring et al., 2010, is not listed in the reference list. Please check.

TS8    Please note that units have been changed to exponential format throughout the text. Please check all instances.

TS9    Please check NGRIP. Is this a reference to the "North Greenland Ice Core Project members"?

TS10   Seroussi et al., 2018, is not listed in the reference list. Please check.

TS11   Please provide last access date.

TS12   Please provide a direct link to the data set and, if possible, a DOI instead of a URL.

TS13   Please provide full first names.

TS14   Please confirm author names.

TS15   Please note that there is funding information given in the acknowledgements but you have not indicated any funding upon manuscript registration. Therefore, we were not able to complete the financial support statement. Please fill the missing information and double-check your acknowledgements to see whether repeated information can be removed from the acknowledgement. Thanks.

TS16   Please check DOI.

TS17   Please confirm added year.

TS18   Please provide volume and page range or article number.

TS19   Goehring et al., 2011, is not mentioned within the text; please add or delete it from the reference list. Please also check whether the years are switched – should the references in the text for Goehring et al., 2010, be changed to 2011, or should this reference entry be changed to 2010?

TS20   Please provide volume.

TS21   Please check DOI.

TS22   Please check DOI.

TS23   Please confirm added year.

TS24   Please provide volume.

TS25   "Seroussi and Morlighem (2018) is not mentioned within the text, please add or delete it from the reference list.

TS26   Please confirm added year.

TS27   Please provide page range or article number and DOI or publisher and place of publication.

TS28   Please check DOI.

TS29   Please provide page range or article number.

TS30   Please check DOI.