# Peer review of "In situ* cosmogenic 10Be-14C-26Al measurements from recently deglaciated bedrock as a new tool to decipher changes in Greenland Ice Sheet size"

_Climate of the Past, 2020_

## Referee Comment (RC1) · Anne Sofie Søndergaard (Referee) · 3 Nov 2020

The paper by Young et al. provides new cosmogenic exposure ages (26AL, 10Be and 14C) from recently deglaciated bedrock in the KNS region, southwest Greenland. Based on their data, the authors find an early Holocene ice retreat behind its modern margin, where after the ice stabilized for several thousand years. The minimum extent of the GrIS likely occurred between c. 5 and 2 ka. Including previous studies and modelling the authors look into the inland retreat of the southwest GrIS. The study is interesting, especially the three isotope combination, which is intriguing and hopefully can help to further implement the use of especially in-situ 14C for chronological con-

straints in Greenland. The study is well explained and adds to our knowledge on the glacial history in the KNS region and combined with previous studies conclude on the broader southwest Greenland glacial history. The implementation of modelling of the GrIS margin highlight not only differences within southwest Greenland and the KNS region but also emphasize data model misfits and the overall importance of implementing ocean-forcing into ice sheet models.

I have a few main comments/suggestions listed here below. The rest of my comments are divided into specific and technical comments all regarding both the text, figures, and tables. I hope the authors will address these prior to publication. Thank you for an interesting read and I look forward to see the final version in print!

MAIN COMMENTS

While reading the manuscript I missed having some of the most relevant tables included into the main text. In general, there are a lot of figures in the text, so I suggest to either move some of those figures to the supplementary material or possible merge some of the figures together, to make room for tables in the main manuscript. You have several figures with pictures of samples, I suggest to move/merge some of these or perhaps make more figures like figure 12.

In section 5.3 you focus on the inheritance in the 10Be samples, and conclude that the easiest explanation for this is exposure during MIS 5e. While you do comment and elaborate on the possibility of MIS 3 exposure, I miss more firm evidence for excluding this possibility. I acknowledge that the in-situ 14C ages do not seem to be affected by inheritance, which therefore limits the possibilities of MIS 3 exposure, but could it be so, that the sample areas experienced exposure during MIS 3, possibly in the earlier part, were then buried for >20 ka which together with a certain amount of erosion could make the samples reach undetectable limits more quickly (as you state in the text: previously accumulated in situ 14C to decay to undetectable levels after ~30 ka of simple burial of a surface by ice; with the aid of subglacial erosion, in situ 14C can reach undetectable

levels more quickly). Could the authors elaborate on why this is not the case? Would it be possible to include some simple model runs, to further exclude MIS 3 exposure?

A personal comment on the title of the manuscript: I struggle with calling the combination of the three isotopes a "new tool" to track GrIS changes – it is a rather new approach to combine these three isotopes, but all of them are commonly used to track ice sheet changes. This is optional, but consider changing the title to something less promising like "Combining 10Be-26Al-14C cosmogenic isotope measurements from recently deglaciated bedrock reveal changes in Greenland Ice Sheet size".

SPECIFIC COMMENTS

Text Lines 17-18: What about the size of the GrIS during the Neoglacial? I believe it was larger than its current configuration in some places in Greenland? Possibly define late Holocene differently or make a comment regarding the Neoglacial/southwestern Greenland.

Lines 188-189: Could you elaborate a bit on the chosen scaling scheme? Why choose that, when, as you mention, changes in the geomagnetic field over time are minimal at high latitudes? Could you make a small comment on how much ages deviate using the other scaling schemes?

Lines 289-291: Consider moving the lines "Silt. . .. diverted elsewhere" to the methods section.

Lines 305-325: As I read it here you have a maximum limiting age outboard the moraines of 10.23 ka, date the moraines to 10.24 ka and have minimum limiting ages inside the moraines of >10.25 ka – I know the ages overlap within uncertainty, but could the authors comment on this age distribution? Does it show a very rapid deglaciation and how does it fit with moraine formation?

Line 324: Consider ending the sentence at "ka BP" – and then move the rest of the sentence "all available supporting" to line 330-332 where you comment on the same.

Or you could simply delete it.

Lines 337-339: As this might be true, I feel it is a rather big conclusion made from two samples/ages – could the authors elaborate a bit, possibly include other data to underly the statement?

Lines 354-355: How "well" do you believe this age to constrain the timing of local deglaciation? The age seem relatively young compared to the KNS site, but fits relatively well with previous findings from this area. You discuss this in greater detail later, but could you use a sentence here to give the readers a sense of how much value you put into this age?

Lines 435-436: This is interesting, do you have any idea why that is so? Geomorphology, samples, erosional features? It seems the three pairs of ages where the 10Be age » 14C age are from the same sampling site - what is special about it?

Lines 444-446: Could the authors elaborate on these combinations (less Holocene exposure and/or more subglacial erosion)? What do you consider more likely?

Line 501: Suggesting to delete "The inferred... Jakobshavn Isbræ" and instead start the sentence "However, there are key differences" – as it is now you repeat yourself.

Lines 508-512: I read here that you favour a scenario in which the GrIS deposited the moraines at c. 10 ka, and then stayed within very close proximity over the next 2-3 ka? How does this compare to your conclusions in section 4.2 (lines 330-332 – here you state that the ice retreated within the historical limit shortly after deposition at c. 10 ka? Could you elaborate a bit more on the spatial extent of this retreat in section 4.2?

Line 604: Could you briefly include a definition of "Baffin Bay" here? It is a rather large area and I don't believe widespread moraine deposition at this time interval is known from northern Baffin Bay/Northwest Greenland? As I read it you mention southwest and west Greenland as well as Baffin Island.

Lines 620-624: I suggest moving these lines "Lastly, we note.... advance of the GrIS",

to section 6.2, as you here discuss the retreat of the GrIS behind the historical maximum/modern margin.

Lines 636-641: Including data from Saqqap Sermia, you argue for a temporal difference of more than 5 kyr between deglaciation outboard the historical moraines in the KNS region – is the data from Saqqap Sermia the only to represent this relatively late deglaciation in the entire region, and if so, how much do you rely on this?

Lines 649-652: This is interesting, could you perhaps comment on where you would expect a greater or smaller re-advance of the ice margin, and what that would mean for the interpretation of your data? Could it be so, that places with younger deglaciation ages have experienced a smaller re-advance and areas with older deglaciation ages experienced a larger re-advance – so you possibly have a contemporary deglaciation across the region as oppose to the 5 kyr difference? Or can you completely reject this scenario?

Lines 692-694: From what I read here you base the 5-2 ka BP "window" mostly on data from other studies – could you make a small comment on your own findings in according to this age constraint – based solely on your findings would the "window" not be a couple of thousand years longer, with initial retreat c. 7 ka BP? I assume some of the explanation lies in the discussion of different ice-margin environments, that you give in lines 722-734?

Lines 748-764: It seems that model runs simulate an ice sheet minimum that to some degree fits with your data from the KNS region (as stated in the comment above) – could you briefly outline why/why not the models and your data fit/does not fit? Why you believe in the 5-2 minimum, and not an earlier retreat behind the present day margin?

Figures and Tables

I find that much of the text in the figures (place names, ages, lat/long) is difficult to read

[Figure]

and could benefit from a larger font size.

Table S3: Just a comment on the high accuracy of the sample thicknesses. As this might be the accuracy of the caliper (or whatever instrument you have used), I find it difficult to work with high accuracy numbers like this, on what I assume are rather uneven samples. Are sample thicknesses a mean of several measurements?

TECHNICAL CORRECTIONS

Text

Line 60: Consider to use only surface or surfaces in this sentence. "..when those surfaces are exposed. . .

Line 93: "reworked" is spelled "re-worked" everywhere else in the text. Consider changing this.

Line 105: Check spelling of Kangâsarsuup throughout the text – some places you spell it with "â" and some places with "a". Further, judging from Figure 1, it seems that Kangâsarsuup Sermia is located about 20 km more than 35-40 km south of KNS?

Line 109: "Trim-lines" is spelled without "-" a few places, consider changing for consistency.

Line 115: You have "cosmogenic-nuclide" (as here) and "cosmogenic nuclide" (eg. line 800) in the text. I assume you want one consistent way of spelling?

Line 120: Consider using the abbreviation "GOOF" for Goose Feather Lake in the text. You use it in the caption to Figure 6, but nowhere else. Same comment for Marshall Lake.

Line 128: Space missing between "61" and "10Be".

Lines 162+165: I assume you mean "in situ 14C" here?

Line 323: I believe you mean production "rate" uncertainty?

Line 369: Delete the "space" between "therein)" and "." at the end of the line.

Lines 538: Is there a "to" missing between "relative" and "10Be"?

Line 551: I think you miss a "with" before "recent modelled".

Line 588: I think there is a "the" missing before "so-called"?

Figures and tables

Tables: Generally, consider consistency – you have Latitude (N)/Longitude (W) (Table S2), Latitude/Longitude (Table S3), Latitude (°N)/Longitude (°W) (Table S7).

Figure 1: Suggesting to either find another symbol or make the orange diamond larger – it is difficult to distinguishing from the orange circles.

Figure 2: I would remove "(ka BP +- 1 S.D.)" to right after "in situ 14C ages" instead of having it at the end of the line.

Figure 6: You have (panel A) and panel (a). Consider writing it in the same way.

Figure 16: A "." is missing after "(section 5.2)".

Table S8: I think you have a double spacing at the beginning of the last line.

---

## Referee Comment (RC2) · David Ullman (Referee) · 13 Nov 2020

General Comments

This manuscript presents a new cosmogenic isotope chronology to help constrain retreat of the western Greenland Ice Sheet (GrIS) during the earlier Holocene. It has long been surmised that the GrIS was smaller than present during parts of early Holocene, but constraining the magnitude of margin retreat and its timing has been difficult due to the late-Holocene ("Historical") readvance covering much of the previously exposed surfaces. By utilizing 3 different cosmogenic nuclides (10Be, 14C, and 26Al) on sampling locations, the authors provide a clear picture of the complexity of nuclide inheritance, erosion, and exhumation that can sometimes confound cosmogenic nuclide interpretations. This is especially true in regions with a complex ice retreat history, such as that of the southwestern GrIS during the Holocene. This new dataset is compared with a robust compilation of existing exposure and lake chronologies from all along the historical extent of the southwestern GrIS that broadens the scope and significance into that of a larger regional signal. In addition to the chronological constraints provided in this manuscript, the authors also present new results from a high-resolution ice sheet model for the southwestern GrIS to help constrain the magnitude of retreat past the modern margin extent. These simulations are forced with temperature and precipitation from recent data assimilation efforts for Greenland. These model experiments explore a range of climate space, thus incorporating some parametric uncertainty in the results. Such a robust set of experiments allows for the exploration of a variety of possible solutions for Holocene margin retreat, a comparison with chronological constraints to constrain the model, and an assessment of model limitations (e.g. lack of iceberg calving). Generally, the manuscript provides a set of results that are consistent with improving our understanding of GrIS retreat during the Holocene. It is clear and well structured. I have a few concerns related to statistical significance and a need for a little more description about the modelling approach. After addressing these comments (mostly technical), I believe this manuscript is worthy of publication.

Specific Comments

Separate retreat timing from Kapisigdlit moraines? - One of the main conclusions is that the Kapisigdlit moraine deposition occurred with different timing for KNS (10.24 ± 0.36 ka) and Qamanaarsuup Sermia (9.57 ± 0.38 ka), thus suggesting a new mode of GrIS moraine deposition during the Holocene. However, don't these two mean ages overlap at $1\sigma$? The overlap is small, but an overlap nonetheless. It seems statistically possible that these two ages are equivalent. At the very least, it would be good to show t-test statistics to help show the level of statistical difference between these ages. If the difference between these ages is not significant, then some of the wording of the paper

may need to be modified to indicate the possibility of synchronous moraine deposition.

Ice Sheet Model Methods (section 2.7) – I think it would be good to offer a little more description on the nature of the model setup and experiment design. I realize this model is extensively described in Cuzzone et al (2019) and Briner et al. (2020), but there are some crucial distinctions that could be added here that would help in understanding the results. For example: -What is the nature of the surface mass balance calculations? PDD? -Provide a general description of the flow dynamics. -mention the lack of calving in the methods (it gets brought up later, but it would be good to mention such model limitations in the methods -More description of the 9 model combinations/experiments (line 238-239). How were these 9 permutations selected? How do they differ?

Discussion of marine Terminating Dynamics (lines 748-764) – How much of the full model domain is influenced by marine dynamics and iceberg calving? Since these regions minimize retreat in the model, is there a way to show or discuss how much of the model domain would be influenced by this model limitation. Are any of the margins still marine-terminating at the minimum Holocene extent?

Technical Comments

Lines 38-40 – This compound sentence is confusing. Consider breaking up into 2 sentences

Line 201 – Specifically, what is the production rate uncertainty that was used in quadrature? Is it a constant percentage? Or is it spatially varying?

Line 365 – "mean age of 10.20 ± 0.14 ka (10.27 ± 0.23 ka with production-rate uncertainty)." Why are the mean ages (10.20 and 10.27) different? Shouldn't the production rate uncertainty added in quadrature only effect the uncertainty value? Also, double check on these values are appropriately displayed in Figure 2.

Lines 373-377 – Would it be possible to include this early photograph with permission? It would be nice to see this photograph annotated to show the ice extent and trim-line

as described in the text.

Line 437 – I think the use of the phrase "more proximal" is confusing here. These high elevation sites are less proximal from the historical limit, when considering ice position. I would say that the recently deglaciated sites are "more proximal" than the historical limit. Do you mean that the high elevation sites are closer to the historical maximum limit?

Lines 508-509 – "we favor an interpretation that couples less site exposure over significant amounts of subglacial abrasion" – I think the use of the word "couples" is confusing here. What is being coupled to what?

Figure 1 – The orange diamond of Weidick et al (2012) is very hard to distinguish from other yellow circles. Consider using an alternative symbol.

Figure 2 & Figure 3 – In general it can be difficult to distinguish between italicized outliers and non-outliers. Is there a different way to distinguish outliers other than italics?

Figure 17c and 17d – What does this distance axis mean? Is it distance from the coast, or some other arbitrary point? Would be good to clarify in the caption.

Figure 17c and 17d – Is there a way to better display the present day location? The yellow dots are hard to see on the first pass

Figure 17c-17f – On each of the model result figures, it is hard to distinguish between the green lines and the blue lines. Would it be possible to use a more distinct color gradient for these groupings of simulations?

---

## Editor Comment (EC1) · Alessio Rovere (Editor) · 26 Nov 2020

Dear Authors, I am closing the interactive discussion. Both reviewers agree that the MS is a good contribution to Climate of the Past, but have moderate/minor suggestions. I strongly advise you to take them into account while you prepare the next version of your MS.

---

## Author Response (AR1)

We have included the original reviewer comments, our initial response to comments, and list of edits within this document. The original comments are in **black,** our response in **red**, and our manuscript edits in **blue.**

**Response to Comments**
*Reviewer #1: Anne Sofie Søndergaard*

The paper by Young et al. provides new cosmogenic exposure ages (26AL, 10Be and 14C) from recently deglaciated bedrock in the KNS region, southwest Greenland. Based on their data, the authors find an early Holocene ice retreat behind its modern margin, where after the ice stabilized for several thousand years. The minimum extent of the GrIS likely occurred between c. 5 and 2 ka. Including previous studies and modelling the authors look into the inland retreat of the southwest GrIS. The study is interesting, especially the three-isotope combination, which is intriguing and hopefully can help to further implement the use of especially in-situ 14C for chronological constraints in Greenland. The study is well explained and adds to our knowledge on the glacial history in the KNS region and combined with previous studies conclude on the broader southwest Greenland glacial history. The implementation of modelling of the GrIS margin highlight not only differences within southwest Greenland and the KNS region but also emphasize data model misfits and the overall importance of implementing ocean-forcing into ice sheet models. I have a few main comments/suggestions listed here below. The rest of my comments are divided into specific and technical comments all regarding both the text, figures, and tables. I hope the authors will address these prior to publication. Thank you for an interesting read and I look forward to see the final version in print!

Thanks for taking the time to read this manuscript and provide feedback.

*While reading the manuscript I missed having some of the most relevant tables included into the main text. In general, there are a lot of figures in the text, so I suggest to either move some of those figures to the supplementary material or possible merge some of the figures together, to make room for tables in the main manuscript. You have several figures with pictures of samples, I suggest to move/merge some of these or perhaps make more figures like figure 12.*

We thought a lot about the balance of figures, tables, and the overall level of detail in the manuscript's text during our initial submission. We understand the reviewer's comment, but prefer to keep the figures and tables as is in the main manuscript text and supplement for a number of reasons. As it stands, this is a lengthy manuscript due to the different types of data that we present (e.g. number of isotopes, ice-margin chronology, erosion constraints, ice-sheet modelling), so we are hesitant to move the supplemental tables, and there are 9 of them, into the main text. As the reviewer points out, one of the key contributions in this manuscript is our triple isotope work from recently deglaciated bedrock surfaces. In this regard, we think it is important to showcase the types of bedrock surfaces and their geologic context we are targeting for this study.

As a consequence, we prefer to keep these sample-site figures in the main text. We also chose COP for this submission because it is a long-format journal (and open-access) and we wanted to use this format to highlight the unique field-based Quaternary expertise of our team and how these field observations are critical for this study. We think this is particularly true for this manuscript as it relies on subtle differences in morphostratigraphy, perhaps more so than typical exposure dating-based studies that we are accustomed to. Thus, we think keeping the sample-based and morphostratigraphic setting-based figures in the main text is critical to the interpretation of our results and overall manuscript presentation.

As the cosmogenic nuclide method is applied more widely, it has become commonplace to include the relevant tables in supplemental material, which are mainly used to re-calculate exposure ages. Indeed, we were inspired by current PhD student and co-author Alexandra Balter-Kennedy's recent publication in COP's sister journal *Cryosphere* where that author team merged a robust cosmogenic nuclide-based dataset, numerous field/sample photos, and a thorough description and discussion, into a well-balanced manuscript, while at the same time leaving many of the geochemical details in the supplemental tables that can be accessed as needed.

No changes made. We prefer to keep the tables in the supplemental material and the field photos in the main text.

*In section 5.3 you focus on the inheritance in the 10Be samples, and conclude that the easiest explanation for this is exposure during MIS 5e. While you do comment and elaborate on the possibility of MIS 3 exposure, I miss more firm evidence for excluding this possibility. I acknowledge that the in-situ 14C ages do not seem to be affected by inheritance, which therefore limits the possibilities of MIS 3 exposure, but could it be so, that the sample areas experienced exposure during MIS 3, possibly in the earlier part, were then buried for >20 ka which together with a certain amount of erosion could make the samples reach undetectable limits more quickly (as you state in the text: previously accumulated in situ 14C to decay to undetectable levels after ~30 ka of simple burial of a surface by ice; with the aid of subglacial erosion, in situ 14C can reach undetectable levels more quickly). Could the authors elaborate on why this is not the case? Would it be possible to include some simple model runs, to further exclude MIS 3 exposure?*

At the core of our discussion we state "we cannot rule out small amounts of inherited $^{10}$Be…..is a result of exposure during MIS3" (lines 565-566).

We are aware that exposure during MIS3 is technically possible, as is exposure during the last Interglacial, and we do not "exclude" MIS3 exposure as a possibility. At this point, the discussion is simply which is more likely, exposure during MIS3 or MIS5e as both are mathematically possible. In this section we point out that the consistency in all of our cosmogenic isotope measurements, coupled with the somewhat widely accepted hypothesis that the GrIS was likely smaller than today during the last Interglacial and that the region was likely warmer during MIS 5e than MIS 3, points to MIS 5e exposure as being the more likely culprit for the inherited $^{10}$Be in our samples. In contrast, we are a bit hesitant to suggest that the consistency in our cosmogenic isotope measurements coupled with scattered and somewhat ambiguous evidence of a reduced GrIS during MIS 3 (i.e. based on very low carbon $^{14}$C ages that could suffer from contamination) is just as likely responsible for the inherited $^{10}$Be. We certainly can do a simple model where we accumulate $^{14}$C early in MIS 3 and then just let that inventory of $^{14}$C decay away, but the concern is that we would give readers the impression that we think the MIS3-exposure scenario is just as likely as the MIS5e-expsure scenario. All things being equal and in the absence of any complimentary evidence, then surface exposure during MIS3 or MIS5e seems equally likely. However, after considering sedimentological evidence from offshore southern Greenland that points to a reduced GrIS during MIS 5e (Colville et al., 2011, *Science*), the idea that closing the global MIS 5e eustatic sea-level budget likely requires at least some contribution from Greenland (i.e. a reduced MIS 5e GrIS; Dutton et al., 2015, *Science*), and that the region was likely warmer during MIS 5e vs. MIS 3 (NGRIP, 2004, Nature; NEEM, 2013, Nature) we simply prefer what appears to be the more straightforward explanation that our inherited $^{10}$Be is a product of MIS 5e exposure. This is how we currently have it presented in the text while readily acknowledging that MIS 3 exposure cannot be ruled out.

To some degree this issue highlights the inherent weakness in multiple isotope systems where there are technically infinite solutions. For example, brief exposure during interstadial MIS 5a, instead of 5e or 3, is also mathematically possible. Nonetheless, we can include this simple model following the reviewer's suggestion, but we prefer to make it clear in the text that while MIS 3 exposure is possible, we think MIS 5e exposure is a simpler explanation after considering additional lines of geologic evidence.

After having re-read the comment and our initial manuscript text again, we feel that we adequately addressed the question surrounding MIS 5e vs MIS 3 exposure. We readily admit in our text that our sites could have been exposed during MIS 3 instead of MIS 5e, but we point out that MIS 5e is the more likely culprit. We think if this were a case where significant complimentary evidence suggested the GrIS margin was significantly retracted during MIS 3, or we were somehow able to rule out MIS 5e exposure through some other line of evidence, we would be more inclined to explore this scenario with our dataset. We feel that additional modelling of the accumulation and decay histories of $^{10}$Be, $^{14}$C, and $^{26}$Al, within the context of MIS 3, would be uncomfortably speculative.

In summary, in line 646 we state:

*"At the same time, we cannot rule out that small amounts of inherited $^{10}Be$ coupled with $^{26}Al/^{10}Be$ ratios consistent with constant exposure, is a result of exposure during MIS 3."*

In line 656 we state: *"It is certainly possible that our sites were exposed during the MIS 3…"*

In addition, we added in lines 662-664: *"…yet considering that 1) ice-core records reveal that the region was likely warmer during MIS 5e versus MIS 3 (NGRIP, 2004; NEEM, 2013), 2) balancing the MIS 5e eustatic sea-level budget likely requires a significant contribution from Greenland (Dutton et al., 2015), and 3) the lack of any additional terrestrial evidence in southwestern Greenland for a MIS 3 ice-sheet configuration similar to or more restricted than today, we simply favor the more straightforward explanation of MIS 5e exposure at our sample sites."*

*A personal comment on the title of the manuscript: I struggle with calling the combination of the three isotopes a "new tool" to track GrIS changes – it is a rather new approach to combine these three isotopes, but all of them are commonly used to track ice sheet changes. This is optional, but consider changing the title to something less promising like "Combining 10Be-26Al-14C cosmogenic isotope measurements from recently deglaciated bedrock reveal changes in Greenland Ice Sheet size"*

We disagree with this assessment. Widespread application of $^{10}Be$ and $^{26}Al$ is common, application of $^{14}C$ is not. Combining all three isotopes in the same sample is rare, in fact, the only other example that comes to mind is Miller et al (2006; QSR). What is unique here is the widespread combination of all three isotopes within surfaces that have only become ice-free in the last few decades to a century. We are unaware of any set of measurements that focuses on recently deglaciated bedrock in this manner. And, a key aspect of the work presented here is that detailed knowledge of the initial (Holocene) surface dosing history makes this triple isotope tool much more useful. We prefer to leave the title unchanged.

After thinking about this more, we have replaced "isotope" in the title with "$^{10}Be$-$^{14}C$-$^{26}Al$" and added "In-situ" at the start. This should satisfy everyone. We have also added a statement that elaborates a bit more on the (extremely limited) history of using triple 10Be-14C-26Al measurements. In lines 80-82, we added:

*"Triple $^{10}Be$-$^{14}C$-$^{26}Al$ measurements have, to the best of our knowledge, rarely been made (e.g., Miller et al., 2006; Briner et al., 2014), and have not been utilized in any systematic fashion in recently deglaciated environments."*

We have added Miller et al., 2006 and Briner et al., 2014 to the reference list and note that Miller et al was published in Quaternary Geochronology and not Quaternary Science Reviews as we stated in the reply to comment above.

*Text Lines 17-18: What about the size of the GrIS during the Neoglacial? I believe it was larger than its current configuration in some places in Greenland? Possibly define late Holocene differently or make a comment regarding the Neoglacial/southwestern Greenland.*

We can update the wording. However, we are providing the broad strokes about the GrIS here in the abstract. In general, and what this manuscript focuses on, is that the current configuration and modern/Little Ice Age limit are quite often one in the same or extremely similar. This was really meant to highlight that in most places the GrIS margin was inland during the middle Holocene.

The first sentence of the abstract has been changed to *"Sometime, during the middle to late Holocene (8.2 ka BP to ~CE 1850-1900), the Greenland Ice Sheet (GrIS) was smaller than its current configuration"*

This is meant to convey the broad strokes of GrIS change, not every little detail where this might not be true in the strictest sense. This edit also defines the late Holocene as ending with the historical maximum/Little Ice age extent (CE 1850-1900), which should help clarify this sentence.

*Lines 188-189: Could you elaborate a bit on the chosen scaling scheme? Why choose that, when, as you mention, changes in the geomagnetic field over time are minimal at high latitudes? Could you make a small comment on how much ages deviate using the other scaling schemes?*

Even though changes in the geomagnetic field is extremely minimal over time at this high latitude, we should, as good practice, probably use a scaling scheme that at least attempts to account for these changes (e.g. Lm). Using St scaling, which does not account for geomag changes results in a nearly identical age (often within a year) because the sites are at such high latitude and all the production rate calibration datasets are located at high latitude.

Added on line 200: "*ages calculated using 'St' scaling, which does not account for changes in the magnetic field results in almost identical ages (<10 years) because the calibration sites are all located at high latitudes*"

*Lines 289-291: Consider moving the lines "Silt. . .. diverted elsewhere" to the methods section*

We think the introduction of what a proglacial-threshold lake actually is should remain where we have it instead of in the methods sections. Any lake in a glacial environment can be a proglacial-threshold lake, but you do not know if it is or not until you core the lake and see the sediment stratigraphy. For example, if we cored these lakes and cored nothing but organic sediments, then we would be hesitant to call it a threshold lake. In the methods section, we prefer to leave the wording as is where we simply state that we cored two lakes. It is not until results that we really know these are threshold lakes, and then we explain how alternating silt and minerogenic sequences are achieved.

No change has been made. See reply to comment.

*Lines 305-325: As I read it here you have a maximum limiting age outboard the moraines of 10.23 ka, date the moraines to 10.24 ka and have minimum limiting ages inside the moraines of >10.25 ka – I know the ages overlap within uncertainty, but could the authors comment on this age distribution? Does it show a very rapid deglaciation and how does it fit with moraine formation?*

All of these ages overlap meaning that deglaciation and moraine deposition all occurred within the resolution of our chronometer.

In lines 366-368 we added:

"*Furthermore, our statistically identical $^{10}$Be ages from outboard and inboard of the Kapisigdlit stade moraine, as well as from moraine boulders themselves, indicate that moraine deposition occurred rapidly within the resolution of our chronometer.*"

*Lines 337-339: As this might be true, I feel it is a rather big conclusion made from two samples/ages – could the authors elaborate a bit, possibly include other data to underly the statement?*

We do not really think this is a big conclusion....I suppose we can cite a paper or two that list basal radiocarbon ages from southwestern Greenland that are not too different (i.e. older; Bennike and Bjork, 2002) than what is known about the regional deglaciation chronology. We can perhaps word this so we don't give the impression that we endorse dating bulk sediments (we certainly do not), but broad strokes, a bulk basal age is unlikely to give you a seriously erroneous age in southwestern Greenland. For example, in Young et al., 2015 (QSR) two sets of paired macrofossil-bulk radiocarbon ages from the same horizon yielded statistically identical ages.

Added beginning on current line 384:

"*We do not advocate the use of bulk sediments for developing down-core chronologies when macrofossils are available, but paired macrofossil-bulk sediment measurements from the same horizon often yield similar or indistinguishable radiocarbon ages in southwestern Greenland (e.g., Kaplan et al., 2002; Young and*

*Briner, 2015) suggesting that bulk sediments will not yield significantly erroneous radiocarbon ages. These similarities in southwestern Greenland likely result from several factors 1) a large fraction of humic acid extracts are aquatic in origin (Wolfe et al., 2004), 2) mainland southwestern Greenland comprises almost entirely of crystalline bedrock thereby minimizing potential hard-water effects, and 3) the lack of a significant accumulated carbon pool during the initial phase of ecosystem development (i.e., Wolfe et al., 2004). This latter point may be particularly influential in southwestern Greenland as this region rests well inboard of the GrIS margin during glacial maxima (located on the continental shelf), resulting in a landscape that is likely ice covered for a significant fraction of each glacial cycle. Furthermore, this sector of the GrIS appears is primarily warm-based and erosive thereby further minimizing the likelihood of old carbon accumulating on the landscape at lower elevations.*"

*Lines 354-355: How "well" do you believe this age to constrain the timing of local deglaciation? The age seem relatively young compared to the KNS site, but fits relatively well with previous findings from this area. You discuss this in greater detail later, but could you use a sentence here to give the readers a sense of how much value you put into this age?*

It is a little unclear what the reviewer is asking. All this section does is list the best, and sometimes only (as in this case here) available deglaciation constrain beyond the historical maximum position. We state here that this constraint is only from a single $^{10}$Be age. We certainly wish we had more $^{10}$Be ages, but this a bit of "it is what it is" situation. In addition, this single age constraint isn't really discussed later in the text as it is not really important to the overall deglaciation chronology. On line 635-636 we again mention that this constraint is from a single age, but beyond that there is not really anything we can do with this. We thought that by highlighting twice that this deglaciation age is based on a single $^{10}$Be age it would be implied that it is not the most robust age constraint in the region.

No changes were made. See reply to comment.

*Lines 435-436: This is interesting, do you have any idea why that is so? Geomorphology, samples, erosional features? It seems the three pairs of ages where the 10Be age » 14C age are from the same sampling site - what is special about it?*

We can mention in a revised version that there we didn't notice anything different about any of the bedrock sites; they look the same. In fact, considering our team's extensive experience sampling in southwestern Greenland, we were surprised to find any inheritance whatsoever.

Added on line 500: "*despite all sample sites appearing to have undergone significant subglacial erosion*"

*Lines 444-446: Could the authors elaborate on these combinations (less Holocene exposure and/or more subglacial erosion)? What do you consider more likely?*

This comment is a bit confusing. Lines 444-446 is the last sentence of section 5.1 and is meant to act as a segue to the next section. All of section 5.2 elaborates on the very question the reviewer asks here.

No changes have been made. The entire section following this sentence is devoted to this exact question, now beginning on line 517.

*Line 501: Suggesting to delete "The inferred. . . Jakobshavn Isbræ" and instead start the sentence "However, there are key differences" – as it is now you repeat yourself.*

Ok

Yes, good catch, cleaned up the wording. This is on current line 574.

*Lines 508-512: I read here that you favour a scenario in which the GrIS deposited the moraines at c. 10 ka, and then stayed within very close proximity over the next 2-3 ka? How does this compare to your conclusions in section*

*4.2 (lines 330-332 – here you state that the ice retreated within the historical limit shortly after deposition at c. 10 ka? Could you elaborate a bit more on the spatial extent of this retreat in section 4.2?*

The reviewer is absolutely correct - we favor a scenario where the GrIS deposits moraines and crosses behind the historical maximum at 10 ka and then likely stays within close proximity for another 2-3 ka. I think we are all on the same page at this point. However, this manuscript is structured so that first we use [10]Be and/or traditional [14]C from beyond the historical maximum to simply constrain the timing of landscape deglaciation. This is what we do in section 4.2, and this section's only purpose is to develop classic deglaciation constraints and set the stage for the rest of the manuscript. Another way to put it is that section 4's only conclusions are stating the local to regional deglaciation constraints, nothing more. The next part of this manuscript is then addressing how do we gain any insight into what happens after initial deglaciation. To do this we have to introduce all these new tools, including the triple isotope measurements from recently exposed bedrock. This is what we have done in section 5. Therefore, we cannot take what we have learned about the ice margin in section 5 and then insert it into section 4. We prefer to leave the manuscript's structure as is.

No changes made and manuscript structure remains unchanged. See response to comment.

*Line 604: Could you briefly include a definition of "Baffin Bay" here? It is a rather large area and I don't believe widespread moraine deposition at this time interval is known from northern Baffin Bay/Northwest Greenland? As I read it you mention southwest and west Greenland as well as Baffin Island.*

I suppose we can add "southern" in front of Baffin Bay. The broader point here seems to be we have worked across a significant part of the Baffin Bay region (e.g. west and southwest Greenland, Baffin Island).

Our team has worked in many places across Baffin Bay. We use Baffin Bay here in the same sense that the term "North Atlantic" is often generically used in our field when nobody has worked everywhere in the "North Atlantic". We added "*in southwestern Greenland and Baffin Island*" and "*across several locations in Baffin Bay*" starting at line 713

*Lines 620-624: I suggest moving these lines "Lastly, we note. . .. advance of the GrIS", to section 6.2, as you here discuss the retreat of the GrIS behind the historical maximum/modern margin.*

In a previous internal draft of this manuscript, we actually had this statement in the next section as the reviewer suggests. However, after further consideration and a round of internal comments, the consensus among authors was to mention any potential 8.2 ka event moraines along with the rest of the broader southwest Greenland moraine chronology discussed in this section. Having these two sentences in the next section proved to be distracting as this section is primarily concerned with the mid-Holocene minimum extent of the GrIS. In its current position, we think these few lines serve as a nice segue to the next section where we fully discuss retreat behind the modern margin.

No changes made. See response to comment.

*Lines 636-641: Including data from Saqqap Sermia, you argue for a temporal difference of more than 5 kyr between deglaciation outboard the historical moraines in the KNS region – is the data from Saqqap Sermia the only to represent this relatively late deglaciation in the entire region, and if so, how much do you rely on this?*

We looked at the Saqqap dataset in detail (Levy et al) and consider it exceptionally solid; we have no reason not to trust the Saqqap dataset. The broader message here is that the 5 kyr spread in deglaciation ages in the KNS region that includes the Saqqap dataset is similar to the spread in deglaciation ages when you consider all of southwestern Greenland (our Fig. 16a).

No changes made. See response to comment.

*Lines 649-652: This is interesting, could you perhaps comment on where you would expect a greater or smaller re-advance of the ice margin, and what that would mean for the interpretation of your data? Could it be so, that places with younger deglaciation ages have experienced a smaller re-advance and areas with older deglaciation ages experienced a larger re-advance – so you possibly have a contemporary deglaciation across the region as oppose to the 5 kyr difference? Or can you completely reject this scenario?*

Most of this is addressed in lines 638-648 directly prior to what the reviewer is pointing out, and the rest of this section after line 652 is highlighting that we probably shouldn't place too much emphasis on the deglaciation age beyond the historical limit. For example, lines 653-667 highlight how these site-to-site differences in deglaciation ages might occur yet at the same time not really signify drastic differences in ice-margin behavior. Site to site differences are almost certainly dictated by local topographic conditions or slight variations in ice margin behavior but, overall, deglaciation is somewhat/largely contemporaneous.

The reviewer is correct in their reasoning, but again, all of this is already addressed in this section.

For example, we already have this on current lines 758-763.

*"Alternatively, the pattern of early deglaciation at KNS with later deglaciation in adjacent margins can be entirely explained by the magnitude of the late Holocene re-advance of the ice margin. For example, if the late Holocene readvance of KNS was of greater magnitude than that of adjacent margins, then the KNS terminus would overrun and rest upon a landscape that deglaciated earlier, and thus 10Be ages from outboard of the historical moraine would be older. The greater the magnitude of ice-margin readvance, the older the 10Be ages just outboard of the historical moraine will be."*

We think the reviewer may have glossed over this point a bit, that yes, all of this is certainly possible, and if we focus solely on the deglaciation ages outboard of the historical moraine, these ages may not give us a complete picture of ice-margin behavior. But if we include constraints that are able to provide some idea of ice-margin behavior when it was **slightly** inboard of today (lakes, proglacial bedrock) then the apparent differences in deglaciation based solely on the deglaciation constraints from beyond the historical moraine, no longer appear so extreme. A perfect example of this is at KNS where at least 3 kyr of time is represented in a very narrow lateral zone on the landscape.

*Lines 692-694: From what I read here you base the 5-2 ka BP "window" mostly on data from other studies – could you make a small comment on your own findings in according to this age constraint – based solely on your findings would the "window" not be a couple of thousand years longer, with initial retreat c. 7 ka BP? I assume some of the explanation lies in the discussion of different ice-margin environments, that you give in lines 722-734?*

Agreed, we can add a sentence that addresses this. It would most likely fit around current line 637. I think part of the confusion here is what one considers the KNS region. In the strictest sense the reviewer is correct in that our new dataset identifies a window between ~7 ka and 1 ka, but that also ignores the Saqqap region that we consider to be in the broader KNS region, and this manuscript has a much wider scope than just the KNS region (e.g. Fig 16 and our entire modeling effort). We can add a sentence that identifies a "window" solely based on new KNS data presented here.

Note, original Figure 16 is now Figure 17.

On lines 797-805 we added:

*"The combination of new [10]Be ages, records from proglacial-threshold lakes, and paired [14]C-[10]Be measurements from KNS and Qamanaarsuup Sermia defines a window between ~10-7 ka BP when the GrIS margin was likely near its present position. After 7 ka BP, the GrIS margin retreated inland before re-approaching its current configuration sometime in the last millennium. Considering the ice-margin constraints from nearby Saqqap Sermia to the north (Levy et al., 2017), the GrIS in the KNS region likely remained near its current position as late as ~5 ka."*

*Lines 748-764: It seems that model runs simulate an ice sheet minimum that to some degree fits with your data from the KNS region (as stated in the comment above) – could you briefly outline why/why not the models and your data fit/does not fit? Why you believe in the 5-2 minimum, and not an earlier retreat behind the present day margin?*

We do not think that any of the model runs do a particularly good job in the KNS region, and much of the modelling section (section 6.3) is devoted to pointing out the difficulties with incorporating calving in the KNS region as well as the influence of the KNS region's unique topography. We next discuss how these same issues don't really exist in the Kangerlussauq region north of KNS and how a few of the model runs appear to fit geologic constraints quite well. The modelling effort applies to the entire southwestern Greenland domain, which is why we compiled geologic constraints (many developed by our group) along southwestern Greenland beyond the KNS region so we could conduct a better assessment of the model results. When considering the geological constraints across southwestern Greenland, it appears to us that there is a pretty clear window of when the GrIS achieved its minimum (Fig. 16a; ~5-2 ka). Our group and Nicolaj Larsen's group has spent nearly a decade developing these geologic constraints and collectively, the GrIS minimum appears to be robustly constrained as spelled out in Section 6.2 (and Fig. 16a).

After reading this reviewer comment again, in particular the last statement, the only thing that comes to mind is that the reviewer is suggesting we ignore all of the fairly robust geological evidence that constrains the ice-sheet minimum to 5-2 ka BP (all of section 6.2 and now Figure 17) and focus on only a few deglaciation constraints from the immediate KNS region?

After reading Sections 6.2 (geologic data) and 6.3 (model-data comparison), we think we have more that sufficiently discussed the robustness of the 5-2 ka BP GrIS minimum and how the model simulations do a better job at matching the deglaciation constraints from the Kangerlussuaq region than the KNS region. No changes were made.

*I find that much of the text in the figures (place names, ages, lat/long) is difficult to read and could benefit from a larger font size.*

We will look into making some of the text larger.

We have slightly increased the text size in almost every figure.

*Table S3: Just a comment on the high accuracy of the sample thicknesses. As this might be the accuracy of the caliper (or whatever instrument you have used), I find it difficult to work with high accuracy numbers like this, on what I assume are rather uneven samples. Are sample thicknesses a mean of several measurements?*

We will add a footnote mentioning that we make several thickness measurements on samples and the accuracy given in the table reflects those measurements.

Change made. We mentioned in the footnote how sample thickness was calculated.

*Technical Corrections.*

Thank you for finding these. We will make the necessary corrections.

All minor technical corrections have been made.

*Text Line 60: Consider to use only surface or surfaces in this sentence. "..when those surfaces are exposed. . .*

Fixed.

*Line 93: "reworked" is spelled "re-worked" everywhere else in the text. Consider changing this.*

Fixed.

*Line 105: Check spelling of Kangâsarsuup throughout the text – some places you spell it with "â" and some places with "a". Further, judging from Figure 1, it seems that Kangâsarsuup Sermia is located about 20 km more than 35-40 km south of KNS?*

Fixed.

Not entirely sure what the reviewer is referring to with the distance. I adjusted the glacier label a bit on the figure but, if anything, the exact distance is less than 35-40 km. This might be a typo on our part. We just re-measured and its closer to 20-25 km depending on which specific reference point on the glacier one uses (updated in text).

*Line 109: "Trim-lines" is spelled without "-" a few places, consider changing for consistency.*

Yes, fixed.

*Line 115: You have "cosmogenic-nuclide" (as here) and "cosmogenic nuclide" (eg. line 800) in the text. I assume you want one consistent way of spelling?*

The hyphen is on purpose and both are correct and necessary based on usage. If "cosmogenic nuclide" is as a standalone noun, then no hyphen is needed. But for "cosmogenic-nuclide analysis" you would have the hyphen because technically the analysis refers to "cosmogenic-nuclide" and not solely "nuclide". Similar to "sea level" that is written without a hyphen, but "sea-level rise" should have a hyphen.

*Line 120: Consider using the abbreviation "GOOF" for Goose Feather Lake in the text. You use it in the caption to Figure 6, but nowhere else. Same comment for Marshall Lake.*

These abbreviations are meant more for the sediment core-labeling themselves and to identify the core site on Figure 3, and not structured to be full place-name abbreviations (e.g., KNS glacier). We only use these lake names a few times and prefer to just state the lake name. For example, in lines 221-223 we state:

*"In addition, we discuss two previously reported radiocarbon ages from Goose Feather Lake, located adjacent to Marshall Lake"*

Here, it would be awkward and unnecessary to substitute in GOOF and MAR. No changes made.

*Line 128: Space missing between "61" and "10Be".*

Fixed.

*Lines 162+165: I assume you mean "in situ 14C" here?*

Yes, this entire section is solely devoted to in situ $^{14}$C processing; the section title is "2.4 In situ 14C measurements."

*Line 323: I believe you mean production "rate" uncertainty?*

Yes, fixed.

*Reviewer #2: David Ullman*

*General Comments This manuscript presents a new cosmogenic isotope chronology to help constrain retreat of the western Greenland Ice Sheet (GrIS) during the earlier Holocene. It has long been surmised that the GrIS was smaller than present during parts of early Holocene, but constraining the magnitude of margin retreat and its timing*

*has been difficult due to the late-Holocene ("Historical") readvance covering much of the previously exposed surfaces. By utilizing 3 different cosmogenic nuclides (10Be, 14C, and 26Al) on sampling locations, the authors provide a clear picture of the complexity of nuclide inheritance, erosion, and exhumation that can sometimes confound cosmogenic nuclide interpretations. This is especially true in regions with a complex ice retreat history, such as that of the southwestern GrIS during the Holocene. This new dataset is compared with a robust compilation of existing exposure and lake chronologies from all along the historical extent of the southwestern GrIS that broadens the scope and significance into that of a larger regional signal. In addition to the chronological constraints provided in this manuscript, the authors also present new results from a high-resolution ice sheet model for the southwestern GrIS to help constrain the magnitude of retreat past the modern margin extent. These simulations are forced with temperature and precipitation from recent data assimilation efforts for Greenland. These model experiments explore a range of climate space, thus incorporating some parametric uncertainty in the results. Such a robust set of experiments allows for the exploration of a variety of possible solutions for Holocene margin retreat, a comparison with chronological constraints to constrain the model, and an assessment of model limitations (e.g. lack of iceberg calving). Generally, the manuscript provides a set of results that are consistent with improving our understanding of GrIS retreat during the Holocene. It is clear and well structured. I have a few concerns related to statistical significance and a need for a little more description about the modelling approach. After addressing these comments (mostly technical), I believe this manuscript is worthy of publication.*

Thanks for reviewing the manuscript and providing some constructive comments.

*Separate retreat timing from Kapisigdlit moraines? - One of the main conclusions is that the Kapisigdlit moraine deposition occurred with different timing for KNS (10.24 ± 0.36 ka) and Qamanaarsuup Sermia (9.57 ± 0.38 ka), thus suggesting a new mode of GrIS moraine deposition during the Holocene. However, don't these two mean ages overlap at 1σ? The overlap is small, but an overlap nonetheless. It seems statistically possible that these two ages are equivalent. At the very least, it would be good to show t-test statistics to help show the level of statistical difference between these ages. If the difference between these ages is not significant, then some of the wording of the paper may need to be modified to indicate the possibility of synchronous moraine deposition.*

Yes, this is a good catch by the reviewer. This is certainly something we have thought about but did not explain well in the text. The reviewer is absolutely correct that the two mean ages barely overlap at 1-sigma, no question. What we failed to mention in the text, and what we can address in the revision, is the various differences in moraine setting and the distribution of [10]Be ages. From our extensive experience in southwestern Greenland, we have found that the [10]Be ages from erratic boulders perched on bedrock immediately inside a moraine serve an extremely close limiting age on the moraine itself. In fact, often [10]Be ages from moraine boulders and inboard erratics are statistically identical. This is the case with the older Kapisigdlit moraine that the reviewer mentions. However, at Qamanaarsup Sermia, while the mean age from the moraine boulders themselves barely overlaps the mean age of the Kapisigdlit moraine boulders (what the reviewer is referring to here), the inboard erratic ages at Qamanaarsup Sermia are much younger (~9.3 ka) than the inboard erratic ages for the Kapisigdlit moraine. Considering what we typically see on Greenland, if the moraine at Qamanaarsup Sermia were actually a ~10.3-10.2 ka moraine, we might expect the inboard erratic ages to be ~10.2-10.0 ka, similar to what we see at the Kapisigdlit moraine. However, because the erratics ages at Qamanaarsup Sermia are tightly clustered, we think ~9.3 ka is a robust close minimum constraint on the age of the moraine at Qamanaarsup Sermia. In other words, this moraine was likely deposited just prior to ~9.3 ka.

On the other hand, there is a decent amount of scatter in the Qamanaarsup Sermia moraine boulder dataset and the multi-crest nature of the moraine here points to an oscillating or stagnating ice margin. Therefore, it possible that there may be several distinct episodes of ice-margin advance/stillstand represented on the landscape and by combining all of our [10]Be ages, we have inadvertently incorporated a small degree of inheritance into our preferred moraine age. For example, if this moraine complex represents advances of the ice margin at 10.3 ka and 9.3 ka, and boulders from both these episodes are getting re-worked with each other, then one might expect to get a mean moraine age of 9.6 or 9.7 ka if one samples boulders across the entire moraine complex. Regardless, the reviewer makes a good point here that this isn't the easiest comparison to make between different moraine ages, and something we are a bit unaccustomed to in southwestern Greenland. However, we actually think that while the Kapisigdlit moraine was likely deposited at 10.3-10.2 ka, the moraine complex at Qamanaarsup Sermia

was deposited either at 9.57 ± 0.38 ka (stated in text), or we inadvertently sampled a composite feature and part of our boulder population actually constrains an ice advance ca. 9.3 ka, which is further supported by the 10Be ages from erratic boulders resting immediately inboard of the moraine that are also 9.3 ka. We can expand on this in the text.

Looking at the reviewer comment, our response, and our initial manuscript text again, we have clarified a few things. The overlap that the reviewer is referring to, which we failed to realize when responding to the comment, is referring to moraine ages that include the uncertainty in the production-rate calibration. Because any shifts in the production rate would result in systematic changes to 10Be ages on all dated features, one does not include this production-rate uncertainty when comparing the ages of 10Be-dated features to each other. When the production rate uncertainty is not included, the features the reviewer mentions are barely distinguishable at 1-sigma uncertainties. Nonetheless, because these ages are so close, much of this discussion above in our response is still applicable and probably warrants further consideration.

On lines 689-706 we have addressed this by adding a significant block of text"

*"Another possibility is that the Qamanaarsuup Sermia moraine complex is an amalgamation of moraines relating to chronologically distinct advances or stillstands of the ice margin. The numerous and tightly packed moraines here, versus the more well-defined Kapisigdlit stade moraine at KNS, is suggestive of a stagnating or oscillating ice margin. Moreover, our moraine boulder dataset contains more scatter than we typically observe in southwest Greenland (e.g., Young et al., 2020a), suggesting it is possible that we sampled moraine boulders from 2 or more distinct advances. In this case, combining all of our $^{10}$Be ages from moraine boulders at Qamanaarsuup Sermia would inadvertently mask the timing of 2 or more advances; for example, if advances occurred ca. 10.4-10.3 ka BP and 9.3-9.0 ka BP, combining all $^{10}$Be ages might result in an average $^{10}$Be age of ~9.7 ka BP, especially if moraine boulders from each advance are reworked by the ice margin. In several instances in southwest Greenland, $^{10}$Be ages from erratics just inboard of a moraine are statistically identical to the moraine boulders themselves (e.g., Young et al., 2013b; Young et al., 2020a), perhaps indicating that our minimum-limiting age of 9.29 ± 0.07 ka BP constrains an advance of the GrIS in this sector to ~9.3 ka BP (moraine closest to erratics) and moraines located farther away from the ice margin might relate to an advance of the GrIS closer in age to our maximum-limiting $^{10}$Be ages (10.29 ± 0.14 ka BP; Fig. 3). Our $^{10}$Be ages from moraine boulders at Qamanaarsuup Sermia, however, show no trend across moraines or with distance from the ice margin. We prefer the more conservative interpretation that acknowledges that if two or more distinct advances occurred, we cannot resolve these advances with our dataset. We can confidently say that all moraines were deposited between 10.29 ± 0.14 ka BP and 9.29 ± 0.07 ka BP, and a moraine age of 9.57 ± 0.33 ka BP is consistent with these bracketing ages."*

Also, on line 709-712 we added:

*"Note that these moraine ages overlap at 1$\sigma$ only when including the $^{10}$Be production-rate uncertainty, which results in systematic shifts in age and is only needed when comparing these moraine ages to independent chronometers; these moraines are distinguishable at 1$\sigma$ in $^{10}$Be space."*

*Ice Sheet Model Methods (section 2.7) – I think it would be good to offer a little more description on the nature of the model setup and experiment design. I realize this model is extensively described in Cuzzone et al (2019) and Briner et al. (2020), but there are some crucial distinctions that could be added here that would help in understanding the results. For example: -What is the nature of the surface mass balance calculations? PDD? -Provide a general description of the flow dynamics. -mention the lack of calving in the methods (it gets brought up later, but it would be good to mention such model limitations in the methods -More description of the 9 model combinations/experiments (line 238-239). How were these 9 permutations selected? How do they differ?*

Yes, we can expand on the model experimental design. Initially, we just cited the relevant references that have been recently published in order to help limit the length of this already long manuscript. Yet, we do

recognize that a few more details about the model and climatology set-up in this manuscript might be helpful to readers. Following the first paragraph of Section 2.7, added text will include:

*"The higher-order approximation (Blatter 1995; Pattyn 2003) is used to solve the momentum balance equations. We use an enthalpy formulation (Ashwanden et al., 2012) to simulate the thermal evolution of the ice, using geothermal heat flux from Shapiro and Ritzwoller et al. (2004). The ice model uses quadratic finite elements (P1 x P2) along the z axis for the vertical interpolation, which allows the ice-sheet model to capture sharp thermal gradients near the bed, while reducing computational costs associated with running a linear vertical interpolation with increased vertical layers (Cuzzone et al., 2018). Subelement grounding-line migration (Serrousi et al., 2013) is included in these simulations, however, due to prohibitive costs associated with running a higher-order ice model over paleoclimate timescales these simulations do not include calving parameterizations nor any submarine melting of floating ice.*

*Nine ice-sheet simulations are forced with paleoclimate reconstructions from Badgeley et al. (2020) who used paleoclimate data assimilation to merge information from paleoclimate proxies and global climate models. The temperature reconstructions rely on oxygen-isotope records from eight ice cores, the precipitation reconstructions use accumulation records from five ice cores, and all are guided by spatial relationships derived from the transient climate-model simulation TraCE-21ka (Liu et al., 2009; He et al., 2013). The climate reconstructions are shown to be in good agreement with independent paleoclimate proxy data (Badgeley et al., 2020 and references therein). Along with a main temperature and precipitation reconstruction, Badgeley et al. (2020) provide two sensitivity precipitation reconstructions due to uncertainty in the accumulation records and four sensitivity temperature reconstructions due to uncertainty in the relationship between oxygen isotopes and surface air temperature. Briner et al. (2020) pair three of the temperature reconstructions with each of the three precipitation reconstructions to yield nine combinations that are used as transient climate boundary conditions to force the nine ice sheet simulations. Two of the five temperature reconstructions were not used because they yield Younger Dryas ice-sheet margins that are inconsistent with geologic data.*

*In order to compute the surface mass balance from temperature and precipitation, a positive degree day (PDD) method is used (Tarasov and Peltier, 1999). We use degree-day factors of 4.3 mm °C$^{-1}$ day$^{-1}$ for snow and 8.3 mm °C$^{-1}$ day$^{-1}$ for ice, with allocation for the formation of superimposed ice (Janssens and Huybrechts, 2000). A lapse rate of 6 °C km$^{-1}$ is used to adjust the temperature of the climate forcings to the ice surface elevation."*

All of the above text was added on lines 245-268.

*Discussion of marine Terminating Dynamics (lines 748-764) – How much of the full model domain is influenced by marine dynamics and iceberg calving? Since these regions minimize retreat in the model, is there a way to show or discuss how much of the model domain would be influenced by this model limitation. Are any of the margins still marine-terminating at the minimum Holocene extent?*

We are attaching a figure that shows our model domain, and our initial marine and land terminating regions. The marine areas, where the ice base is below 0 are shown in red. Land terminating regions, where the ice base is above sea-level are shown in blue. Throughout our simulations, RSL changes, and therefore sea-level varies. So marine and land terminating portions of the domain will change through time. But this figure should illustrate that we capture (with our model mesh) marine terminating margins in many of the fjord regions. Our 9 different simulations all have distinct ice extents during the Holocene minimum. Because we capture the KNS fjord geometry reasonably, these margins would be marine terminating, and some portions of the outlet glaciers are floating in our simulations (Although we do not simulate calving, we do simulate grounding line migration). If it is necessary to show, we can try to put together a figure showing elements where there is floating ice during the simulations. But we note that this makes up a very small % of the model domain (<<1%).

It is hard to determine exactly how our model domain would be impacted by this limitation (no calving) without performing the experiments, however, areas at the ice front and upstream might be affected by

calving even in warm climates since the ice is fast flowing and tends to maintain contact with the ocean during those times.

Looked through this comment and our response again and agree that we could add a few lines that address this. However, after reading our manuscript text again, we wonder if the reviewer meant to point to some of our text slightly before the lines that they mention? Nevertheless, in lines 855-862 we added:

*"We also note that there are portions of the model domain that are below sea level and susceptible to marine influence (Fig. S1). Throughout our simulations, relative sea level varies through time, which could change the portions of our model domain that are marine versus land terminating. Although marine processes (e.g., submarine melting of floating ice and calving) are not included in our simulations, we do include grounding line migration, and our model also simulates floating ice at outlet glacier termini through the Holocene. It is difficult to determine how our simulations would be impacted by including marine processes without performing additional experiments, but areas at the ice front and immediately upstream could be particularly affected in warmer climates coincident with the Holocene minimum extent as fast-flowing ice tends to maintain contact with the ocean."*

This text has been added about 2 paragraphs prior to where the reviewer was referring to in the initial manuscript text, but we think this is a much more appropriate place for this brief discussion.

In addition, we have now included Supplementary Figure 1, which outlines the portion of our model domain that is above sea level versus below sea level.

Technical comments

Again, thanks for catching these. We will address them in the revision.

*Line 201 – Specifically, what is the production rate uncertainty that was used in quadrature? Is it a constant percentage? Or is it spatially varying? Line 365 – "mean age of 10.20 ± 0.14 ka (10.27 ± 0.23 ka with production-rate uncertainty)." Why are the mean ages (10.20 and 10.27) different? Shouldn't the production rate uncertainty added in quadrature only effect the uncertainty value? Also, double check on these values are appropriately displayed in Figure 2.*

We will clarify. 1.8% used in quadrature. Also, thanks for catching the moraine age typos, will double-check.

This was just a typo. Everything has been checked.

*Lines 373-377 – Would it be possible to include this early photograph with permission? It would be nice to see this photograph annotated to show the ice extent and trim-line as described in the text.*

We can inquire, but make no promises here – although we agree this would be nice. This is an old photograph that currently exists in a GEUS bulletin focused on the KNS region that is somewhat widely available (Weidick, 2012).

Great suggestion by the reviewer. Thanks to a colleague of ours, Ole Bennike, we were able to obtain a digital copy of this photograph from the mid 1850s. We have included this photo as a new standalone figure (Fig. 10). Panel A shows the original photo looking up towards KNS and panel B is zoomed in where the historical maximum trimlines are clearly visible. Caption has credited our colleague and the National Museum in Copenhagen where the original photo is housed.

*Line 437 – I think the use of the phrase "more proximal" is confusing here. These high elevation sites are less proximal from the historical limit, when considering ice position. I would say that the recently deglaciated sites are "more proximal" than the historical limit. Do you mean that the high elevation sites are closer to the historical maximum limit? Lines 508-509 – "we favor an interpretation that couples less site exposure over significant amounts of subglacial abrasion" – I think the use of the word "couples" is confusing here. What is being coupled to what?*

We can clean up the wording here.

On line 501 the wording has been changed to "*directly adjacent to the historical maximum limit*"

Right, "couples" is the wrong word here, perhaps leftover from a previous draft. Changed to "favors" on line 589

*Figure 1 – The orange diamond of Weidick et al (2012) is very hard to distinguish from other yellow circles. Consider using an alternative symbol. Figure 2 & Figure 3 – In general it can be difficult to distinguish between italicized outliers and non-outliers. Is there a different way to distinguish outliers other than italics? Figure 17c and 17d – What does this distance axis mean? Is it distance from the coast, or some other arbitrary point? Would be good to clarify in the caption. Figure 17c and 17d – Is there a way to better display the present day location? The yellow dots are hard to see on the first pass Figure 17c-17f – On each of the model result figures, it is hard to distinguish between the green lines and the blue lines. Would it be possible to use a more distinct color gradient for these groupings of simulations?*

These minor figure edits are pretty straightforward. Our model results color scheme was used in order to remain consistent with the color scheme used in Briner et al (2020), but we can address this.

Fig. 1: Orange diamond has been changed to a blue star and we added a "rw" abbreviation in the label for "reworked"

Fig. 2; Fig. 3; Fig. 7. We have kept the admittedly tough to distinguish italics for outliers, but in addition, we have marked outliers with black boxes. Figure caption text has been updated.

Fig. 18 (previous Fig. 17): Caption has been updated making it clear that y-axis is "distance from the coast". We have also marked the modern margin on these plots with an orange star and an abbreviation for "modern" (m). In addition, we assigned different symbols for the geologic data points on panels c and d. Lastly, per the reviewer request, we changed the color gradient for panels c-f. These colors should be easier to distinguish for the reader. Same color gradient is now used in the bottom panel of Figure 17 (previous Fig. 16),